# Place-cell capacity and volatility with grid-like inputs

Man Yi Yim[1,2,3], Lorenzo A Sadun[4], Ila R Fiete[1,3]*, Thibaud Taillefumier[1,2,4]*

[1]Center for Theoretical and Computational Neuroscience, University of Texas, Austin, United States; [2]Department of Neuroscience, University of Texas, Austin, United States; [3]Department of Brain and Cognitive Sciences and McGovern Institute, MIT, Austin, United States; [4]Department of Mathematics and Neuroscience, The University of Texas, Austin, United States

**Abstract** What factors constrain the arrangement of the multiple fields of a place cell? By modeling place cells as perceptrons that act on multiscale periodic grid-cell inputs, we analytically enumerate a place cell's *repertoire* – how many field arrangements it can realize without external cues while its grid inputs are unique – and derive its *capacity* – the spatial range over which it can achieve any field arrangement. We show that the repertoire is very large and relatively noise-robust. However, the repertoire is a vanishing fraction of all arrangements, while capacity scales only as the sum of the grid periods so field arrangements are constrained over larger distances. Thus, grid-driven place field arrangements define a large response scaffold that is strongly constrained by its structured inputs. Finally, we show that altering grid-place weights to generate an arbitrary new place field strongly affects existing arrangements, which could explain the volatility of the place code.

*For correspondence:
fiete@mit.edu (IRF);
ttaillef@austin.utexas.edu (TT)

**Competing interest:** The authors declare that no competing interests exist.

## Introduction

As animals run around in a small familiar environment, hippocampal place cells exhibit localized firing fields at reproducible positions, with each cell typically displaying at most a single firing field (*O'Keefe and Dostrovsky, 1971*; *Wilson and McNaughton, 1993*). However, a place cell generates multiple fields when recorded in single large environments (*Fenton et al., 2008*; *Park et al., 2011*; *Rich et al., 2014*) or across multiple environments (*Muller et al., 1987*; *Colgin et al., 2008*), including different physical and nonphysical spaces (*Aronov et al., 2017*).

Within large spaces, the locations seem to be well-described by a random process (*Rich et al., 2014*; *Cheng and Frank, 2011*), and across spaces the place-cell codes appear to be independent or orthogonal (*Muller et al., 1987*; *Colgin et al., 2008*; *Alme et al., 2014*), also potentially consistent with a random process. However, a more detailed characterization of possible structure in these responses is both experimentally and theoretically lacking, and we hypothesize that there might be structure imposed by grid cells in place field arrangements, especially when spatial cues are sparse or unavailable.

Our motivation for this hypothesis arises from the following reasoning: grid cells (*Hafting et al., 2005*) are a critical spatially tuned population that provides inputs to place cells. Their codes are unique over very large ranges due to their modular, multi-periodic structure (*Fiete et al., 2008*; *Sreenivasan and Fiete, 2011*; *Mathis et al., 2012*). They appear to integrate motion cues to update their states and thus reliably generate fields even in the absence of external spatial cues (*Hafting et al., 2005*; *McNaughton et al., 2006*; *Burak and Fiete, 2006*; *Burak and Fiete, 2009*). Thus, it is possible that in the absence of external cues spatially reliable place fields are strongly influenced by grid-cell inputs.

To generate theoretical predictions under this hypothesis, we examine here the nature and strength of potential constraints on the arrangements of multiple place fields driven by grid cells. On the one hand, the grid inputs are nonrepeating (unique) over a very large range that scales exponentially with the number of grid modules (given roughly by the product of the grid periods), and thus rich (*Fiete et al., 2008*; *Sreenivasan and Fiete, 2011*; *Mathis et al., 2012*); are these unique inputs sufficient to enable arbitrary place field arrangements? On the other hand, this vast library of unique coding states lies on a highly nonlinear, folded manifold that simple read-outs might not be able to discriminate (*Sreenivasan and Fiete, 2011*). This nonlinear structure is a result of the geometric, periodically repeating structure of individual modules (*Stensola et al., 2012*); should we expect place field arrangements to be constrained by this structure?

These questions are important for the following reason: a likely role of place cells, and the view we espouse here, is to build consistent and faithful associations (maps) between external sensory cues and an internal *scaffold* of motion-based positional estimates, which we hypothesize is derived from grid inputs. This perspective is consistent with the classic ideas of cognitive maps (*O'Keefe and Nadel, 1978*; *Tolman, 1948*; *McNaughton et al., 2006*) and also relates neural circuitry to the computational framework of the simultaneous localization and mapping (SLAM) problem for robots and autonomously navigating vehicles (*Leonard and Durrant-Whyte, 1991*; *Milford et al., 2004*; *Cadena et al., 2016*; *Cheung et al., 2012*; *Widloski and Fiete, 2014*; *Kanitscheider and Fiete, 2017a*; *Kanitscheider and Fiete, 2017b*; *Kanitscheider and Fiete, 2017c*). We can view the formation of a map as 'decorating' the internal scaffold with external cues. For this to work across many large spaces, the internal scaffold must be sufficiently large, with enough unique states and resolution to build appropriate maps.

A self-consistent place-cell map that associates a sufficiently rich internal scaffold with external cues can enable three distinct inferences: (1) allow external cues to correct errors in motion-based location estimation (*Welinder et al., 2008*; *Burgess, 2008*; *Sreenivasan and Fiete, 2011*; *Hardcastle et al., 2014*), through cue-based updating; (2) predict upcoming external cues over novel trajectories through familiar spaces by exploiting motion-based updating (*Sanders et al., 2020*; *Whittington et al., 2020*); and (3) drive *fully intrinsic* error correction and location inference when external spatial cues go missing and motion cues are unreliable by imposing self-consistency (*Sreenivasan and Fiete, 2011*).

In what follows, we characterize which arrangements of place fields are realizable based on grid-like inputs in a simple perceptron model, in which place cells combine their multiple inputs and make a decision on whether to generate a field ('1' output) or not ('0' output) by selecting input weights and a firing threshold (*Figure 1A,B*). However, in contrast to the classical perceptron results, which are derived under the assumption of random inputs that are in general position (a property related to the linear independence of the inputs), grid inputs to place cells are structured, which adds substantial complexity to our derivations.

We show analytically that each place cell can realize a large repertoire of arrangements across all possible space where the grid inputs are unique. However, these realizable arrangements are a special and vanishing subset of all arrangements over the same space, suggesting a constrained structure. We show that the capacity of a place cell or spatial range over which all field arrangements can be realized equals the sum of distinct grid periods, a small fraction of the range of positions uniquely encoded by grid-like inputs. Overall, we show that field arrangements generated from grid-like inputs are more robust to noise than those driven by random inputs or shuffled grid inputs.

Together, our results imply that grid-like inputs endow place cells with rich and robust spatial scaffolds, but that these are also constrained by grid-cell geometry. Rigorous proofs supporting all our mathematical results are provided in Appendix 1. Portions of this work have appeared previously in conference abstract form (*Yim et al., 2019*).

## Modeling framework
### Place cells as perceptrons

The perceptron model (*Rosenblatt, 1958*) idealizes a neuron as computing a weighted sum of its inputs ($x_j \in \mathbb{R}^N$) based on learned input weights ($w \in \mathbb{R}^N$) and applying a threshold ($\theta$) to generate a binary response that is above or below threshold. A perceptron may be viewed as separating its



**Figure 1.** The grid-like code and modeling place cells as perceptrons. (**A**) Grid-like inputs and a conceptual view of a place cell as a perceptron: each place cell combines its feedforward inputs, including periodic drive from grid cells (responses simplified here to one spatial dimension) of various periods and phases (blue and red cells are from modules with different periods) to generate location-specific activity that might be multiply peaked across large spaces. Can these place fields be arranged arbitrarily? (**B**) Idealization of a place cell as a perceptron: in discretized 1-D space, the grid-like inputs are discrete patterns that for simplicity we consider to be binary; place fields are assigned at locations where the weighted input sum exceeds a threshold $\theta$. A place field arrangement can be considered as a set of binarized output labels (1 for each field, 0 for non-field locations) for the set of input patterns. We count field arrangements over the range of locations where the grid-like inputs have unique states; for two modules with periods $\{2, 3\}$, this range is 6 (the LCM of the grid periods). LCM = least common multiple; GCD = greatest common divisor.



**Figure 2.** Linear separability, counting dichotomies, and separating capacity for perceptrons. (**A**) A set of patterns (locations given by circles) that are assigned positive and negative labels (filled versus open), called a dichotomy of the patterns, is realizable by a perceptron if positive examples can be linearly separated (by a hyperplane) from the rest. The perceptron weights $w$ encode the direction normal to the separating hyperplane, and the threshold sets its distance from the origin. (**B**) An example with input dimension $N = 3$ (the input dimension is the length of each input pattern vector, which equals the number of input neurons). When placed randomly, $P = 4$ random real-valued patterns optimally occupy space and are said to be in general position (left); these patterns define a tetrahedron and all dichotomies are linearly separable. By contrast, structured inputs may occupy a lower-dimensional subspace and thus not lie in general position (right). This square configuration exhibits unrealizable dichotomies (as in **A**, bottom). (**C**) Cover's results (**Cover, 1965**): for patterns in general position, the number of realizable dichotomies is $2^P$, and thus the fraction of realizable dichotomies relative to all dichotomies is 1, when the number of patterns is smaller than the input dimension ($P < N$). The fraction drops rapidly to zero when the number of patterns exceeds twice the input dimension (the separating capacity).

high-dimensional input patterns into two output categories ($y \in \{0, 1\}$) (**Figure 2A**), with the categorization depending on the weights and threshold so that sufficiently weight-aligned input patterns fall into category 1 and the rest into category 0:

$$y(\boldsymbol{x}_j) = \begin{cases} 1 & \text{if } \boldsymbol{w} \cdot \boldsymbol{x}_j - \theta > 0, \\ 0 & \text{otherwise.} \end{cases} \tag{1}$$

If each partitioning of inputs into the $\{0, 1\}$ categories is called a dichotomy, then the only dichotomies 'realizable' by a perceptron are those in which the inputs are linearly separable – that is, the set of inputs in category 0 can be separated from those in category 1 by some linear hyperplane (**Figure 2**). Cover's counting theorem (**Cover, 1965**; **Vapnik, 1998**) provides a count of how many dichotomies a perceptron can realize if input patterns are random (more specifically, in general position). A set of patterns $\{\boldsymbol{x}_1, \dots, \boldsymbol{x}_P\}$ in an $N$-dimensional space is in general position if no subset of size smaller than

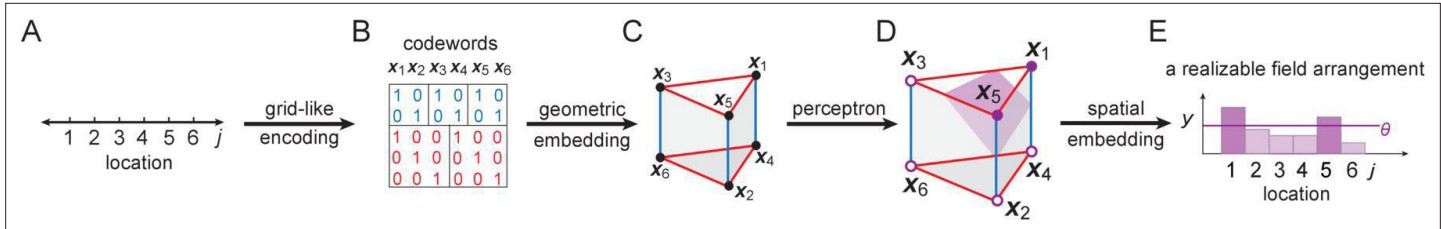

**Figure 3.** Our overall approach. (**A, B**) Locations (indexed by $j$) map onto grid-like coding states ($\{x_i\}$, defining the grid-like codebook) through the assignment of spatially periodic responses to grid cells, with different cells in a module having different phases and different modules having different periods. (This example: periods 2,3.) (**C**) The patterns in the grid-like codebook form some nonrandom, geometric structure. (**D**) The geometric structure defines which dichotomies are realizable by separating hyperplanes. (**E**) A realizable dichotomy in the abstract codebook pattern space, when mapped back to spatial locations, corresponds to a realizable field arrangement. Shown is a place field arrangement realized by the separating hyperplane from (**D**). Similarly, an unrealizable field arrangement can be constructed by examination of (**D**): it would consist of, for instance, fields at locations $j = 1, 2$ only (or, e.g., at $j = 3, 4, 6$ only): vertices that cannot be grouped together by a single hyperplane.

$N + 1$ is affinely dependent. In other words, no subset of $n + 1$ points lies in a $(n-1)$-dimensional plane for all $n \le N$. (**Figure 2B**) and establishes that for $P \le N$ patterns, every dichotomy is realizable by a perceptron – this is the perceptron capacity (**Figure 2C**). For $P = 2N$, exactly half of the $2^P$ possible dichotomies are realizable; when $P \gg N$ for fixed $N$, the realizable dichotomies become a vanishing fraction of the total (**Figure 2C**).

Here, to characterize the place-cell scaffold, we model a place cell as a perceptron receiving grid-like inputs (**Figure 1B**). Across space, a particular 'field arrangement' is realizable by the place cell if there is some set of input weights and a threshold (**Lee et al., 2020**) for which its summed inputs are above threshold at only those locations and below it at all others (**Figure 1A,B**). We call an arrangement of exactly $K$ fields a 'K-field arrangement.'.

In the following, we answer two distinct but related questions: (1) out of all potential field arrangements over the entire set of unique grid inputs, how many are realizable, and how does the realizable fraction differ for grid-like inputs compared to inputs with matched dimension but different structure? This is akin to perceptron function counting (**Cover, 1965**) with structured rather than general-position inputs and covers constraints within and across environments. We consider all arrangements regardless of sparsity, on one extreme, and $K$-field (highly sparse) arrangements on the other; these cases are analytically tractable. We expect the regime of sparse firing to interpolate between these two regimes. (2) Over what range of positions is any field arrangement realizable? This is analogous to computing the perceptron-separating capacity (**Cover, 1965**) for structured rather than general-position inputs.

Although the structured rather than random nature of the grid code adds complexity to our problem, the symmetries present in the code also allow for the computation of some more detailed quantities than typically done for random inputs, including capacity computations for dichotomies with a prescribed number of positive labels (K-field arrangements).

## Results

Our approach, summarized in **Figure 3**, is as follows: we define a mapping from space to grid-like input codes (**Figure 3A,B**), and a generalization to what we call modular-one-hot codes (**Figure 3B**). We explore the geometric structure and symmetries of these codes (**Figure 3C**). Next, we show how separating hyperplanes placed on these structured inputs by place-cell perceptrons permits the realization of some dichotomies (**Figure 3D**) and thus some spatial field arrangements (**Figure 3E**), but not others, and obtains mathematical results on the number of realizable arrangements and the separating capacity.

### The structure of grid-like input patterns

Grid cells have spatially periodic responses (**Figure 1A,B**). Cells in one grid module exhibit a common spatial period but cover all possible spatial phases. The dynamics of each module are low-dimensional (**Fyhn et al., 2007**; **Yoon et al., 2013**), with the dynamics within a module supporting and stabilizing

a periodic phase code for position. Thus, we use the following simple model to describe the spatial coding of grid cells and modules: a module with spatial period $\lambda_m$ (in units of the spatial discretization) consists of $\lambda_m$ cells that tile all possible phases in the discretized space while maintaining their phase relationships with each other. Each grid cell's response is a $\{0, 1\}$-valued periodic function of a discretized 1D location variable (indexed by $j$); cell $i$ in module $m$ fires (has response 1) whenever $(j - i) \mod \lambda_m = 0$, and is off (has response 0) otherwise (*Figure 1B*). The encoding of location $j$ across all Mm modules is thus an $N$-dimensional vector $x_j$, where $N = \sum_{m=1}^{M} \lambda_m$. Nonzero entries correspond to co-active grid cells at position $j$. The total number of unique grid patterns is $L = \mathrm{LCM}(\{\lambda_1, \ldots, \lambda_M\})$, which grows exponentially with $M$ for generic choices of the periods $\{\lambda_m\}$ (*Fiete et al., 2008*). We refer to $L$ as the 'full range' of the code. We call the full ordered set of unique coding states $\{x_j\}$ the grid-like 'codebook' $X_g$.

Because $X_g$ includes all unique grid-like coding states across modules, it includes all possible relative phase shifts or 'remappings' between grid modules (*Fiete et al., 2008*; *Monaco et al., 2011*). Thus, this full-range codebook may be viewed as the union of all grid-cell responses across all possible space and environments. We assume implicitly that 2D grid modules do not rotate relative to each other across space or environments. Permitting grid modules to differentially rotate would lead to more input pattern diversity, more realizable place patterns, and bigger separating capacity than in our present computations.

The grid-like code belongs to a more general class that we call 'modular-one-hot' codes. In a modular-one-hot code, cells are divided into modules; within each module only one cell is allowed to be active (the within-module code is one-hot), but there are no other constraints on the code. With $m = 1, ..., M$ modules of sizes $\lambda_m$, the modular-one-hot codebook $X_{mo}$ contains $P = \prod_{m=1}^{M} \lambda_m$ unique patterns, with $P \geq L$ for a corresponding grid-like code. When $\{\lambda_1, \cdots, \lambda_M\}$ are pairwise coprime, $P = L$ and the grid-like and modular-one-hot codebooks contain identical patterns. However, even in this case, modular-one-hot codes may be viewed as a generalization of grid-like codes as there is no notion of a spatial ordering in the modular-one-hot codes, and they are defined without referring to a spatial variable.

Of our two primary questions introduced earlier, question (1) on counting the size of the place-cell repertoire (the number of realizable field arrangements) depends only on the geometry of the grid coding states, and not on their detailed spatial embedding (i.e., it depends on the mappings in *Figure 3B–D*, but not on the mapping between *Figure 3A,B,D,E*). In other words, it does not depend on the spatial ordering of the grid-like coding states and can equivalently be studied with the corresponding modular-one-hot code instead, which turns out to be easier. Question (2), on place-cell capacity (the spatial range $l \leq L$ over which any place field arrangement is realizable), depends on the spatial embedding of the grid and place codes (and on the full chain of *Figure 3A-E*). For $l < L$, this would correspond to a particular rather than random subset of $X_{mo}$, thus we cannot use the general properties of this generalized version of the grid-like code.

## Alternative codes

In what follows, we will contrast place field arrangements that can be obtained with grid-like or modular-one-hot codes with arrangements driven by alternatively coded inputs. To this end, we briefly define some key alternative codes, commonly encountered in neuroscience, machine learning, or in the classical theory of perceptrons. For these alternative codes, we match the input dimension (number of cells) to the modular-one-hot inputs (unless stated otherwise).

Random codes $X_r$, used in the standard perceptron results, consist of real-valued random vectors. These are quite different from the grid-like code and all the other codes we will consider, in that the entries are real-valued rather than $\{0, 1\}$-valued like the rest. A set of up to $N$ random input patterns in $N$ dimensions is linearly independent; thus, they have no structure up to this number.

Define the one-hot code $X_{oh}$ as the set of vectors with a single nonzero element whose value is 1. It is a single-module version of the modular-one-hot code or may be viewed as a binarized version of the random patterns since $N$ patterns in $N$ dimensions are linearly independent. In the one-hot code, all neurons are equivalent, and there is no modularity or hierarchy.

Define the 'binary' code $X_b$ as all possible binary activity patterns of $N$ neurons (*Figure 4B*, right). We distinguish $\{0, 1\}$-valued codes from binary codes. In the binary code, each cell represents a specific position (register) according to the binary number system. Thus, each cell represents numbers



**Figure 4.** The geometry of structured inputs. (**A**) Though the grid-like input patterns in the example *Figure 1B* are 5D, they have a simplified structure that can be embedded as a 3D triangular prism given by the product of a 2-graph (blue, middle) and 3-graph (red, right) because of the independently updating modular structure of the code. (**B**) Different codebooks and their geometries. At one end of the spectrum (left), one-hot codes consist of a single module; they are not hierarchical, and their geometry is always an elementary simplex (left). Grid cells and modular-one-hot codes (middle) have an intermediate level of hierarchy and consist of an orthogonal product of simplices. At the opposite end, the binary code (right) is the most hierarchical, consisting of as many modules as cells; the code has a hypercube geometry: vertices (codewords or patterns) on each face of the hypercube are far from being in general position.

at a different resolution, differing in powers of 2, and the code has no neuron permutation invariance since each cell is its own module; thus, it is both highly hierarchical and modular.

The grid-like and modular-one-hot codes exhibit an intermediate degree of modularity (multiple cells make up a module). If the modules are of a similar size, the code has little hierarchy.

## The geometry of grid-like input patterns

We first explore question (1). The modular-one-hot codebook $X_{\text{mo}}$ is invariant to permutations of neurons (input matrix rows) within modules, but rows cannot be swapped across modules as this would destroy the modular structure. It is also invariant to permutations of patterns (input matrix columns $\boldsymbol{x}_j$). Further, the codebook includes all possible combinations of states across modules, so that modules function as independent encoders. These symmetries are sufficient to define the geometric arrangement of patterns in $X_{\text{mo}}$, and the geometry in turn will allow us to count the number of field arrangements that are realizable by separating hyperplanes.

To make these ideas concrete, consider a simple example with module sizes $\{2, 3\}$ (corresponding to the periods in the grid-like code), as in *Figure 1B* and *Figure 3B*. Independence across modules causes the code to have a product structure in the code: the codebook consists of six states that can be obtained as products of the within-module states: $\{10100, 10010, 10001, 01100, 01010, 01001\}$ = $\{10, 01\} \times \{100, 010, 001\}$, where $\{10, 01\}$ and $\{100, 010, 001\}$ are the coding states within the size-2 and size-3 modules, respectively. We represent the two states in the size-2 module by two vertices, connected by an edge, which shows allowed state transitions within the module (*Figure 4A*, right). Similarly, the three states in the size-3 module and transitions between them are represented by a triangular graph (*Figure 4A*, right). The product of this edge graph and the triangle graph yields the full codebook $X_{\text{mo}}$. The resulting product graph (*Figure 4A*, left) is an orthogonal triangular prism with vertices representing the combined patterns.

This geometric construction generalizes to an arbitrary number of modules $M$ and to arbitrary module sizes (periods) $\lambda_m$, $1 \leq m \leq M$: by permutation invariance of neurons within modules, and independence of modules, the patterns of the codebook $X_{\text{mo}}$ and thus of the corresponding grid-like codebook $X_g$ always lie on the vertices of some convex polytope (e.g., the triangular prism), given by an orthogonal product of $M$ simplices (e.g., the line and triangle graphs). Each simplex represents one of the modules, with simplex dimension $\lambda_m - 1$ for module size (period) $\lambda_m$ (see Place-cell capacity and volatility with grid-like inputs).

This geometric construction provides some immediate results on counting: in a convex polytope, any vertex can be separated from all the rest by a hyperplane; thus, all one-field arrangements are realizable. Pairs of vertices can be separated from the rest by a hyperplane if and only if the pair is directly connected by an edge (*Figure 3D*). Thus, we can now count the set of all realizable two-field arrangements as the number of adjacent vertices in the polytope. Unrealizable two-field arrangements, which consist geometrically of positive labels assigned to nonadjacent vertices, correspond algebraically to firing fields that are not separated by integer multiples of either of the grid periods (*Figure 3D,E*).

Moreover, note that the convex polytopes obtained for the grid-like code remain qualitatively unchanged in their geometry if the nonzero activations within each module are replaced by graded tuning curves as follows: convert all neural responses within a module into graded values by convolution along the spatial dimension by a kernel that has no periodicity over distances smaller than the module period (thus, the kernel cannot, for instance, be flat or contain multiple bumps within one module period). This convolution can be written as a matrix product with a circulant matrix of full rank and dimension equal to the full range $L$. Thus, the rank of the convolved matrix $\tilde{X}_g$ remains equal to the rank of $X_g$. Moreover, $\tilde{X}_g$ maintains the modular structure of $X_g$: it has the same within-module permutation invariance and across-module independence. Thus, the resulting geometry of the code – that it consists of convex polytopes constructed from orthogonal products of simplices – remains unchanged. As a result, all counting derivations, which are based on these geometric graphs, can be carried out for $\{0, 1\}$-valued codes without any loss of generalization relative to graded tuning curves. (However, the conversion to graded tuning will modify the distances between vertices and thus affect the quantitative noise robustness of different field arrangements, as we will investigate later.) Later, we will also show that the counting results generalize to higher dimensions and higher-resolution phase representations within each module.

Given this geometric characterization of the grid-like and modular-one-hot codes, we can now compute the number of realizable field arrangements it is possible to obtain with separating hyperplanes.

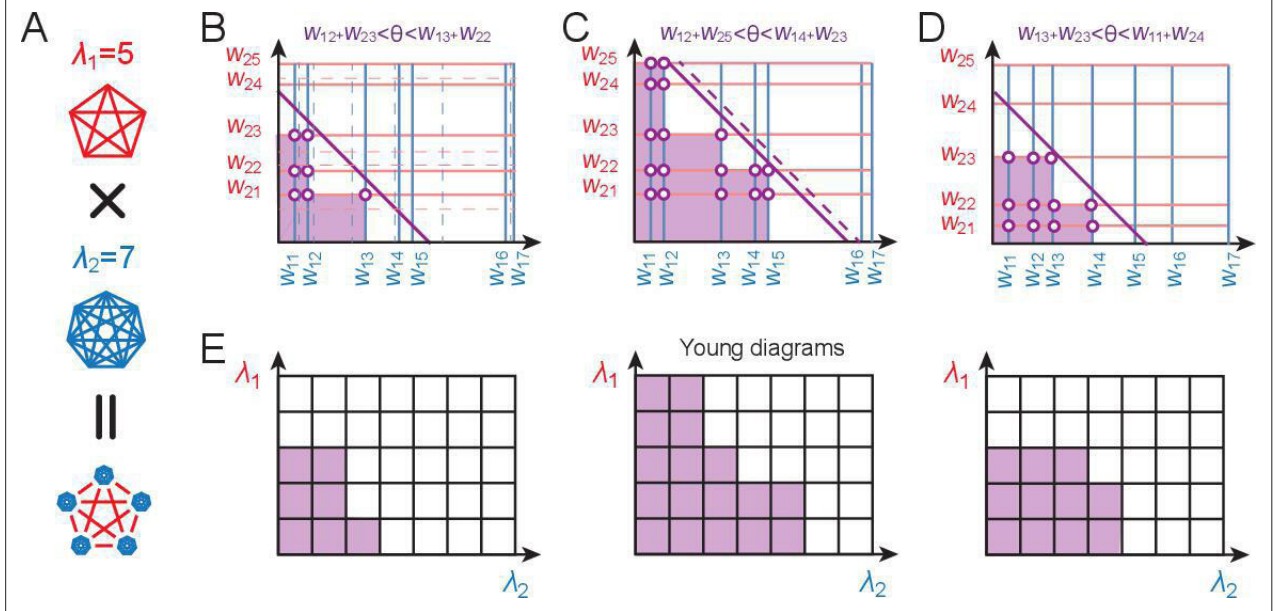

**Figure 5.** Counting realizable place field arrangements. (**A**) Geometric structure of a modular-one-hot code with two modules of periods $\lambda_1 = 5$ and $\lambda_2 = 7$. (**B–D**) Because cells within a module can be freely permuted, we can arrange the cells in order of increasing weights and keep this ordering fixed during counting, without loss of generality. We arrange the cells in modules 1 and 2 along the ordinate and abcissa in increasing weight order (solid blue and red lines, respectively). Because the weights can all be assumed to be non-negative for modular-one-hot codes, the threshold can be interpreted as setting a summed-weight budget: no cell (weight) combinations (purple regions with purple-white circles) below the threshold (diagonal purple line) can contribute to a place field arrangement, while all cell combinations with larger summed weights (unmarked regions) can. Increasing the threshold (from **B** to **C**) decreases the number of permitted combinations, as does decreasing the weights (**B** to **D**). Weight changes (**B**, from solid to dashed lines) and threshold changes (**C**, solid to dashed line), so long as they do not change which lines are to the bottom-left of the threshold, do not affect the number of permitted combinations, reflecting the topological structure of the counting problem. (**E**) With Young diagrams (each corresponding to **B–D** above), we extract the purely topological part of the problem, stripping away analog weights to simplify counting. A Young diagram consists of stacks of blocks in rows of nonincreasing width within a grid of a maximum width and height. The number of realizable field arrangements is simply the total number and multiplicity of distinct Young diagrams that can be built of the given height and width (see Appendix 3), which in our case is given by the periods of the two modules.

## Counting realizable place field arrangements

For modular-one-hot codes (but not for random codes), it is possible to specify any separating hyperplane using only non-negative weights and an appropriate threshold. This is an interesting property in the neurobiological context because it means that the finding that projections from entorhinal cortex to hippocampus are excitatory (*Steward and Scoville, 1976*; *Witter et al., 2000*; *Shepard, 1998*) does not further constrain realizable field arrangements.

It is also an interesting property mathematically, as we explore below: combined with the within-module permutation invariance property of modular-one-hot codes, the non-negative weight observation allows us to map the problem onto Young diagrams (*Figure 5*), which enables two things: (1) to move from considering separating hyperplanes geometrically, where infinitesimal variations represent distinct hyperplanes even if they do not change any pattern classifications, to considering them topologically, where hyperplane variations are considered as distinct only if they change the classification of any patterns, and (2) to use counting results previously established for Young diagrams.

Let us consider the field arrangements permitted by combining grid-like inputs from two modules, of periods $\lambda_1$ and $\lambda_2$, (*Figure 5A*). The total number of distinct grid-cell modules is estimated to be between 5 and 8 (*Stensola et al., 2012*). Further, there is a spatial topography in the projection of grid cells to the hippocampus, such that each local patch of the hippocampus likely receives inputs from 2, and likely no more than 3, grid modules (*Witter and Groenewegen, 1984*; *Amaral and Witter, 1989*; *Witter and Amaral, 1991*; *Honda et al., 2012*; *Witter et al., 2000*). We denote cells by their outgoing weights ($w_{ij}$ is the weight from cell $j$ in module  ) and arrange the weights along the axes of a coordinate space, one axis per module, in order of increasing size (*Figure 5B*). Since modular-one-hot codes are invariant to permutation of the cells within a module, we can assume a fixed ordering of

cells and weights in counting all realizable arrangements, without loss of generality. The threshold (dark purple line) sets which combination of summed weights can contribute to a place field arrangement: no cell combinations below the boundary (purple region) have too small a summed weight and cannot contribute, while all cell combinations with larger summed weights (white region) can (*Figure 5B*). Decreasing the threshold (from *Figure 5B* to C) or increasing weights (from *Figure 5B,C* to D) a sufficient amount so some cells cross the threshold increases the number of combinations. But changes that do not cause cells to move past the threshold do not change the combinations (*Figure 5B*, solid versus dashed gray lines).

Young diagrams extract this topological information, stripping away geometric information about analog weights (*Figure 5E*). A Young diagram consists of stacks of blocks in rows of nonincreasing width, with maximum width and height given in this case by the two module periods, respectively. The number of realizable field arrangements turns out to be equivalent to the total number of Young diagrams that can be built of the given maximum height and width (see Appendix 3). With this mapping, we can leverage combinatorial results on Young diagrams (*Fulton and Fulton, 1997*; *Postnikov, 2006*) (commonly used to count the number of ways an integer can be written as a sum of non-negative integers).

As a result, the total number of separating hyperplanes (K-field arrangements for all $K$) across the full range $L$ can be written exactly as (see Appendix 3).

$$\mathcal{N}_{\lambda_1, \lambda_2} = \sum_{k=0}^{\min(\lambda_1, \lambda_2)} (k!)^2 \, S_{k+1}^{(\lambda_1+1)} \, S_{k+1}^{(\lambda_2+1)} = B_{\lambda_2}^{(-\lambda_1)}, \tag{2}$$

where $S_k^{(n)}$ are Stirling numbers of the second kind and $B_k^{(n)}$ are the poly-Bernoulli numbers (*Postnikov, 2006*; *Kaneko, 1997*). Assuming that the two periods have a similar size ($\lambda_1 \approx \lambda_2 \equiv \lambda$), this number scales asymptotically as (*de Andrade et al., 2015*).

$$\mathcal{N}_{\lambda, \lambda} = B_\lambda^{(-\lambda)} = \left( \frac{1}{\log 2 \sqrt{1 - \log 2}} + o(1) \right) \frac{(2\lambda)!}{(2 \log 2)^{2\lambda}} \sim \lambda^{2\lambda}. \tag{3}$$

Thus, the number of realizable field arrangements with $\sim \lambda^2$ distinct modular-one-hot input patterns in a $2\lambda$-dimensional space grows nearly as fast as $\lambda^{2\lambda}$, (*Table 1*, row 2, columns 1–3). The total number of dichotomies over these input patterns scales as $2^{\lambda^2}$. Thus, while the number of realizable arrangements over the full range is very large, it is a vanishing fraction of all potential arrangements (*Table 1*, row 2, column 4).

If $M \geq 3$ modules were to contribute to each place field's response, then all realizable field arrangements still would correspond to Young diagrams; however, not all diagrams would correspond to realizable arrangements. Thus, counting Young diagrams would yield an upper bound on the number of realizable field arrangements but not an exact count (see Appendix 3). The latter limitation is not a surprise: Due to the structure of the grid-like code (a product of simplices), the enumeration of realizable dichotomies with arbitrarily many input modules is expected to be at least as challenging as that of Boolean functions. Counting the number of linearly separable Boolean functions of arbitrary (input) dimension (*Peled and Simeone, 1985*; *Hegedüs and Megiddo, 1996*) is hard.

Nevertheless, we can provide an exact count of the number of realizable $K$-dichotomies for arbitrarily many input modules $M$ if $K$ is small ($K = 1, 2, 3$ and 4). This may be biologically relevant since

**Table 1.** Number and fraction of realizable dichotomies with binary, modular-one-hot ($M = 2$ modules) and one-hot input codes with the same input cell budget ($N = 2\lambda$).

|  | # cells | # input patts (L) | # lin dichot | Frac lin dichot |
|---|---|---|---|---|
|  | $2\lambda$ | $2^{2\lambda}$ | $2^{2\lambda^2}$ | $2^{2\lambda^2 - 2^{2\lambda}}$ |
| Binary | = | << | << | >> |
|  | $2\lambda$ | $\lambda^2$ | $\left( \frac{\lambda}{e \log(2)} \right)^{2\lambda}$ | $2^{2\lambda \log(\lambda) - \lambda^2}$ |
| Modular-one-hot | = | << | << | >> |
| One-hot | $2\lambda$ | $2\lambda$ | $2^{2\lambda}$ | 1 |

place fields tend to fire sparsely even on long tracks and across environments. In this case, the number $\mathcal{N}_K$ of realizable small-$K$ field arrangements scales as (the exact expression is derived analytically in Appendix 3)

$$\mathcal{N}_K \sim M^{K-1}\lambda^{M+K-1}. \tag{4}$$

The scaling approximation becomes more accurate for periods that are large relative to the spatial discretization (see Appendix 3). Since the total number of K-dichotomies scales as $\lambda^{MK}$, the fraction of realizable K-dichotomies scales as $(M/\lambda)^{K-1}\lambda^{-(M-1)}$, which for $\lambda \gg 1, \lambda > M$ vanishes as a power law as soon as $M > 1$.

We can compare this result with the number of K-field arrangements realizable by one-hot codes. Since any arrangement is realizable with one-hot codes, it suffices to simply count all K-field arrangements. The full range of a one-hot code with $M\lambda$ cells is $M\lambda$, thus the number of realizable K-field arrangements is $\mathcal{N}_K = \binom{M\lambda}{K} \sim (M\lambda)^K$, where the last scaling holds for $K \ll M\lambda$. In short, a one-hot code enables $\sim M^K\lambda^K$ arrangements, while the corresponding modular-one-hot code with $M\lambda$ cells enables $\sim M^{K-1}\lambda^{K+M-1}$ field arrangements, for a ratio $\lambda^{M-1}/M \gg 1$ of realizable fields with modular-one-hot versus one-hot codes. Once again, as in the case where we counted arrangements without regard to sparseness, the grid-like code enables far more realizable K-field arrangements than one-hot codes.

In summary, place cells driven by grid inputs can achieve a very large number of unique coding states that grows exponentially with the number of modules. We have derived this result for $M = 2$ and all K-field arrangements, on one hand, and for arbitrary $M$ but ultra-sparse (small-$K$) field arrangements. It is difficult to obtain an exact result for sparse field arrangements for which $K$ is a small but finite fraction of $L$; however, we expect that regime should interpolate between these other two; it will be interesting and important for future work to shed light on this intermediate regime. In all cases, the number of realizable arrangements is large but a vanishingly small fraction of all arrangements, and thus forms a highly structured subset. This suggests that place cells, when driven by grid-cell inputs, can form a very large number of field arrangements that seem essentially unrestricted, but individual cells actually have little freedom in where to place their fields.

## Comparison with other input patterns

How does the number of realizable place field arrangements differ for input codes with different levels of modularity and hierarchy? We directly compare codes with the same neuron budget (input dimension $N$) by taking $N = M\lambda$, where for simplicity, we set $\lambda_i = \lambda$ for all modules in the modular-one-hot codes. This is because the modular-one-hot codes include all permutations of states in each module, the number of unique input states with equal-sized modules still equals the product of periods $L = (N/M)^M = \lambda^M$, as when the periods are different and coprime. The one-hot code generates far fewer distinct input patterns ($L = N = M\lambda$) than the modular-one-hot code, which in turn generates fewer input patterns than the binary code ($L = 2^N = 2^{M\lambda}$) (***Table 1***, column 2). This is due to the greater expressive power afforded by modularity and hierarchy.

Next, we compare results across codes for $M = 2$, the case for which we have an explicit formula counting the total number of realizable field arrangements for any $K$, and which is also best supported by the biology.

How many dichotomies are realizable with these inputs? As for the modular-one-hot codes, the patterns of $X_{\text{oh}}$ and $X_{\text{b}}$ fall on the vertices of a convex polytope. For $X_{\text{oh}}$, that polytope is just a $(N-1)$-dimensional simplex (***Figure 4C***, left), thus any subset of $K$ vertices ($1 \le K \le N$) lies on a $(K-1)$-dimensional face of the simplex and is therefore a linearly separable dichotomy. Thus, all $2^N$ dichotomies of $X_{\text{oh}}$ are realizable and the fraction of realizable dichotomies is 1 (***Table 1***, columns 3 and 4). For $X_{\text{b}}$, the polytope is a hypercube; it therefore consists of square faces, a prototypical configuration of points not in general position (not linearly separable, ***Figure 2B*** and ***Figure 4***, right) even when the number of patterns is small relative to the input dimension (number of cells). Counting the number of linearly separable dichotomies on vertices of a hypercube (also called linear Boolean functions) has attracted much interest (***Peled and Simeone, 1985***; ***Hegedüs and Megiddo, 1996***). It is an NP-hard combinatorial problem, so no exact solution exists. However, in the limit of large dimension ($N \to \infty$), the number of linearly separable dichotomies scales as $2^{N^2/2}$ (***Zuev, 1989***), a much larger

number than for one-hot inputs (*Table 1*, column 3). However, this number is a strongly vanishing fraction of all $2^{2^N}$ hypercube dichotomies (*Table 1*, column 4).

For modular-one-hot codes with $M$ modules, the polytopes contain $M$-dimensional hypercubes and not all patterns are thus in general position. We determined earlier that the total number of realizable dichotomies with $M = 2$ modules scales as $\lambda^{2\lambda}$, permitting a direct comparison with the one-hot and binary codes (*Table 1*, row 2).

Finally, we may compare grid-like codes with random (real-valued) codes, which are the standard inputs for the classical perceptron results. For a fixed input dimension, it is possible to generate infinitely many real-valued patterns, unlike the finite number achievable by $\{0,1\}$-valued codes. We thus construct a random codebook $X_r$ with the same number, $P = \lambda^2$, of input patterns as the modular-one-hot code. We then determine the input dimension $N$ required to obtain the same number of realizable field arrangements as the grid-like code. The number of realizable dichotomies of the random code with $P \gg N$ patterns scales as $P^N \sim \lambda^{2N}$ according to an asymptotic expansion of Cover's function counting theorem (*Cover, 1965*). For this number to match $\sim \lambda^{2\lambda}$, the number of realizable field arrangements with a one-hot-modular code (of two modules of size $\sim \lambda$ each requires) $N \sim \lambda$. This is a comparable number of input cells in both codes, which is an interesting result because unlike for random codes the grid-like input patterns are not in general position, the states are confined to be $\{0,1\}$-valued, and the grid input weights can be confined to be non-negative.

In sum, the more modular a code, the larger the set of realizable field arrangements, but these are also increasingly special subsets of all possible arrangements and are strongly structured by the inputs, with far from random or arbitrary configurations. Modular-one-hot codes are intermediate in modularity. Therefore, grid-driven place-cell responses occupy a middle ground between pattern richness and constrained structure.

## Place-cell-separating capacity

We now turn to question (2) from above: what is the maximal range of locations, $l^*$, over which all field arrangements are realizable? Once we reference a spatial range, the mapping of coding states to spatial locations matters (specifically, the fact that locations in the range are spatially contiguous matters, but given the fact that the code is translationally invariant [*Fiete et al., 2008*], the origin of this range does not). We thus call $l^*$ the 'contiguous-separating capacity' of a place cell (though we will refer to it as separating capacity, for short); it is the analogue of Cover's separating capacity (*Cover, 1965*), but for grid-like inputs with the addition of a spatial contiguity constraint.

We provide three primary results on this question. (1) We establish that for grid-structured inputs, the separating capacity $l^*$ equals the rank $R$ of the input matrix. (2) We establish analytically a formula for the rank $R$ of grid-like input matrices with integer periods and generalize the result to real-valued periods. (3) We show that this rank, and thus the separating capacity for generic real-valued module periods, asymptotically approaches the sum $\Sigma \equiv \sum_{m=1}^{M} \lambda_m$. Our results are verified by numerical simulation and counting (proofs provided in Supporting Information Appendix).

We begin with a numerical example, using periods $\{3,4\}$ (*Figure 6A*): the full range is $L = 12$, while we see numerically that the contiguous-separating capacity is $l^* = 6$. Although the separating capacity with grid-structured inputs is smaller than with random inputs, it is notably not much smaller (*Figure 6B*, black versus cyan curves), and it is actually larger than for random inputs if the read-out weights are constrained to be non-negative (*Figure 6B*, pink curves). Later, we will further show that the larger random-input capacity of place cells with unrestricted weights comes at the price of less robustness: the realizable fields have smaller margins. Next, we analytically characterize the separating capacity of place cells with grid-like inputs.

## Separating capacity equals rank of grid-like inputs

For inputs in general position, the separating capacity equals the rank of the input matrix (plus 1 when the threshold is allowed to be nonzero), and the rank equals the dimension (number of cells) of the input patterns – the input matrix is full rank. When inputs are in general position, all input subsets of size equaling the separating capacity have the same rank. But when input patterns are not in general position, some subsets can have smaller ranks than others even when they have the same size. Thus, when input patterns are not in general position the separating capacity is only upper bounded by the



**Figure 6.** Place-cell-separating capacity. (**A**) Fraction of K-field arrangements that are realizable with grid-like inputs as a function of range ($L$ indicates the full range; in this example, grid periods are $\{3, 4\}$ and $L = 12$). (**B**) Fraction of realizable field arrangements (summed over $K$) as a function of range for grid cells (black); for random inputs, range refers to number of input patterns (solid cyan: random with matching input dimension; open/dashed cyan: random with input dimension equal to rank of the grid-like input matrix; dark teal: same as open cyan, but with weights constrained to be non-negative, as for grid-like inputs). With the non-negative weight constraint for random inputs, different specific input configurations produce quite different results, introducing considerable variability in separating capacity (unlike the unconstrained random input case or the grid code case for which results are exact rather than statistical). (**C**) The grid code is generated by iterated application of a phase-shift operator as a function of one-step updates in position over a contiguous 1D range. This feature of the code leads to a separating capacity that achieves its optimal value, given by the rank of the input matrix. (**D**) Separating capacity as a function of the sum of module periods for real-valued periods (randomly drawn from $\lambda_i \in [3, 20]$ with $M \in \{2, 3, 4, 5, 6\}$, 100 realizations), showing the quality of the integer approximation at different resolutions. Integer approximations to the real-value periods at successively

*Figure 6 continued on next page*

*Figure 6 continued*

finer resolutions quickly converge, with results from $q = 2$ and $q = 4$ nearly indistinguishable from each other. Inset: ratio of separating capacity to sum of periods ($R_{re}^q/\Sigma$ as a function of resolution $q$ quickly approaches 1 from below as $q$ increases). (**E, F**) Capacity results generalize to multidimensional spatial settings: (**E**) in 2D, grid-cell-activity patterns lie on a hexagonal lattice (all circles of one color mark the activity locations of one grid cell). For grid periods $\{2, 3\}$, this code utilizes 4 two-periodic cells and 9 three-periodic cells, respectively. (**F**) Full range of the 2D grid-like code from (**E**). The set of contiguous locations over which any place field arrangement is realizable (the 2D separating capacity) is shown in gray.

rank of the full input matrix. In turn, the rank is only upper bounded by the number of cells (the input matrix need not be full rank).

For the grid-like code, all codewords can be generated by the iterated application of a linear operator $J$ to a single codeword: a simultaneous one-unit phase shift by a cyclic permutation in each grid module is such an operator $J$, which can be represented by a block-form permutation matrix. The sequence $x, Jx, J^2x, \ldots J^m x$ of patterns generated by applying $J$ to a grid-like codeword $x$ with the same module structure represents $m$ contiguous locations (**Figure 6C**).

The separating capacity for inputs generated by iterated application of the same linear operation saturates its bound by equaling the rank of the input pattern matrix. Since a code $x, Jx, J^2x, J^3x, \ldots$, generated by some linear operator $J$ with starting codeword $x$ is translation invariant, the number of dimensions spanned by these patterns strictly increases until some value $l$, after which the dimension remains constant. By definition, $l$ is therefore the rank $R$ of the input pattern matrix. It follows that any contiguous set of $l = R$ patterns is linearly independent, and thus in general position, which means that the separating capacity of such a pattern matrix is $R$.

For place cells, it follows that whenever $l \leq R$, with $R$ the rank of the grid-like input matrix, all field arrangements are realizable, while for any $l > R$, there will be nonrealizable field arrangements (Supporting Information Appendix). Therefore, the contiguous-separating capacity for place cells is $l^* = R$. This is an interesting finding: the separating capacity of a place cell fed with structured grid-like inputs approaches the same capacity as if fed with general-position inputs of the same rank. Next, we compute the rank $R$ for grid-like inputs under increasingly general assumptions.

## Grid input rank converges to sum of grid module periods
### Integer periods

For integer-valued periods $\lambda_m$, the rank of the matrix consisting of the multi-periodic grid-like inputs can be determined through the inclusion-exclusion principle (see Section B.4):

$$R_{\text{int}}(\lambda_1, \cdots, \lambda_M) = \sum_{i=1}^{M} \lambda_i + \sum_{k=2}^{M}(-1)^{k-1} \sum_{i=1}^{\binom{M}{k}} \text{GCD}(S_k^i), \tag{5}$$

where $S_k^i$ is the ith of the k-element subsets of $\{\lambda_1, \ldots, \lambda_M\}$. To gain some intuition for this expression, note that if the periods were pairwise coprime, all the GCDs would be 1 and this formula would quite simply produce $R_{\text{copr}}(\lambda_1, \ldots, \lambda_M) = \Sigma - M + 1$, where $\Sigma$ is defined as the sum of the module periods. If the periods are not pairwise coprime, the rank is reduced based on the set of common factors, as in (5), which satisfies the following inequality: $\Sigma - \sum_{i<j} \text{GCD}(\lambda_i, \lambda_j) \leq R_{\text{int}}(\lambda_1, \cdots, \lambda_M) \leq \Sigma$. When the periods are large ($\lambda \gg 1$), the rank approaches $\Sigma$. Large integers ($\lambda \gg 1$) evenly spaced or uniformly randomly distributed over some range tend not to have large common factors (**Cesaro, 1881**). As a result, even for non-coprime periods, the rank scales like and approaches $\Sigma$ (see below for more elaboration).

### Real-valued periods

Actual grid periods are real- rather than integer-valued, but with some finite resolution. To obtain an expression for this case, consider the sequence of ranks $R_{re}^q$ defined as

$$R_{re}^q(\lambda_1, \cdots, \lambda_M) = R_{int}\left(\lfloor q\lambda_1 \rfloor, \cdots, \lfloor q\lambda_M \rfloor\right), \tag{6}$$

where $\lfloor \cdot \rfloor$ denotes the floor operation, $q$ is an effective resolution parameter that takes integer values (the larger $q$, the finer the resolution of the approximation to a real-valued period), and the periods $0 < \lambda_1 < \ldots < \lambda_M$ are real numbers. The rank of the grid-like input matrix with real-valued

periods is given by $\lim_{q\to\infty} R_{\mathrm{re}}^q(\lambda_1,\cdots,\lambda_M)/q$, if this limit exists. A finer resolution (higher $q$) corresponds to representing phases with higher resolution within each module, and thus intuitively to scaling the number of grid cells in each module by $q$.

Suppose that the periods are drawn uniformly from an interval of the reals, which we take without loss of generality to be $(0,1)$. Then the values $\lfloor q\lambda_1 \rfloor, \cdots, \lfloor q\lambda_M \rfloor$ are integers in $\{1, \ldots, q\}$ and as above we have that $0 \leq q\Sigma - R_{\mathrm{re}}^q(\lambda_1, \cdots, \lambda_M) \leq \sum_{i<j} \mathrm{GCD}(\lfloor \lambda_i q \rfloor, \lfloor \lambda_j q \rfloor)$. In the infinite resolution limit ($q \to \infty$), the probability $\mathrm{GCD}(\lfloor \lambda_i q \rfloor, \lfloor \lambda_j q \rfloor) = g$ scales asymptotically as $1/g^2$, independent of $q$ (**Cesaro, 1881**), which means that large randomly chosen large integers tend not to have large common factors. This implies that with probability 1, the limit $\lim_{q\to\infty} R_{\mathrm{re}}^q(\lambda_1, \cdots, \lambda_M)/q$ is well-defined and equals $\Sigma$, the sum of the input grid module periods.

When assessed numerically at different resolutions ($q$), the approach of the finite-resolution rank to the real-valued grid period rank is quite rapid (**Figure 6D**). Thus, the separating capacity does not depend sensitively on the precision of the grid periods. It is also invariant to the resolution with which phases are represented within each module.

In summary, the place-cell-separating capacity with real-valued grid periods and high-resolution phase representations within each module equals the rank of the grid-like input matrix, which itself approaches $\Sigma$, the sum of the module periods. Thus, a place cell can realize any arrangement of fields over a spatial range given by the sum of module periods of its grid inputs.

It is interesting that the contiguous-separating capacity of a place cell fed with grid-like inputs not in general position approaches the same capacity as if fed with general-position inputs of the same rank. On the other hand, the contiguous-separating capacity is very small compared to the total range over which the input grid patterns are unique: since each local region of hippocampus receives input from 2 to 3 modules (**Witter and Groenewegen, 1984**; **Amaral and Witter, 1989**; **Witter and Amaral, 1991**; **Witter et al., 2000**; **Honda et al., 2012**), the range over which any field arrangement is realizable is at most 2–3 times the typical grid period. By contrast, the total range $L$ of locations over which the grid inputs provide unique codes scales as the product of the periods. The result implies that once field arrangements are freely chosen in a small region, they impose strong constraints on a much larger overall region and across environments. We explore this implication in more detail below.

## Generalization to higher dimensions

We have already argued that our counting arguments hold for realistic tuning curve shapes with graded activity profiles. This follows from the fact that convolution of the grid-like codes with appropriate smoothing kernels does not change the general geometric arrangement of codewords relative to each other as these convolution operations preserve within-module permutation symmetries and across-module independence in the code. We have also shown that the contiguous-separating capacity results apply to real-valued grid periods with dense phase encodings within each module.

Here, we describe the generalization to different spatial dimensions. Consider a $d$-dimensional grid-like code consisting of $(\lambda_m)^d$ cells in the mth module to produce a one-hot phase code for $\lambda_m$ (discrete) positions along each dimension (**Figure 6E**). Since the counting results rely only on the existence of a modular-one-hot code and not any mapping from real spaces to coding states, this code across multiple modules $m = 1, \ldots, M$ is equivalent to a modular-one-hot coding for $\prod_{m=1}^M (\lambda_m)^d$ states, with modules of size $(\lambda_m)^d$ each. All the counting results from before therefore hold, with the simple substitution $\lambda_m \to (\lambda_m)^d$ in the various formulae.

The contiguous-separating capacity in $d$-dimensions is defined as the maximum volume over which all field arrangements are realizable. Like the 1D separating capacity results, this volume depends upon the mapping of physical space to grid-like codes. We are able to show that for grid modules with periods $\lambda_1, \ldots, \lambda_M$ the generalized separating capacity is $l_d^\star = \Sigma_d = \sum_{m=1}^M \lambda_m^d$ (see Section B.4; **Figure 6F**). This result follows from essentially the same reasoning as for 1D environments, but with the use of $d$-dimensional phase-shift operators.

## Robustness of field arrangements to noise and nongrid inputs

An important quality of field arrangements that is neglected when merely counting the number of realizable arrangements or determining the separating capacity is robustness: these computations consider all realizable field arrangements, but field arrangements are practically useful only if they are robust so that small amounts of perturbation or noise in the inputs or weights do not render them

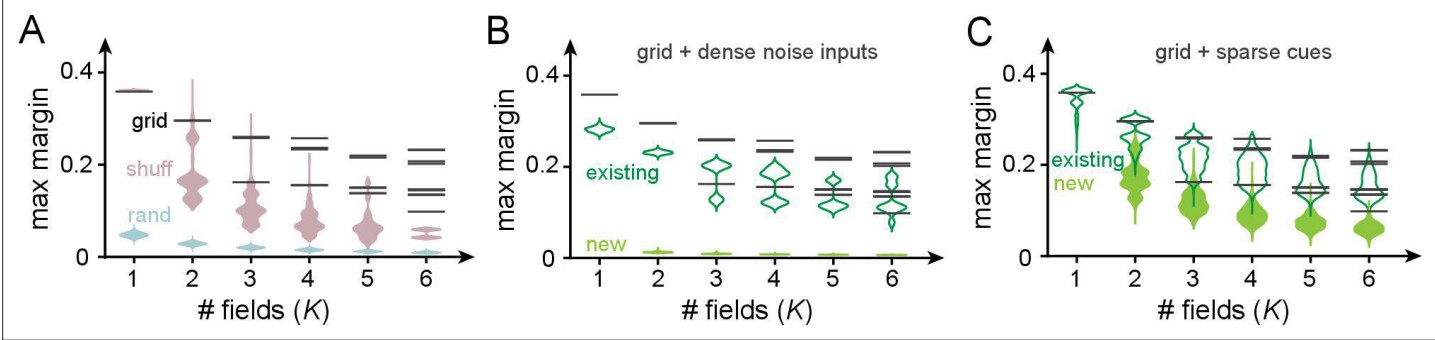

**Figure 7.** Robustness of place field arrangements to noise and nongrid inputs. In (**A–C**), grid periods are $\{31, 43\}$; the number of input patterns is set to $1333 = \text{LCM}(31, 43)$ for all input codes. Input patterns are normalized to have unity $L_1$ norm in all cases. Maximum margins are determined by using SVC in scikit-learn (*Pedregosa et al., 2011*) (with thresholds and no weight constraints). (**A**) Black bars: the maximum margins of all realizable arrangements with grid-like inputs (bars have high multiplicity: across the very large number of realizable field arrangements, the set of distinct maximum margins is small and discrete because of the regular geometric structure of the grid-like code). Pink: margins for shuffled grid inputs that break the code's modularity (shuffling neurons across modules for each pattern; 10 shuffles per $K$ and sampling 1000 realizable field arrangements per shuffle). Blue: margins for random inputs in general position (inputs sampled i.i.d. uniformly from $[0, 1]$; 10 realizations of a random matrix per $K$, 1000 realizable field arrangements sampled per realization). (**B**) Effect of noise on margins. We added dense noise inputs (100 non-negative i.i.d. random inputs at each location) to the place cell, in addition to the 74 grid-like inputs. (The expected value of each random input was 20% of the population mean of the grid inputs; thus, the summed random input was on average $(0.2 \times 100/74)$ the size of the summed grid input.) Black: noise-free margins as in (**A**). Empty green violins: margins of existing field arrangements modestly shrink in size. Solid green violins: margins of some newly created field arrangements: these are small and thus unstable. (**C**) Effect of sparse spatial inputs (plots as in **C**). (We added 100 sparse $\{0, 1\}$ inputs per location; each sparse input had $0.2 \times 2L/74$ fields placed randomly across the full range $L$, so that the summed sparse input was on average $(0.2 \times 100/74)$ the size of the summed grid input. The combined grid and nongrid input at each location was normalized to 1.)

unrealizable. Above, we showed that grid-like codes enable many dichotomies despite being structurally constrained, but that random analog-valued codes as well as more hierarchical codes permit even more dichotomies. Here, we show that the dichotomies realized by grid codes are substantially more robust to noise and thus more stable.

The robustness of a realizable dichotomy in a perceptron is given by its margin: for a given linear decision boundary, the margin is the smallest datapoint-boundary distance for each class, summed for the two classes. The maximum margin is the largest achievable margin for that dataset. The larger the maximum margin, the more robust the classification. We thus compare maximum margins (herein simply referred to as margins) across place field arrangements, when the inputs are grid-like or not.

Perceptron margins can be computed using quadratic programming on linear support vector machines (*Platt, 1998*). We numerically solve this problem for three types of input codes (permitting a nonzero threshold and imposing no weight constraints): the grid-like code $X_g$; the shuffled grid-like code $X_{gs}$ – a row- and column-shuffled version of the grid-like code that breaks its modular structure; and the random code $X_r$ of uniformly distributed random inputs (*Figure 7*). To make distance comparisons meaningful across codes, (1) all patterns (columns) involve the same number of neurons (dimension), (2) have the same total activity level (unity $L_1$ norm), and (3) the number of input patterns is the same across codes, and chosen to equal $L$, the full range of the corresponding grid-like code. To compute margins, we consider only the realizable dichotomies on these patterns.

The margins of *all* realizable place field arrangements with grid-like inputs are shown in *Figure 7A* (black); the margin values for all arrangements are discretized because of the geometric arrangements of the inputs, and each black bar has a very high multiplicity. The grid-like code produces much larger-margin field arrangements than shuffled versions of the same code and random codes (*Figure 7A*, pink and blue). The higher margins of the grid-like compared to the shuffled grid-like code show that it is the structured geometry and modular nature of the code that produce well-separated patterns in the input space (*Figure 4B*) and create wide margins and field stability. In other words, place field arrangements formed by grid inputs, though smaller in number than arrangements with differently coded inputs, should be more robust and stable against potential noise in neural activations or weights.

Next, we directly consider how different kinds of nongrid inputs, driving place cells in conjunction with grid-like inputs, affect our results on place field robustness. We examine two distinct types of

added nongrid input: (1) spatially dense noise that is meant to model sources of uncontrolled variation in inputs to the cell and (2) spatially sparse and reliable cues meant to model spatial information from external landmarks.

After the addition of dense noise, previously realizable grid-driven place field arrangements remain realizable and their margins, though somewhat lowered, remain relatively large (*Figure 7B*, empty green violins). In other words, grid-driven place field arrangements are robust to small, dense, and spatially unreliable inputs, as expected given their large margins. Note that because the addition of dense i.i.d. noise to grid-like input patterns pushes them toward general position, and general-position inputs enable more realizable arrangements, the noise-added versions of grid-like inputs also give rise to some newly realizable field arrangements (*Figure 7B*, full green violins). However, as with arrangements driven purely by random inputs, these new arrangements have small margins and are relatively not robust. Moreover, since by definition noise inputs are assumed to be spatially unreliable, the newly realizable arrangements will not persist across trials.

Next, the addition of sparse spatial inputs (similar to the one-hot codes of *Table 1*, though the sparse inputs here are nearly but not strictly orthogonal) leaves previous field arrangements largely unchanged and their margins substantially unmodified (*Figure 7C*, empty green violins). In addition, a few more field arrangements become realizable and these new arrangements also have large margins (*Figure 7C*, full green violins). Thus, sufficiently sparse spatial cues can drive additional stable place fields that augment the grid-driven scaffold without substantially modifying its structure. Plasticity in weights from these sparse cue inputs can drive the learning of new fields without destabilizing existing field arrangements.

In sum, grid-driven place arrangements are highly robust to noise. Combining grid-cell drive with cue-driven inputs can produce robust maps that combine internal scaffolds with external cues.

## High volatility of field arrangements with grid input plasticity

Our results on the fraction of realizable place field arrangements and on place-cell-separating capacity with grid-like inputs imply that place cells have highly restricted flexibility in laying down place fields (without direct drive from external spatially informative cues) over distances greater than $\Sigma$, the sum of the input grid module periods. Selecting an arrangement of fields over this range then constrains the choices that can be made over all remaining space in the same environment and across environments. Conversely, changing the field arrangement in any space by altering the grid-place weights should affect field arrangements everywhere.

We examine this question quantitatively by constructing realizable K-field arrangements (with grid-like responses generated as 1D slices through 2D grids [*Yoon et al., 2016*]), then attempting to insert one or a few new fields (*Figure 8A,B*). Inserting even a single field at a randomly chosen location through Hebbian plasticity in the grid-place weights tends to produce new additional fields at uncontrolled locations, and also leads to the disappearance of existing fields (*Figure 8A,B*).

Interestingly, though field insertion affects existing arrangements through the uncontrolled appearance or disappearance of other fields, it does not tend to produce local horizontal displacements of existing fields (*Figure 8C*): fields that persist retain their firing locations or they disappear entirely, consistent with the surprising finding of a similar effect in experiments (*Ziv et al., 2013*).

The locations of fields, including of uncontrolled field additions, are well-predicted by the structure (autocorrelation) of that cell's grid inputs (*Figure 8D*). This multi-peaked autocorrelation function, with large separations between the tallest peaks, reflects the multi-periodic nature of the grid code and explains why fields tend to appear or disappear at remote locations rather than shifting locally: modest weight changes in the grid-like inputs modestly alter the heights of the peaks, so that some of the well-separated tall peaks fall below threshold for activation while others rise above.

Quantitatively, insertion of a single field at an arbitrary location in a 20 m span grid-place weight plasticity results in the insertion or deletion, on average, of $\sim 0.2$ uncontrolled fields per meter. The insertion of four fields anywhere over 20 m results in an average of one uncontrolled field per meter (*Figure 8E*).

Thus, if a place cell were to add a field in a new environment or within a large single environment by *modifying* the grid-place weights, our results imply that it is extremely likely that this learning will alter the original grid-cell-driven field arrangements (scaffold). By contrast, adding fields that are driven by spatially specific external cues, though plasticity in the cue input-to-place cell synapses, may not affect

**Figure 8.** Predicted volatility of place field arrangements. (**A**) Top: original field arrangement over a 20 m space (gray line: summed inputs to place cell; purple stars: original field locations; green arrow: location where new field will be induced by Hebbian plasticity in grid-place weights). Bottom: after induction of the new field (green star), two new uncontrolled fields appear (red stars). (**B**) Similar to (**A**): the insertion of a new field at a random location (green star) leads to one uncontrolled new field (red star) and the loss of two original fields (empty red stars). (**C**) Histogram of changes, after single-field insertion, in pairwise inter-field intervals (spacings): the primary off-target effect of field insertion is for other fields to appear or disappear, but existing fields do not tend to move. (**D**) A spatially extended version of (**C**) (purple), together with the (vertically rescaled) autocorrelation of the grid inputs to the cell (gray): new fields tend to appear at spacings corresponding to peaks in the input autocorrelation function. (**E**) Sum of uncontrolled field insertions or deletions per meter, in response to inserted fields when starting with a K-field arrangement over 20 m. (**F**) High place field volatility resulting from plasticity in the grid-to-place synapses suggests the possibility that grid-place weights might be relatively rigid (nonplastic).

field arrangements elsewhere if the cues are sufficiently sparse (unique); in this case, the added field would be a 'sensory' field rather than an internally generated or 'mnemonic' one.

In sum, the small separating capacity of place cells according to our model may provide one explanation for the high volatility of the place code across tens of days (*Ziv et al., 2013*) if grid-place weights are subject to any plasticity over this timescale. Alternatively, to account for the stability of spatial representations over shorter timescales, our results suggest that external cue-driven inputs to place cells can be plastic but the grid-place weights, and correspondingly, the internal scaffold, may be fixed rather than plastic (*Figure 8F*). In experiments that induce the formation of a new place field through intracellular current injection (*Bittner et al., 2015*), it is notable that the precise location of

the new field was not under experimental control: potentially, an induced field might only be able to form where an underlying (near-threshold) grid scaffold peak already exists to help support it, and the observed long plasticity window could enable place cells to associate a plasticity-inducing cue with a nearby scaffold peak.

This alternative is consistent with the finding that entorhinal-hippocamapal connections stabilize long-term spatial and temporal memory (**Brun et al., 2008**; **Brun et al., 2002**; **Suh et al., 2011**).

Finally, we note that the robustness of place field arrangements obtained with grid-like inputs is not inconsistent with the volatility of field arrangements to the addition or deletion of new fields through grid-place weight plasticity. Grid-driven place field arrangements are robust to random i.i.d. noise in the inputs and weights, as well as the addition of nongrid sparse inputs. On the other hand, the volatility results involve associative plasticity that induces highly nonrandom weight changes that are large enough to drive constructive interference in the inputs to add a new field at a specific location. This nonrandom perturbation, applied to the distributed and globally active grid inputs, results in global output changes.

## Discussion

### Grid-driven hippocampal scaffolds provide a large representational space for spatial mapping

We showed that when driven by grid-like inputs, place cells can generate a spatial response scaffold that is influenced by the structural constraints of the grid-like inputs. Because of the richness of their grid-like inputs, individual place cells can generate a large library of spatial responses; however, these responses are also strongly structured so that the realizable spatial responses are a vanishingly small fraction of all spatial responses over the range where the grid inputs are unique. However, realizable spatial field arrangements are robust, and place cells can then 'hang' external sensory cues onto the spatial scaffold by associative learning to form distinct maps spatial maps for multiple environments. Note that our results apply equally well to the situation where grid states are incremented based on motion through arbitrary Euclidean spaces, not just spatial ones (**Killian et al., 2012**; **Constantinescu et al., 2016**; **Aronov et al., 2017**; **Klukas et al., 2020**).

### Summary of mathematical results

Mathematically, formulating the problem of place field arrangements as a perceptron problem led us to examine the realizable (linearly separable) dichotomies of patterns that lie not in general position but on the vertices of convex regular polytopes, thus extending Cover's results to define capacity for a case with geometrically structured inputs (**Cover, 1965**). Input configurations not in general position complicate the counting of linearly separable dichotomies. For instance, counting the number of linearly separable Boolean functions, which is precisely the problem of counting the linearly separable dichotomies on the hypercube, is NP-hard (**Peled and Simeone, 1985**; **Hegedüs and Megiddo, 1996**).

We showed that the geometry of grid-cell inputs is a convex polytope, given by the orthogonal product of simplices whose dimensions are set by the period of each grid module divided by the resolution. Grid-like codes are a special case of modular-one-hot codes, consisting of a population divided into modules with only one active cell (group) at a time per module.

Exploiting the symmetries of modular-one-hot codes allowed us to characterize and enumerate the realizable K-field arrangements for small fixed $K$. Our analyses relied on combinatorial objects called Young diagrams (**Fulton and Fulton, 1997**). For the special case of $M = 2$ modules, we expressed the number of realizable field arrangements exactly as a poly-Bernoulli number (**Kaneko, 1997**). Note that with random inputs, by contrast, it is not well-posed to count the number of realizable K-field arrangements when $K$ is fixed since the solution will depend on the specific configuration of input patterns. While we have considered two extreme cases analytically, one with no constraints on place field sparsity and the other with very few fields, it remains an outstanding question of interest to examine the case of sparse but not ultra-sparse field arrangements in which the number of fields is proportional to the full range, with a constant small prefactor (**Itskov and Abbott, 2008**). Finding results in this regime would involve restricting our count of all possible Young diagrams to a subset with a fixed filled-in area (purple area in **Figure 5**). This constraint makes the counting problem significantly harder.

We showed using analytical arguments that our results generalize to analog or graded tuning curves, real-valued periods, and dense phase representations per module. We also showed numerically that our qualitative results hold when considering deviations from the ideal, like the addition of noise in inputs and weights. The relatively large margins of the place field arrangements obtained with grid-like inputs make the code resistant to noise. In future work, it will be interesting to further explore the dependence of margins, and thus the robustness of the place field arrangements, on graded tuning curve shapes and the phase resolution per module.

## Robustness, plasticity, and volatility

As described in the section on separating capacity, once grid-place weights are set over a relatively small space (about the size of the sum of the grid module periods), they set up a scaffold also outside of that space (within and across environments). Associating an external cue with this scaffold would involve updating the weights from the external sensory inputs to place cells that are close to or above threshold based on the existing scaffold. This does not require relearning grid-place weights and does not cause interference with previously learned maps.

By contrast, relearning the grid-place weights for insertion of another grid-driven field rearranges the overall scaffold, degrading previously learned maps (volatility: *Ziv et al., 2013*). If we consider a realizable field arrangement in a small local region of space then impose some desired field arrangement in a different local region of space through Hebbian learning, we might ask what the effect would be in the first region. Our results on field volatility provide an answer: if the first local region is of a size comparable to the sum of the place cell's input grid periods, then any attempt to choose field locations in a different local region of space (e.g., a different environment) will almost surely have a global effect that will likely affect the arrangement of fields in the first region. A similar result might hold true if the first region is actually a disjoint set of local regions whose individual side lengths add up to the sum of input grid periods. This prediction might be consistent with the observed volatility of place fields over time even in familiar environments (*Ziv et al., 2013*).

Our volatility results alternatively raise the intriguing possibility that grid-place weights, and thus the scaffold, might be largely fixed and not especially plastic, with plasticity confined to the nongrid sensory cue-driven inputs and in the return projections from place to grid cells. The experiments of *Rich et al., 2014* – in which place cells are recorded on a long track, the animal is then exposed to an extended version of the track, but the original fields do not shift – might be consistent with this alternative possibility. These are two rather strong and competing predictions that emerge from our model, each consistent with different pieces of data. It will be very interesting to characterize the nature of plasticity in the grid-to-place weights in the future.

## Alternative models of spatial tuning in hippocampus

This work models place cells as feedforward-driven conjunctions between (sparse) external sensory cues and (dense) motion-based internal position estimates computed in grid cells and represented by multi-periodic spatial tuning curves. In considering place-cell responses as thresholded versions of their feedforward inputs including from grid cells, our model follows others in the literature that make similar assumptions (*Hartley et al., 2000*; *Solstad et al., 2006*; *Sreenivasan and Fiete, 2011*; *Monaco et al., 2011*; *Cheng and Frank, 2011*; *Whittington et al., 2020*). These models do not preclude the possibility that place cells feed back to correct grid-cell states, and some indeed incorporate such return projections (*Sreenivasan and Fiete, 2011*; *Whittington et al., 2020*; *Agmon and Burak, 2020*). It will be interesting in future work to analyze how such return projections affect the capacity of the combined system.

Our assumptions and model architecture are quite different from those of a complementary set of models, which take the view that grid-cell activity is derived from place cells (*Kropff and Treves, 2008*; *Dordek et al., 2016*; *Stachenfeld et al., 2017*). Our assumptions also contrast with a third set of models in which place-cell responses are assumed to emerge largely from locally recurrent weights within hippocampus (*Tsodyks et al., 1996*; *Samsonovich and McNaughton, 1997*; *Battista and Monasson, 2020*; *Battaglia and Treves, 1998*). One challenge for those models is in explaining how to generate stable place fields through velocity integration across multiple large environments: the capacity (number of fixed points) of many fully connected neural integrator models in the style of Hopfield networks tends to be small – scaling as $\sim N$ states with $N$ neurons (*Amit et al., 1985*;

*Gardner, 1988*; *Abu-Mostafa and Jacques, 1985*; *Sompolinsky and Kanter, 1986*; *Samsonovich and McNaughton, 1997*; *Battaglia and Treves, 1998*; *Battista and Monasson, 2020*; *Monasson and Rosay, 2013*) because of the absence of modular structures (*Fiete et al., 2014*; *Sreenivasan and Fiete, 2011*; *Chaudhuri and Fiete, 2019*; *Mosheiff and Burak, 2019*). There are at least two reasons why a capacity roughly equal to the number of place cells might be too small, even though the number of hippocampal cells is large: (1) a capacity equal to the number of place cells would be quickly saturated if used to tile 2D spaces: $10^6$ states from $10^6$ cells supply $10^3$ states per dimension. Assuming conservatively a spatial resolution of 10 cm per state, this means no more than 100 m of coding capacity per linear dimension, with no excess coding states for error correction (*Fiete et al., 2008*; *Sreenivasan and Fiete, 2011*). (2) The hippocampus sits atop all sensory processing cortical hierarchies and is believed to play a key role in episodic memory in addition to spatial representation and memory. The number of potential cortical coding states is vastly larger than the number of place cells, suggesting that the number of hippocampal coding states should grow more rapidly than linearly in the number of neurons, which is possible with our grid-driven model but not with nonmodular Hopfield-like network models with pairwise weights between neurons.

Even if our assumption that place cells primarily derive their responses from grid-like inputs combined with external cue-derived nongrid inputs is correct, place cells may nevertheless deviate from our simple perceptron model if the place response involves additional layers of nonlinear processing. There are many ways in which this can happen: place cells are likely not entirely independent of each other, interacting through population-level competition and other recurrent interactions. Dendritic nonlinearities in place cells act as a hidden layer between grid-cell input and place cell firing (*Poirazi and Mel, 2001*; *Polsky et al., 2004*; *Larkum et al., 2007*; *Spruston, 2008*; *Larkum et al., 2009*; *Harnett et al., 2012*; *Harnett et al., 2013*; *Stuart et al., 2016*). Or, if we identify our model place cells as residing in CA1, then CA3 would serve as an intermediate and locally recurrent processing layer. In principle, hidden layers that generated a one-hot encoding for space from the grid-like inputs and then drove place cells as perceptrons would make all place field arrangements realizable. However, such an encoding would require a very large number of hidden units (equal to the full range of the grid code, while the grid code itself requires only the logarithm of this number). Additionally, place cells may exhibit richer input-output transformations than a simple pointwise nonlinearity, for instance, through cellular temporal dynamics including adaptation or persistent firing. Finding ways to include these effects in the analysis of place field arrangements is a promising and important direction for future study.

In sum, combining modular grid-like inputs produces a rich spatial scaffold of place fields, on which to associate external cues, much larger than possible with nonmodular recurrent dynamics within hippocampus. Nevertheless, the allowed states are strongly constrained by the geometry of the grid-cell drive. Further, our results suggest either high volatility in the place scaffold if grid-to-place-cell weights exhibit synaptic plasticity, or suggest the possibility that grid-to-place-cell weights might be random and fixed.

## Numerical methods

### Random, weight-constrained random, and shuffled inputs

Entries of the random input matrix are uniformly distributed variables in $[0, 1]$. To compare separating capacity (*Figure 4*) of random codes with the grid-like code, we consider matrices of the same input dimension (number of neurons) as the grid-cell matrix, or alternatively of the same rank as the grid-cell matrix, then use Cover's theorem to count the realizable dichotomies (*Cover, 1965*). Weight-constrained random inputs (*Figure 4B–D*) are random inputs with non-negative weights imposed during training.

To compare margins (*Figure 7*), we use matrices with the same input dimension and number of patterns. As margins scale linearly with the norm of the patterns, to keep comparisons fair the input columns (patterns) are normalized to have unity $L_1$ norm.

### Nongrid inputs

To test how nongrid inputs affect our results (*Figure 7C,D*), the $\lambda_1 + \lambda_2$ grid-like inputs from two modules with periods $\lambda_1 = 31$ and $\lambda_2 = 43$ are augmented by 100 additional inputs. In *Figure 7C*, each nongrid dense noisy input is a random variable selected uniformly and identically at each location from

the uniform interval $[0, 2\mu]$, where $\mu = 0.2\mu_g$, and $\mu_g = 2/(\lambda_1 + \lambda_2)$ is the population mean of the grid inputs. In **Figure 7D**, each nongrid sparse input is a $\{0, 1\}$ random variable with $Q$ nonzero responses across the full range $L = \lambda_1 \lambda_2$. We set $Q = 0.2L\mu_g$. In all cases, input columns (patterns with grid and nongrid inputs combined) are finally normalized to have unity $L_1$ norm. Results are based on 1000 realizations (samples) of the nongrid inputs.

## Grid-like inputs with graded tuning curves

We generate periodic grid-like activity with graded tuning curves as a function of 1D space $x$ in cell of module $m$ with period $\lambda_m$ as follows **Sreenivasan and Fiete, 2011**:

$$g(\phi_m(x), \varphi_i) = e^{-\frac{\|\phi_m - \varphi_i\|^2}{2\sigma_g^2}} \quad , \quad \|\alpha\| = \min(|\alpha|, 1 - |\alpha|) \tag{7}$$

where the phase of module $m$ is $\phi_m(x) = (x/\lambda_m \bmod 1)$. The ith cell in a module has a preferred activity phase $\varphi_i$ drawn randomly and uniformly from $(0,1)$. The tuning width $\sigma_g$ is defined in terms of phase, thus in real space the width of the activity bump grows linearly with the module period. We set $\sigma_g = 0.16$ (thus the full-width at half-max of the phase tuning curve equals 3/8 of the period, similar to grid cells).

Finally, to simulate quasi-periodic grid responses in 1D, we first generate 2D responses with Gaussian tuning on a hexagonal lattice, with the same field width as above. 1D responses of grid cells from the same module are then generated as parallel 1D slices of this lattice as in **Yoon et al., 2016**, with phases uniformly drawn at random.

# Acknowledgements

This work was supported by the Simons Foundation through the Simons Collaboration on the Global Brain, the ONR, the Howard Hughes Medical Institute through the Faculty Scholars Program to IRF, and the Alfred P Sloan Research Fellowship FG-2017-9554 to TT. We thank Sugandha Sharma, Leenoy Meshulam, and Luyan Yu for comments on the manuscript.

# Additional information

## Funding

| Funder | Grant reference number | Author |
| --- | --- | --- |
| Simons Foundation | Simons Collaboration on the Global Brain | Man Yi Yim<br>Ila R Fiete |
| Howard Hughes Medical Institute | Faculty Scholars Program | Ila R Fiete |
| Alfred P. Sloan Foundation | Alfred P. Sloan Research Fellowship FG-2017-9554 | Thibaud Taillefumier |
| Office of Naval Research | S&T BAA Award N00014-19-1-2584 | Ila R Fiete |

The funders had no role in study design, data collection and interpretation, or the decision to submit the work for publication.

## Author contributions

Man Yi Yim, conceptualization, data-curation, Formal analysis, Investigation, methodology, visualization, writing-original-draft; Lorenzo A Sadun, Formal analysis, Investigation; Ila R Fiete, Thibaud Taillefumier, conceptualization, Formal analysis, funding-acquisition, Investigation, methodology, project-administration, resources, supervision, validation, visualization, writing-original-draft, writing-review-and-editing

## Author ORCIDs

Lorenzo A Sadun (iD) http://orcid.org/0000-0002-2518-573X
Ila R Fiete (iD) http://orcid.org/0000-0003-4738-2539

Thibaud Taillefumier  http://orcid.org/0000-0003-3538-6882

**Decision letter and Author response**
Decision letter https://doi.org/10.7554/eLife.62702.sa1
Author response https://doi.org/10.7554/eLife.62702.sa2

## Additional files

### Supplementary files
• Transparent reporting form

### Data availability
The authors confirm that the data supporting the findings of this study are available within the article. Implementation details and code are available at: https://github.com/myyim/placecellperceptron copy archived at https://archive.softwareheritage.org/swh:1:rev:8e03b880f47a1f0b7934afd91afb 167f669ceeab.

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

## Appendix 1

### The geometry of the grid code

In this Appendix, we introduce the geometrical framework for the study of place cells modeled as perceptrons reading out the activity of grid cells. First, we define the space of grid-like inputs via symmetry considerations and without considering explicitly their relation to spatial locations. Second, we discuss linearly separable dichotomies in the space of grid-like inputs, whose geometric arrangements are not in general position. Third, we show that the geometry of grid-like inputs is that of a polytope that can be decomposed as an orthogonal product of simplices.

### The space of grid-like inputs

We model grid-cell activity via $\{0, 1\}$ spatial patterns $\boldsymbol{r}$ that take value 1 whenever the cell is active and take value 0 otherwise (*Fyhn et al., 2004*; *Fiete et al., 2008*). To model the periodic spatial response of grid cells, we assume that the activity pattern of a grid cell defines a periodic lattice with integer period $\lambda$. For simplicity, we consider 1D model for which the spatial patterns $\boldsymbol{r}$ are $\lambda$-periodic vectors and for which the set of activity patterns is given by the lattices $i + \lambda\mathbb{Z}$, $1 \leq i \leq L$. We refer to the index   as the phase index of the grid-cell spatial pattern. Our key results will generalize to lattices of arbitrary dimension $n$, for which the set of spatial patterns is given by the hypercube lattices $\boldsymbol{i} + \left(\lambda\mathbb{Z}\right)^n$, with phase indices $\boldsymbol{i}$ in $\{1, \ldots, \lambda\}^n$.

Within a population, grid cells can have distinct periods and arbitrary phases. To model this heterogeneity, we consider a population of grid cells with $M$ possible integer spatial periods $\boldsymbol{\lambda} = (\lambda_1, \ldots, \lambda_M)$, thereby defining $M$ modules of grid cells. We assume that each module comprises all possible grid-cell-activity patterns, that is, $\lambda_m$ grid cells labeled by the phase indices  , $1 \leq i \leq \lambda_m$. For convenience, we index each cell by its module index $m$ and its phase index  , $1 \leq i \leq \lambda_m$, so that the actual component index of cell $(m, i)$, $1 \leq i \leq \lambda_m$, is $\sum_{n<m} \lambda_m + i$. By construction of our model, at every spatial position, each module has a single active cell. Thus, at each spatial position, the grid-like input is specified by $\{0, 1\}$ column vectors $\boldsymbol{c}_\lambda$ of dimension $N = \sum_{m=1}^{M} \lambda_m$, the total number of grid cells.

In principle, the inputs to place cells are defined as spatial locations. Here, by contrast, we consider grid-like inputs as the inputs to place cells, without requiring these patterns to be spatial encodings. This approach is mathematically convenient as it allows us to exploit the many symmetries of the set of grid-like inputs denoted by $\mathcal{C}_\lambda$. The set $\mathcal{C}_\lambda$ contains as many grid-like inputs $\boldsymbol{c}$ as there are choices of phase indices in each module, that is, $\Lambda = \prod_{m=1}^{M} \lambda_m$:

$$\mathcal{C}_\lambda = \left\{ \boldsymbol{c} = (\boldsymbol{c}_1, \ldots, \boldsymbol{c}_M) \in \{0,1\}^{\lambda_1} \times \ldots \times \{0,1\}^{\lambda_M} \;\middle|\; \sum_{i=1}^{\lambda_m} c_{m,i} = 1, \quad 1 \leq m \leq M \right\}. \tag{8}$$

Here follow two examples of grid-like inputs $\mathcal{C}_\lambda$ enumerated in lexicographical order for $\boldsymbol{\lambda} = (2, 3)$ and $\boldsymbol{\lambda} = (2, 2, 2)$.

$$\mathcal{C}_{(2,3)} = \left\{ \begin{array}{cccccc} 1 & 1 & 1 & 0 & 0 & 0 \\ 0 & 0 & 0 & 1 & 1 & 1 \\ \hline 1 & 0 & 0 & 1 & 0 & 0 \\ 0 & 1 & 0 & 0 & 1 & 0 \\ 0 & 0 & 1 & 0 & 0 & 1 \end{array} \right\}, \qquad \mathcal{C}_{(2,2,2)} = \left\{ \begin{array}{cccccccc} 1 & 1 & 1 & 1 & 0 & 0 & 0 & 0 \\ 0 & 0 & 0 & 0 & 1 & 1 & 1 & 1 \\ \hline 1 & 1 & 0 & 0 & 1 & 1 & 0 & 0 \\ 0 & 0 & 1 & 1 & 0 & 0 & 1 & 1 \\ \hline 1 & 0 & 1 & 0 & 1 & 0 & 1 & 0 \\ 0 & 1 & 0 & 1 & 0 & 1 & 0 & 1 \end{array} \right\}. \tag{9}$$

Observe that, albeit inspired by the spatial activity of grid cells, the set of patterns $\mathcal{C}_\lambda$ has broader relevance than suggested by its use for modeling grid-like inputs. In fact, the set of patterns $\mathcal{C}_\lambda$ describes any modular winner-take-all activity, whereby cells are pooled in modules with only one cell active at a time – the winner of the module.

In the following, we consider that linear read-outs of grid-like inputs determine the activity of downstream cells, called place cells (*O'Keefe and Dostrovsky, 1971*). The set of these linear read-outs is the vector space $V_\lambda$ spanned by the grid-like inputs $\mathcal{C}_\lambda$. The dimension of the vector space $V_\lambda$

specifies the dimensionality of the grid code. The following proposition characterizes $V_\lambda$ and shows that its dimension is simply related to the periods $\lambda$.

## Proposition 1

The set of grid-like inputs $C_\lambda$ specified by $M$ grid modules with integer periods $\lambda = (\lambda_1, \ldots, \lambda_M)$ span the vector space

$$V_\lambda = \text{span}\, C_\lambda = \left\{ y = (y_1, \ldots, y_M) \in \mathbb{R}^{\lambda_1} \times \ldots \times \mathbb{R}^{\lambda_M} \ \bigg| \ \sum_{i=1}^{\lambda_1} y_{1,i} = \ldots = \sum_{i=1}^{\lambda_M} y_{M,i} \right\}, \tag{10}$$

In particular, the embedding dimension of the grid code is $\dim V_\lambda = \sum_{m=1}^{M} \lambda_m - M + 1$.

Proof. Let us denote by $A_\lambda$ a matrix formed by collecting all the column vectors from $C_\lambda$. The vector space $V_\lambda$ is the range of the matrix $A_\lambda$, which is also the orthogonal complement of $\ker A_\lambda^T$. A vector $x = (x_{1,1}, \ldots, x_{1,\lambda_1} | \ldots \ \ldots | x_{M,1}, \ldots, x_{M,\lambda_M})$ in $\mathbb{R}^{\lambda_1} \times \ldots \times \mathbb{R}^{\lambda_M}$ belongs to $\ker A_\lambda^T$ if and only if $x^T A_\lambda = 0$. By construction of the matrix $A_\lambda$:

$$x^T A_\lambda = 0 \quad \Longleftrightarrow \quad \sum_{m=1}^{M} x_{m,i_m} = 0, \quad \text{for all} \quad 1 \le i_m \le \lambda_m, \tag{11}$$

where $i_m$ refers to the index of the active cell in module $m$. The latter characterization implies that

$$\ker A_\lambda^T = \left\{ x = (a_1, \ldots, a_1 | \ldots \ \ldots | a_M, \ldots, a_M) \in \mathbb{R}^{\lambda_1} \times \ldots \times \mathbb{R}^{\lambda_M} \ \bigg| \ \sum_{m=1}^{M} a_m = 0 \right\}. \tag{12}$$

In turn, a vector $y = (y_{1,1}, \ldots, y_{1,\lambda_1} | \ldots \ \ldots | y_{M,1}, \ldots, y_{M,\lambda_M})$ of the orthogonal complement of $\ker A_\lambda^T$, that is, in the range of $A_\lambda$, is determined by $x^T y = 0$ for all $x$ in $\ker A_\lambda^T$. From the above characterization of $\ker A_\lambda^T$, this means that $y$ is in the range of $A_\lambda$, that is, in $V_\lambda$, if and only if for all $a_1, \ldots, a_M$ such that $\sum_{m=1}^{M} a_m = 0$, we have

$$\sum_{m=1}^{M} a_m \sum_{i=0}^{\lambda_m - 1} y_{m,i} = 0. \tag{13}$$

Substituting $a_M = -\sum_{m=0}^{M-1} a_m$ in the above relation, we have that for all $a_1, \ldots, a_{M-1}$ in $\mathbb{R}^{M-1}$,

$$\sum_{m=1}^{M-1} a_m \left( \sum_{i=0}^{\lambda_m - 1} y_{m,i} - \sum_{i=0}^{\lambda_M} y_{M,i} \right) = 0, \tag{14}$$

which is equivalent to $\sum_{i=0}^{\lambda_m - 1} y_{m,i} = \sum_{i=0}^{\lambda_M - 1} y_{M,i}$ for all $m$, $1 \le m < M$. The above relation entirely specifies the range of the activity matrix $A_\lambda$, that is, $V_\lambda$, as a vector space of dimension $\sum_{m=1}^{M} \lambda_m - M + 1$.

### Linear read-outs of grid-like inputs

We model the response of a place cell as that of a perceptron, which takes grid-like inputs $c$ in $\mathcal{C}_\lambda$ as inputs (**Rosenblatt, 1958**). Such a perceptron is parametrized by a decision threshold $\theta$ and by a vector of read-out weights $w = (w_{1,1}, \ldots, w_{1,\lambda_1} | \ldots \ldots | w_{M,1}, \ldots, w_{M,\lambda_M})$, where the vertical separators delineate the grid-cell modules with periods $\lambda_m$, $1 \le m \le M$. By convention, we consider that a place cell is active for grid-like inputs $c$ such that $w^T c > \theta$ and inactive otherwise. Thus, in the perceptron framework, a place cell has a multi-field structure if it is active on a set of several grid-like inputs $\mathcal{S} \subset \mathcal{C}_\lambda$, with $|\mathcal{S}| > 1$ (**Rich et al., 2014**). Considering grid-like inputs as inputs allows one to restrict the class of perceptrons under consideration.

## Proposition 2

Every realizable multi-field structure can be implemented by a perceptron with (*i*) non-negative weights, or (*ii*) with zero threshold.

Proof. (*i*) If $M$ is the total number of modules and 1 is the $N$-dimensional column vectors of 1, for all grid-like inputs $c$ in $\mathcal{C}_\lambda$ we have $\mathbf{1}^T c = (1, \ldots, 1)c = M$. Thus, for all perceptron $(w, \theta)$ and for all real μ, we have

$$(w + \mu\mathbf{1})^T c = w^T c + \mu\mathbf{1}^T c = p + \mu M, \tag{15}$$

where $p$ is the place-cell-activity level for grid-cell pattern $c$ in $\mathcal{C}_\lambda$. Consequently, setting $\mu \geq \max_{1 \leq i \leq N} |w_i|$, $w' = w + \mu\mathbf{1}$ and $\theta' = \theta + \mu M$ defines a new perceptron $(w', \theta')$ with non-negative weights, which operates the same classification as the perceptron $(w, \theta)$ is equivalent to $p > \theta$ The result directly follows from a similar argument by observing that for all grid-populations pattern $c$ in$\mathcal{C}_\lambda$

$$w^T c - \theta = (w - \theta\mathbf{1})^T c, \tag{16}$$

which implies that if the perceptron models $(w, \theta)$ and $(w - \theta\mathbf{1}, 0)$ achieve the same linear classification.

Our goal is to study the multi-field structure of place-cell perceptrons, which amounts to characterize the two-class linear classifications of grid-like inputs $\mathcal{C}_\lambda$. The study of linear binary classifications has a long history in machine learning. Given a collection of $\Lambda$ input patterns, there are $2^\Lambda$ possible assignments of binary labels to these patterns, also referred to as dichotomies. In general, not all dichotomies can be linearly classified by a perceptron. Those dichotomies that can be classified are called linearly separable. An important question is to compute the number of linearly separable dichotomies, which depends on the geometrical arrangement of the inputs presented to the perceptron. Remarkably, Cover's function counting theorem specifies the exact number of linearly separable dichotomies for $P$ inputs represented as points in a $N$-dimensional space (**Cover, 1965**). For inputs in general position, the number of dichotomies realizable by a zero-threshold perceptron is given by

$$\mathcal{N}_{P,N} = 2 \sum_{k=0}^{N-1} \binom{P-1}{k}, \tag{17}$$

which shows that all dichotomies are possible as long as $P \leq N$. A collection of points $\{x_1, \ldots, x_P\}$ in an $N$-dimensional space is in general position if no subset of $n + 1$ points lies on a $(n - 1)$-dimensional plane for all $n \leq N$. In our modeling framework, the inputs are collections of points representing grid-like inputs $\mathcal{C}_\lambda$. As opposed to Cover's theorem assumptions, these grid-like inputs are not in general position as soon as we consider grid code with more than one module. For instance, it is not hard to see that for $\lambda = (2, 3)$, the patterns $(1, 0|1, 0, 0)$, $(1, 0|0, 1, 0)$, $(0, 1|1, 0, 0)$ and $(0, 1|0, 1, 0)$ are not in general position for being the vertices of a square, therefore lying in a 2D plane. Nongeneric arrangements of grid-like inputs are due to symmetries that are inherent to the modular structure of the grid code. We expect such symmetries to heavily restrict the set of linearly separable dichotomies, therefore constraining the multi-field structure of a place cell perceptron.

We justify the above expectation by discussing the problem of linear separability for two codes that are related to the grid code. These two codes are the 'one-hot' code, whereby a single cell is active for all input pattern, and the 'binary' code, whereby the set of input patterns enumerate all possible binary vectors of activity. Exemplars of grid-like inputs for the one-hot code and the binary code are given for $N = 3$ input cells by

$$\mathcal{C}_{\mathrm{oh}} = \left\{ \begin{matrix} 1 & 0 & 0 \\ 0 & 1 & 0 \\ 0 & 0 & 1 \end{matrix} \right\} \quad \text{and} \quad \mathcal{C}_{\mathrm{b}} = \left\{ \begin{matrix} 0 & 0 & 0 & 0 & 1 & 1 & 1 & 1 \\ 0 & 0 & 1 & 1 & 0 & 0 & 1 & 1 \\ 0 & 1 & 0 & 1 & 0 & 1 & 0 & 1 \end{matrix} \right\}. \tag{18}$$

From a geometrical point of view, a set of points representing the grid-like inputs $\mathcal{S}_J \subset \mathcal{C}$ is linearly separable if there is a hyperplane separating the points $\mathcal{S}$ from the other points $\mathcal{C} \setminus \mathcal{S}$. The existence of a hyperplane separating a single point from all other points is straightforward when the set of patterns correspond to the vertices of a convex polytope. Then, every vertex can be linearly separated from the other points for being an extreme point. It turns out that both the population patterns of the one-hot code and of the binary code represent the vertices of a convex polytope:

a simplex for the single-cell code and a hypercube for the binary code. However, because these vertices are in general position for the single-cell code but not for the binary code, the fraction of linearly separable dichotomies drastically differs for the two codes.

Let us first consider the $N$ points whose coordinates are given by $C_{\mathrm{oh}}$. The convex hull of $C_{\mathrm{oh}}$ is the canonical $(N-1)$-dimensional simplex. Thus, any sets of $k$ vertices, $1 \leq k \leq N$, specifies a $(k-1)$-dimensional face of the simplex, and as such, is a linearly separable $k$-dichotomy. This immediately shows that all dichotomies are linearly separable. This result follows from the fact that the $N$ points in $C_{\mathrm{oh}}$ are in general position. Let us then consider the $2^N$ points whose coordinates are given by $C_{\mathrm{b}}$. The convex hull of $C_{\mathrm{b}}$ is the canonical $N$-dimensional hypercube. Thus, by contrast with $C_{\mathrm{oh}}$, the points in $C_{\mathrm{b}}$ are not in general position. As a result, there are dichotomies that are not linearly separable as shown by considering. For instance, the pair $\{(1,0,0),\,(0,1,0)\}$ and the pair $\{(0,0,0),\,(1,1,1)\}$ can be linearly separated from the other points of the hypercube. Determining the number of linearly separable sets of hypercube vertices is a hard combinatorial problem that has attracted a lot of interest (**Peled and Simeone, 1985**; **Hegedüs and Megiddo, 1996**). Unfortunately, there is no efficient characterization of that number as a function of the dimension $N$. However, it is known that out of the $2^{2^N}$ possible dichotomies, the total number of linearly separable dichotomies scales as $2^{N^2}$ in the limit of large dimension $N \to \infty$ (**Irmatov, 1993**). This shows that only a vanishingly small fraction of hypercube dichotomies are also linearly separable.

## Grid code convex polytope

It is beneficial to gain geometric intuition about grid-like inputs to characterize their linearly separable dichotomies. As binary vectors of length $N$, grid-like inputs form a subset of the $2^N$ vertices of the $N$-dimensional hypercube. Just as for the one-hot and binary codes, linear separability of sets of grid-like inputs can be seen as a geometric problem about polytopes. To clarify this point, let us denote by $H_{\boldsymbol{\lambda}}$ the convex hull of grid-like inputs $C_{\boldsymbol{\lambda}}$. By definition, we have

$$H_{\boldsymbol{\lambda}} = \left\{ \sum_{i=1}^{L} \alpha_i \boldsymbol{c}_i \,\middle|\, \alpha_i \geq 0 \,, gt \sum_i \alpha_i = 1 \right\}, \tag{19}$$

where $\boldsymbol{c}_i$ in $C_{\boldsymbol{\lambda}}$ denotes the ith column of $A_{\boldsymbol{\lambda}}$. The convex hull $H_{\boldsymbol{\lambda}}$ turns out to have a simple geometric structure.

## Proposition 3

For integer periods $\boldsymbol{\lambda} = (\lambda_1, \ldots, \lambda_M)$, the convex hull generated by $C_{\boldsymbol{\lambda}}$, the set of grid-cell-population patterns, determines a $d$-dimensional polytope $H_{\boldsymbol{\lambda}}$, with $d = \sum_{m=1}^{M} \lambda_m - M$, defined as $H_{\boldsymbol{\lambda}} = \Delta^{\lambda_1} \times \ldots \times \Delta^{\lambda_M}$ where $\Delta^{\lambda_m}$, $1 \leq m \leq M$, denotes the $(\lambda_m - 1)$-simplex specified by the $\lambda_m$ points: $(1, 0, \ldots, 0)\,, (0, 1, 0 \ldots, 0)\,, \ldots, (0, \ldots, 0, 1)$.

Before proving the product decomposition of $H_{\boldsymbol{\lambda}}$, let us make a couple of observations. First, observe that all the vectors $\boldsymbol{c}$ in $C_{\boldsymbol{\lambda}}$ satisfies $\mathbf{1}^T \boldsymbol{c} = M$, so that all edges $\boldsymbol{c} - \boldsymbol{c}'$, with $\boldsymbol{c}$, $\boldsymbol{c}'$ in $C_{\boldsymbol{\lambda}}$, lie in the same hyperplane of the vector space $V_{\boldsymbol{\lambda}}$. By Proposition 1, $V_{\boldsymbol{\lambda}}$ has dimension $N = \sum_m \lambda_m - M + 1$, this implies that the dimension of the polytope $H_{\boldsymbol{\lambda}}$ is at most $d = N - 1$. Second, observe that the set $C_{\boldsymbol{\lambda}}$ is left unchanged by the symmetry operators $J_{\lambda_m}$, $1 \leq m \leq M$, where $J_{\lambda_m}$ cyclically shifts downward the mth module coordinates of the vectors in $C_{\boldsymbol{\lambda}}$. The operators $J_{\lambda_m}$ admit the matrix representation

$$J_{\lambda_m} = \begin{pmatrix} I_{\lambda_1 + \ldots + \lambda_{m-1}} & & \\ & J_{\lambda_m} & \\ & & I_{\lambda_{m+1} + \ldots + \lambda_M} \end{pmatrix} \quad \text{with} \quad J_n = \begin{pmatrix} 0 & 1 & 0 & 0 & \ldots \\ 0 & 0 & 1 & 0 & \ldots \\ \vdots & & \ddots & \ddots & \ddots \\ 1 & 0 & \ldots & & \end{pmatrix}, \in \mathbb{R}^n \times \mathbb{R}^n, \tag{20}$$

where $I_n$ denotes the identity matrix in $\mathbb{R}^n \times \mathbb{R}^n$. Notice that the matrices $J_{\lambda_m}$ satisfy $J_{\lambda_m}^T J_{\lambda_m} = I_{\lambda_1 + \ldots + \lambda_M}$ showing that the operators $J_{\lambda_m}$ are isometries in $V_{\boldsymbol{\lambda}}$. Moreover, observe that for all $\boldsymbol{c}$, $\boldsymbol{c}'$ in $C_{\boldsymbol{\lambda}}$, there are integers of $k_1, \ldots, k_M$ such that $J_{\lambda_1}^{k_1} \ldots J_{\lambda_1}^{k_M} \boldsymbol{c} = \boldsymbol{c}'$. This shows that each vector in $C_{\boldsymbol{\lambda}}$ plays the same role in defining the geometry of $H_{\boldsymbol{\lambda}}$, and thus $H_{\boldsymbol{\lambda}}$ is vertex-transitive. In particular, every vector in $C_{\boldsymbol{\lambda}}$ represents an extreme point of the convex hull $H_{\boldsymbol{\lambda}}$. As a result, $H_{\boldsymbol{\lambda}}$ is a polytope with as many vertices

as the cardinality of $C_\lambda$, that is, $\Lambda = \prod_{m=1}^{M} \lambda_m$. The product decomposition of the polytope $H_\lambda$ then follows from a simple recurrence argument over the number of modules $M$.

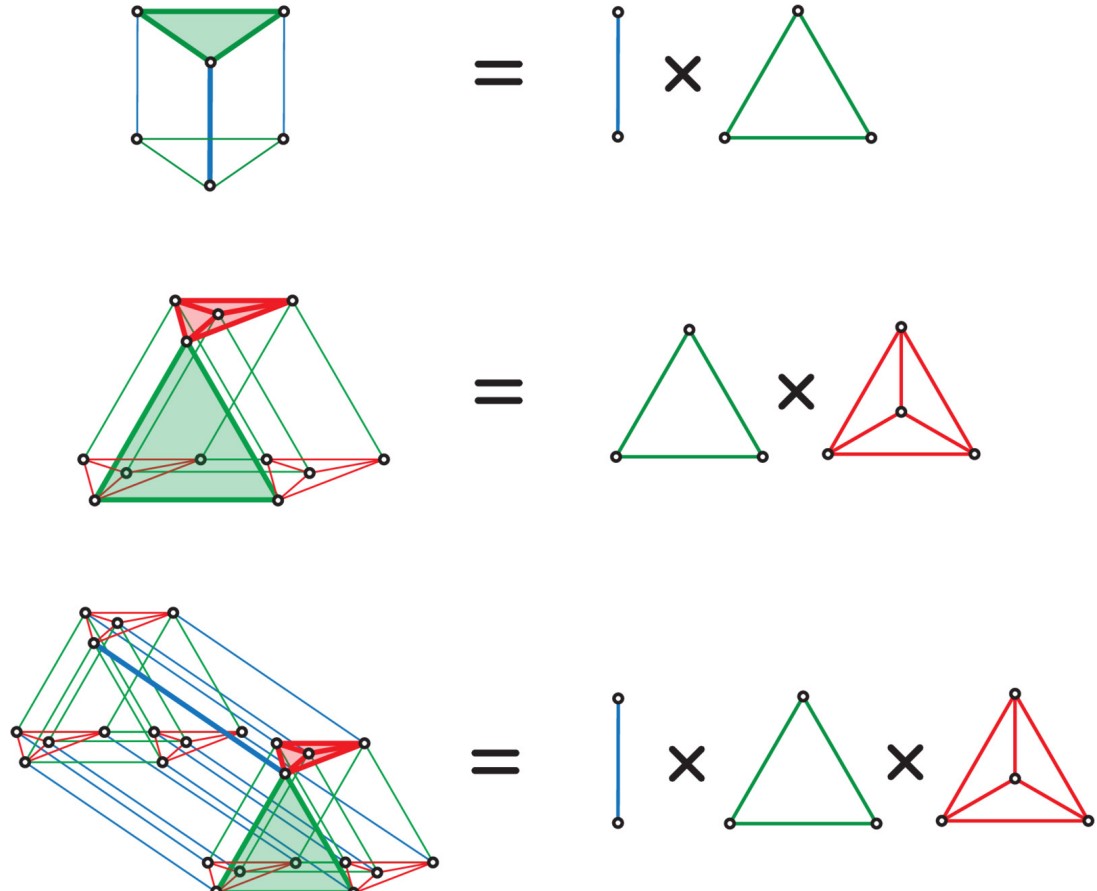

**Appendix 1—figure 1.** Simplicial decomposition. The convex hull generated by the grid code activity patterns is a product of simplices.

Proof. In order to relate the geometrical structure of $H_\lambda$ to that of simplices, let us introduce $e_i$, $1 \le i \le \lambda_M$, the elementary unit vector corresponding to the -th coordinate of $\mathbb{R}^{\lambda_M}$. The set $C_\lambda$ has the following product structure

$$C_\lambda = \left\{ c = (c', e_i) \,\middle|\, c' \in C_{\lambda'}, \, 0 < \lambda_M \right\}, \tag{21}$$

where $C_{\lambda'}$ is the set of vectors for $M-1$ modules with periods $\lambda' = \{\lambda_1, \ldots, \lambda_{M-1}\}$. The product structure of the set $C_\lambda$ transfers to the convex hull $H_\lambda$ it generates. Specifically, we have

$$H_\lambda = \left\{ \sum_{i=1}^{\lambda_M} \sum_{j=1}^{L/\lambda_M} \alpha_{ij}(c_j, e_i) \,\middle|\, \sum_{i=1}^{\lambda_M} \sum_{j=1}^{L/\lambda_M} \alpha_{ij} = 1 \right\}, \tag{22}$$

$$= \left\{ \left( \sum_{j=1}^{L/\lambda_M} \left( \sum_{i=1}^{\lambda_M} \alpha_{ij} \right) c_j, \sum_{i=1}^{\lambda_M} \left( \sum_{j=1}^{L/\lambda_M} \alpha_{ij} \right) e_i \right) \,\middle|\, \sum_{i=1}^{\lambda_M} \sum_{j=1}^{L/\lambda_M} \alpha_{ij} = 1 \right\}, \tag{23}$$

$$= \left\{ \left( \sum_{j=1}^{L/\lambda_M} \beta_j c_j, \sum_{i=1}^{\lambda_M} \gamma_i e_i \right) \,\middle|\, \sum_{j=1}^{L/\lambda_M} \beta_j = 1, \, \sum_{i=1}^{\lambda_M} \gamma_i = 1 \right\}, \tag{24}$$

$$= \left\{ (\boldsymbol{c}',\boldsymbol{\delta}) \ \middle| \ \boldsymbol{c}' \in H_{\boldsymbol{\lambda}'} , \ \boldsymbol{\delta} \in \Delta^{\lambda_M} \right\} , \tag{25}$$

where we have recognized that the convex hull of the set of elementary basis vectors $\boldsymbol{e}_i, 1 \leq i \leq \lambda_M$, is precisely the canonical $(\lambda_M - 1)$-simplex. Thus, we have shown that $H_{\boldsymbol{\lambda}} = H_{\boldsymbol{\lambda}'} \times \Delta^{\lambda_M}$. Proceeding by recurrence on the number of modules, one obtains the announced decomposition of the convex hull as a product $H = \Delta^{\lambda_1} \times \ldots \times \Delta^{\lambda_M}$, where $\Delta^{\lambda_M}, 1 \leq m \leq M$, is the canonical $(\lambda_m - 1)$-simplex.

The above orthogonal decomposition suggests that the problem of determining the linearly separable dichotomies of grid-like inputs is related to that of determining the linearly separable Boolean functions. Indeed, the polytope defined by grid-like inputs with $M$ modules contains $M$-dimensional hypercubes, for which many dichotomies are not linearly separable. As counting the linearly separable Boolean functions is a notoriously hard combinatorial problem, it is unlikely that one can find a general characterization of the linearly separable dichotomies of grid-like inputs. However, it is possible to give some explicit results for the case of two modules $M$ or for the case of $k$-dichotomies for small cardinality $k$.

# Appendix 2

## Combinatorics of linearly separable dichotomies

In this Appendix, we establish combinatorial results about the properties and the cardinality of linearly separable dichotomies of grid-like inputs. First, we show that linearly separable dichotomies can be partitioned in classes, each indexed by a combinatorial object called Young diagram. Second, we exploit related combinatorial objects, called Young tableaux, to show that not all Young diagrams correspond to linearly separable dichotomies. Third, we utilize Young diagrams to characterize dichotomies for which one class of labeled patterns has small cardinality $k = 1, \ldots, 4$. Fourth, we count the exact number of linearly separable dichotomies for grid-like inputs with two modules.

## Relation to Young diagrams

To count linearly separable dichotomies, we first show that these dichotomies can be partitioned in classes that are indexed by Young diagrams. Young diagrams are useful combinatorial objects that have been used to study, e.g., the properties of the group representations of the symmetric group and of the general linear group. Young diagrams are formally defined as follows:

## Definition 1

A $d$-dimensional Young diagram is a subset $D$ of lattice points in the positive orthant of a d-dimensional integral lattice, which satisfies the following:

 i. If $(n_1, \ldots, n_i, \ldots, n_d) \in D$ and $n_i > 0$, then $(n_1, \ldots, n_i - 1, \ldots, n_d) \in D$.

 ii. For any positive integer $i \le d$, and any non negative integers, $m, p$, with $m > p$, the restriction of $D$ to the hyperplane $n_i = m$ is a (d−1)-dimensional Young diagram that covers the (d − 1)-dimensional Young diagram formed by the restriction of S to the hyperplane $n_i = p$.

Moreover, the size of the diagram $D$, denoted by $|D|$, is defined as the number of lattice points in $D$.

 Young diagrams have been primarily studied for $d = 2$ because their use allows oneto conveniently enumerate the partitions of the integers. For $d = 2$, there are differentconventions for representing Young diagrams pictorially. Hereafter, we follow the Frenchnotations, where Young diagrams are left justified lattice rows, whose length decreaseswith height. For the sake of clarity, Fig. 1a depicts the 5 Young diagrams associated to thepartitions of 4: 4, 3 + 1, 2 + 2, 2 + 1 + 1 and 1 + 1 + 1 + 1: Young diagrams have been less studiedfor dimensions $d \ge 3$ and only a few of their combinatorial properties are known. Fig. 1brepresents a 3-dimensional diagram, together with two 2-dimensional restrictions (red edgesfor $n_3 = 1$ and yellow edges for $n_3 = 3$). Observe that these restrictions are 2-dimensionalYoung diagrams, and that the restriction corresponding to $n_3 = 1$ covers the restriction corresponding to $n_3 = 3$. Young diagrams can equivalently be viewed as arrays of boxesrather than lattice points in the positive orthant. This corresponds to identifying each latticepoint $(n_1, \ldots, n_d) \in D$ with the unit cube $(n_1 - 1, n_1) \times \ldots \times (n_d - 1, n_d)$.

 Before motivating the use of Young diagrams, let us make a few remarks about the set ofdichotomies that can be realized by a perceptron with fixed weight vector $(\omega, \theta)$. First, recallthat with no loss of generality we can restrict the weight vectors $\omega$ to be nonnegative byProposition 2. Second, by permutation invariance, there is no loss of generality in consideringa perceptron $(\omega, \theta)$ for which the weight vector.

$$w = (w_{1,1}, ..., w_{1,\lambda_1} | ... \ ... | w_{M,1}, ..., w_{M,\lambda_M})$$

(26)

 is such that the weights are ordered within each module: $w_{m,1} < \ldots < w_{m,\lambda_m}$ for all $m, 1 \le m \le M$. We refer to weight vectors having this module-specific, increasing order propertyas being a modularly ordered weight vector. Bearing these observations in mind, the following proposition establishes the link between Young diagrams and perceptrons.

## Proposition 4

Given integer periods $\boldsymbol{\lambda} = (\lambda_1, \ldots, \lambda_M)$, for all modularly ordered, non-negative, weight vectors $\boldsymbol{w}$ and for all thresholds $\theta$, the lattice set

$$\mathcal{D}(\boldsymbol{w}, \theta) = \left\{ (i_1, \ldots, i_M) \in \{1, \ldots \lambda_1\} \times \ldots \times \{1, \ldots \lambda_M\} \ \middle| \ \sum_{m=1}^{M} w_{m,i_m} \le \theta \right\}$$

(27)

is a $M$-dimensional Young diagram in $\{1, \ldots \lambda_1\} \times \ldots \times \{1, \ldots \lambda_M\}$.

In other words, under assumption of modularly ordered, non-negative weights, the phase indices of inactive grid cells form a Young diagram.

Proof. The Young diagram properties directly follow from the ordering of weights within modules. For instance, it is easy to see that if $(i_1, \ldots, i_M) \leq (j_1, \ldots, j_M)$ for the component-wise partial order in $\{1, \ldots \lambda_1\} \times \ldots \times \{1, \ldots \lambda_M\}$, then $(j_1, \ldots, j_M) \in \mathcal{D}(\boldsymbol{w}, \theta)$ implies $(i_1, \ldots, i_M) \in \mathcal{D}(\boldsymbol{w}, \theta)$. Indeed, we necessarily have

$$\sum_{m=1}^{M} w_{m,i_m} \leq \sum_{m=1}^{M} w_{m,j_m} < \theta \,. \tag{28}$$

By the above proposition, given a grid code with $M$ modules, every perceptron $(\boldsymbol{w}, \theta)$ acting on that grid code can be associated to a unique $M$-dimensional Young diagram $\mathcal{D}(\boldsymbol{w}, \theta)$ after ordering the components of $\boldsymbol{w}$ within each module. Conversely, if a $M$-dimensional Young diagram $\mathcal{D}'$ can be associated to a perceptron $(\boldsymbol{w}, \theta)$ with modularly ordered, non-negative weights, we say that $\mathcal{D}' = \mathcal{D}(\boldsymbol{w}, \theta)$ is realizable. Then a natural question to ask is: are all $M$-dimensional Young diagrams realizable by perceptrons? It turns out that perceptrons exhaustively enumerate all $M$-dimensional Young diagrams if $M \leq 2$, but there are unrealizable Young diagrams as soon as $M > 2$.

## Relation to Young tableaux

Understanding why there are unrealizable Young diagrams as soon as $M > 2$ involves using combinatorial objects that are closely related to Young diagrams, called Young tableaux.

## Definition 2

Given a Young diagram $\mathcal{D}$, a Young tableau $\mathcal{T}$ is obtained by labeling the lattice points – or filling in the boxes – of $\mathcal{D}$ with the integers $1, 2, \ldots, |\mathcal{D}|$, such that each number occurs exactly once and such that the entries are increasing across each row (to the right) and across each column (to the top).

Here are two examples of Young tableaux that are distinct labeling of the same Young diagram:

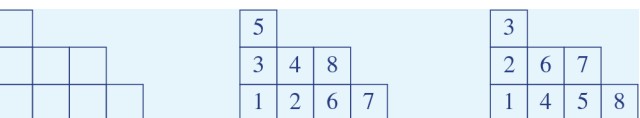

**Appendix 2—scheme 1.** Examples of Young tableaux.

Just as Young diagrams, Young tableaux are naturally associated to perceptrons. The following arguments specify the correspondence between perceptrons and Young tableaux. Given a perceptron $(\boldsymbol{w}, \theta)$ with modularly ordered, non-negative weights, let us order all patterns in $\mathcal{C}_{\boldsymbol{\lambda}}$ by increasing level of perceptron activity. Specifically, set $\mathcal{J}_0 = \mathcal{C}_{\boldsymbol{\lambda}}$ and define iteratively for $k$, $0 \leq k < \Lambda$,

$$\boldsymbol{c}_{k+1}^{\star}(\boldsymbol{w}) = \arg \min_{\boldsymbol{c} \in \mathcal{J}_k(\boldsymbol{w})} \boldsymbol{w}^T \boldsymbol{c} \,, \quad \mathcal{J}_{k+1}(\boldsymbol{w}) = \mathcal{J}_k(\boldsymbol{w}) \setminus \{\boldsymbol{c}_{k+1}^{\star}(\boldsymbol{w})\} \,. \tag{29}$$

With no loss of generality, we can assume that all patterns achieve distinct levels of activity, so that there is a unique minimizer for all $k$, $0 \leq k < \Lambda$. With that assumption, the sequence $\boldsymbol{c}_k^{\star}(\boldsymbol{w})$, $1 \leq k \leq \Lambda$, enumerates unambiguously all patterns in $\mathcal{C}_{\boldsymbol{\lambda}}$ by increasing level of activity. The Young tableau associated to the perceptron $(\boldsymbol{w}, \theta)$, denoted by $\mathcal{T}(\boldsymbol{w}, \theta)$, is then obtained by labeling lattice points of the Young diagram $\mathcal{D}(\boldsymbol{w}, \theta)$ by increasing level of activity as in the sequence $\boldsymbol{c}_k^{\star}(\boldsymbol{w})$, $1 \leq k \leq |\mathcal{D}(\boldsymbol{w}, \theta)|$. One can check that such labeling yields a tableau as the resulting labels increase along each rows (to the right) and columns (to the top). Within this framework, we say that a Young tableau $\mathcal{T}'$ is realizable if there is a perceptron $(\boldsymbol{w}, \theta)$ such that $\mathcal{T}' = \mathcal{T}(\boldsymbol{w}, \theta)$. Finally, let us define the sequence of thresholds $\theta_k(\boldsymbol{w})$, $0 \leq k \leq \Lambda + 1$, such that $\theta_0 = -\infty$, $\theta_{\Lambda+1}(\boldsymbol{w}) = \infty$, and for $0 < k \leq \Lambda$

$$\theta_k(\boldsymbol{w}) = \min_{\boldsymbol{c} \in \mathcal{J}_{k-1}(\boldsymbol{w})} \boldsymbol{w}^T \boldsymbol{c} = \boldsymbol{w}^T \boldsymbol{c}_k^{\star}(\boldsymbol{w}) \,. \tag{30}$$

Then, observe that for all $k$, $0 \leq k \leq \Lambda$, the set of active patterns $\mathcal{J}_k(\boldsymbol{w})$ is linearly separable for threshold $\theta$ satisfying $\theta_k(\boldsymbol{w}) \leq \theta < \theta_{k+1}(\boldsymbol{w})$. In fact, the sequence $\{\mathcal{J}_k(\boldsymbol{w})\}_{0 \leq k \leq \Lambda}$ represents all the

linearly separable dichotomies realizable by changing the threshold of a perceptron with weight vector $w$. This fact will be useful to prove the following proposition, which justifies considering Young tableaux.

## Proposition 5

All $M$-dimensional Young diagrams are realizable if and only if all $(M-1)$-dimensional Young tableaux are realizable.

Observe that the above proposition does not mention the periods $\lambda_1, \ldots, \lambda_M$. This is because the proposition deals with the correspondence between $m$-dimensional Young diagrams and $(M-1)$-dimensional Young tableaux for all possible assignments of periods.

Proof. In this proof, we use prime notations for quantities relating to $M-1$ modules and regular notations for quantities relating to $m$ modules. For instance, $\boldsymbol{\lambda}$ denotes an arbitrary assignment of $m$ periods $\{\lambda_1, \ldots, \lambda_M\}$ and $\boldsymbol{\lambda}'$ denotes its $m-1$ first components $\{\lambda_1, \ldots, \lambda_{M-1}\}$. With this preamble, we give the 'if' part of proof in ($i$) and the 'only if' part in ($ii$).

($i$) Given a $(M-1)$-dimensional Young tableau $\mathcal{T}'$ with diagram $\mathcal{D}'$, let us consider the smallest periods $\boldsymbol{\lambda}'$ such that $\mathcal{D}' \subset \{1, \ldots, \lambda_1\} \times \ldots \times \{1, \ldots, \lambda_{M-1}\}$. The 'if' part of the proof will follow from showing that if all $(M-1)$-dimensional tableaux $\mathcal{T}'$ with Young diagram $\mathcal{D}'$ are realizable, than all $M$-dimensional Young diagrams whose restriction to $\{1, \ldots \lambda_1\} \times \ldots \times \{1, \ldots \lambda_{M-1}\} \times \{1\}$ is $\mathcal{D}'$ are realizable. To prove this property, observe that all the $M$-dimensional Young diagrams with restriction $\mathcal{D}'$ are obtained as finite sequences of $(M-1)$-dimensional Young diagrams $\mathcal{D}' = \mathcal{D}'_1 \supset \mathcal{D}'_2 \supset \ldots \supset \mathcal{D}'_{\lambda_M}$, for some $\lambda_M$ specifying the minimum period in the mth dimension. For all such sequences, consider a tableau $\mathcal{T}'$ labeling $\mathcal{D}'$ such that for all , $1 \leq i \leq \lambda_M - 1$, the labels of $\mathcal{D}'_{i+1}$ are smaller than the labels $\mathcal{D}'_i \setminus \mathcal{D}'_{i+1}$. Such a tableau is always possible because of the nested property of the sequence of diagrams $\mathcal{D}'_i$, $1 \leq i \leq \lambda_M$. Now, suppose that the Young tableau $\mathcal{T}'$ is realizable. This means that there is a perceptron $(w', \theta')$ acting on the grid-like inputs in $\mathcal{C}_{\boldsymbol{\lambda}'}$ such that $\mathcal{T}' = \mathcal{T}(w', \theta')$. With no loss of generality, the weight vector $w'$ specifies a sequence of patterns $c_k^\star(w')$, $1 \leq k \leq \Lambda'$, and a sequence of thresholds $\theta_k(w')$, $1 \leq k \leq \Lambda'$, such that (1) enumerates the elements of $\mathcal{C}_{\boldsymbol{\lambda}'}$ by increasing level of activity and (2) for all $0 \leq k \leq |\mathcal{D}'|$, the set of active patterns $\mathcal{J}_k(w)$ defined in (29) is linearly separable if and only if $\theta_k(w') \leq \theta < \theta_{k+1}(w')$. Then by construction, the diagrams $\mathcal{D}'_i$, $1 \leq i \leq \lambda_M$, are realized by a perceptron $(w', \theta'_i)$, where every $\theta'_i \geq \theta'$ is such that $\theta_{\Lambda - |\mathcal{D}'_i|}(w') < \theta'_i < \theta_{\Lambda - |\mathcal{D}'_i|+1}(w')$. We are now in a position to construct a $M$-module perceptron $(w, \theta')$ realizing the sequence $\mathcal{D}' = \mathcal{D}'_1 \supset \mathcal{D}'_2 \supset \ldots \supset \mathcal{D}'_{\lambda_M}$. To do so, it is enough to specify the components $w_{M,1}, \ldots, w_{M,\lambda_M}$ of the Mth module of a weight vector $w$ since the other components will coincide with $w'$. One can check that choosing $w_{M,i} = \theta'_i - \theta'$ defines an admissible increasing sequence of non-negative weights.

($ii$) For the 'only if' part, let us consider an arbitrary $(M-1)$-dimensional Young tableau $\mathcal{T}'$, with diagram $\mathcal{D}'$ such that $|\mathcal{D}'| = p$. Then let us consider the $m$-dimensional Young diagram $\mathcal{D}$ obtained via the sequence of $(M-1)$-dimensional diagrams $\mathcal{D}' = \mathcal{D}'_1 \supset \mathcal{D}'_2 \supset \ldots \supset \mathcal{D}'_p$, where for all $q$, $1 \leq q < p$, $\mathcal{D}'_q \setminus \mathcal{D}'_{q+1}$ is a singleton containing the lattice point labeled by $p - q + 1$. Moreover, let us consider the smallest periods $\boldsymbol{\lambda}$ such that $\mathcal{D} \subset \{1, \ldots, \lambda_1\} \times \ldots \times \{1, \ldots, \lambda_M\}$. Now, suppose that all $m$-dimensional Young diagrams are realizable. Then, there is a perceptron $(w, \theta)$ acting on $\mathcal{C}_{\boldsymbol{\lambda}}$ with modularly ordered, non-negative weights such that $\mathcal{D} = \mathcal{D}(w, \theta)$. This means that for all , $1 \leq q \leq p$, the diagram $\mathcal{D}'_q$ is realized by the perceptron $(w', \theta - w_{M,q})$, where $w'$ collect the components of $w$ that correspond to $m-1$ first modules. Then, let us consider the pattern $c_q$ represented by the lattice point in the singleton $\mathcal{D}'_q \setminus \mathcal{D}'_{q+1}$. Remember that a pattern $c$ is identified to the lattice point $(i_1, \ldots, i_M)$, whose coordinates are given by the phase of the active neuron within each module. Then, by the increasing property of the weights, we necessarily have $\theta - w_{M,q+1} \leq w'^T c_q < \theta - w_{M,q}$, which implies that the Young tableaux $\mathcal{D}'$ is realized by the perceptron $(w', \theta - w_{M,1})$.

It is straightforward to check that all 1D Young tableaux are realizable, so that all 2D Young diagrams are realizable. However, the following counterexample shows that not all 2D Young tableaux are realizable, so that $M$-dimensional Young diagrams with $M > 2$ are not all realizable.

Counterexample 1. The 2D Young tableaux defined as

|     | 4 | 8 | 9 |
|-----|---|---|---|
| T=  | 3 | 5 | 7 |
|     | 1 | 2 | 6 |

is not realizable.

Proof. Suppose there is a perceptron with modularly ordered, non-negative, weight vector $\boldsymbol{w} = (w_{1,1}, w_{1,2}, w_{1,3}, w_{2,1}, w_{2,2}, w_{2,3})$ realizing $\mathcal{D}$. By convention, we consider that the first module corresponds to the horizontal axis and the second module corresponds to the vertical axis. The labeling of $\mathcal{T}$ implies order relations among read-out activities via $\boldsymbol{w}$. Specifically, the activities can be listed by increasing order as $w_{1,1} + w_{2,1} < w_{1,2} + w_{2,1} < w_{1,1} + w_{2,2} < w_{1,1} + w_{2,3} < \ldots$. We are going to show that such an order is impossible by contradiction. To do so, let us introduce the weight differences $u_1 = w_{1,2} - w_{1,1}$, $u_2 = w_{1,3} - w_{1,2}$ associated to the first module and the weight differences $v_1 = w_{2,2} - w_{2,1}$, $v_2 = w_{2,3} - w_{2,2}$ associated to the second module. These differences satisfy incompatible order relations. Specifically: (1) the sequence $2 \to 3$ in $\mathcal{T}$ implies that the cost to go right, that is, $u_1 = w_{1,2} - w_{1,1}$, is less than the cost to go up, that is, $v_1 = w_{2,2} - w_{2,1}$. Otherwise, the label 2 would be on top the label 1. Thus, we necessarily have $u_1 < v_1$. The same reasoning for the sequence $4 \to 5$ implies $v_2 < u_1$, so that we have $v_2 < u_1$ The sequence $5 \to 6$ implies $v_1 < u_2$, and the sequence $7 \to 8$ implies $u_2 < v_2$, so that we have $v_1 < v_2$. Thus, assuming that $\mathcal{T}$ is realizable leads to considering weights for which $v_2 < v_1$ and $v_1 < v_2$—a contradiction.

## Linearly separable dichotomies for realizable Young diagrams

Consider a Young $M$-dimensional diagram $\mathcal{D} \in \{1, \ldots \lambda_1\} \times \ldots \times \{1, \ldots \lambda_M\}$ that can be realized by a perceptron with modularly ordered, non-negative weights. Such a Young diagram $\mathcal{D}$ is the lattice set whose points represent the phase indices of inactive grid-like inputs. Indeed, if $(i_1, \ldots, i_M) \in \mathcal{D}$, we have $\sum_{m=1}^{M} w_{m,i_m} \leq \theta$, which means that the perceptron is inactive for the grid-like input $\boldsymbol{c}$ in $\mathcal{C}_\lambda$ obtained by setting $c_{m,i_m} = 1$ for all $1 \leq m \leq M$. Thus, the perceptron implements the dichotomy for which the inactive grid-like inputs are exactly represented by $\mathcal{D}$. Are there more dichotomies associated to $\mathcal{D}$? Answering this question requires revisiting the correspondence between perceptrons and Young diagrams. The key property in establishing this correspondence is the assumption of modularly ordered weights. In Section B.1, we justified that such an assumption incurs no loss of generality by permutation invariance of the grid cells within each modules. Thus, each Young diagram $\mathcal{D}$ is in fact associated to the class of perceptrons

$$\{(P\boldsymbol{w}, \theta) \,|\, \mathcal{D} = \mathcal{D}(\boldsymbol{w}, \theta) \quad P \in \Pi_\lambda\}, \tag{31}$$

where $\Pi_\lambda$ denotes the set of permutation matrix stabilizing the modules of periods $\lambda$. Clearly, for $P \neq P'$, the perceptron $(P\boldsymbol{w}, \theta)$ generally implements a distinct dichotomy than that of $(P'\boldsymbol{w}, \theta)$. As a result, there is a class of dichotomies indexed by the Young diagram $\mathcal{D}$, which we denote by $C(\mathcal{D})$.

Evaluating the cardinality of $C(\mathcal{D})$ via simple combinatorial arguments first requires a crude description of the geometry of $\mathcal{D}$, and specifically of its degenerate symmetries. For all $1 \leq m \leq M$, $1 \leq i \leq \lambda_m$, let us denote the restriction of $\mathcal{D}$ to the hyperplane $i_m = i$ by

$$R_{m,i}\left(\mathcal{D}\right) = \{(i_1, \ldots, i_M) \in \mathcal{D} \,|\, i_m = i\}. \tag{32}$$

By definition of the Young diagrams, we have $R_{m,i}\left(\mathcal{D}\right) \supset R_{m,i+1}\left(\mathcal{D}\right)$ for all $1 \leq i < \lambda_m$. We say that a Young diagram exhibits a degenerate symmetry along the mth dimension whenever two consecutive restrictions coincide: $R_{m,i}\left(\mathcal{D}\right) = R_{m,i+1}\left(\mathcal{D}\right)$. To make the notion of degeneracy more precise, let us consider the equivalence relation on $\{1, \ldots, \lambda_m\}$ defined by $i \sim j \Leftrightarrow R_{m,i}\left(\mathcal{D}\right) = R_{m,j}\left(\mathcal{D}\right)$. Given in $\{1, \ldots, \lambda_m\}$, the equivalence class of is then $\{j \in \{1, \ldots, \lambda_m\} \,|\, R_{m,i}\left(\mathcal{D}\right) = R_{m,j}\left(\mathcal{D}\right)\}$. Let us denote the total number of such equivalence classes by $k_m$, $1 \leq k_m \leq \lambda_m$. Then, the set $\{1, \ldots, \lambda_m\}$ can be partitioned in $k_m$ classes, $C_{m,1}, \ldots, C_{m,k_m}$, where the classes are listed by decreasing order of Young diagrams. For instance $C_1$ comprises all the indices for which the restriction along the mth dimension yields the same Young diagram as $R_{m,1}\left(\mathcal{D}\right)$. We denote the cardinality of the thus-ordered equivalence classes by $\sigma_{m,k} = |C_{m,k}|$, $1 \leq k \leq k_m$, so that we have $\lambda_m = \sigma_{m,1} + \ldots + \sigma_{m,k_m}$. We refer to the

$\sigma_{m,k}$ as the degeneracy indices. Degenerate symmetries correspond to degeneracy indices $\sigma_{m,k} > 1$. We are now in a position to determine the cardinality of $C(\mathcal{D})$:

## Proposition 6

For integer periods $\lambda_1, \ldots, \lambda_M$, let us consider a realizable Young diagram $\mathcal{D}$ in $\{1, \ldots, \lambda_1\} \times \ldots \times \{1, \ldots, \lambda_M\}$. Then, the class of linearly separable dichotomies with Young diagram $\mathcal{D}$, denoted by $C(\mathcal{D})$, has cardinality

$$|C(\mathcal{D})| = \prod_{m=1}^{M} \frac{\lambda_m!}{\sigma_{m,k_1}! \ldots \sigma_{m,k_m}!} \, . \tag{33}$$

where $\sigma_{m,k}$, $1 \leq k \leq m$ are the degeneracy indices of the Young diagram along the $m$th dimension.

Proof. A dichotomy is specified by enumerating the set of inactive grid-like inputs $c$ in $\mathcal{C}_\lambda$. Each pattern $c$ can be conveniently represented as a lattice point in $\{1, \ldots, \lambda_1\} \times \ldots \times \{1, \ldots, \lambda_M\}$ by considering the phase indices of the active cell in the $M$ modules of pattern $c$. Thus, a generic dichotomy is just a configuration of lattice points in $\{1, \ldots, \lambda_1\} \times \ldots \times \{1, \ldots, \lambda_M\}$. The class of dichotomies $C(\mathcal{D})$ comprises all lattice-point configurations in $\{1, \ldots, \lambda_1\} \times \ldots \times \{1, \ldots, \lambda_M\}$ obtained by permutations of the indices along the $c$ dimensions:

$$C(\mathcal{D}) = \left\{ \pi_1 \ldots \pi_M \mathcal{D} \mid \pi_1 \in \mathbb{S}_{\lambda_1}, \ldots, \pi_M \in \mathbb{S}_{\lambda_M} \right\}, \tag{34}$$

where we define

$$\pi_1 \ldots \pi_M \mathcal{D} = \left\{ \left( \pi_1(i_1), \ldots, \pi_M(i_M) \right) \mid (i_1, \ldots, i_M) \in \mathcal{D} \right\}, \tag{35}$$

and where $\mathbb{S}_{\lambda_m}$ denotes the set of permutation of $\{1, \ldots, \lambda_m\}$. Let us denote a generic lattice-point configuration in $\{1, \ldots, \lambda_1\} \times \ldots \times \{1, \ldots, \lambda_M\}$ by $\mathcal{S}$. By permuting the indices of the points in $\mathcal{S}$, each transformation $\pi_m$ is actually permuting $R_{m,i}(\mathcal{S})$, $1 \leq i \leq m$, the restrictions of the lattice-point configuration along the $m$th dimension. The partial order defined by inclusion is preserved by permutations in the sense that given $\pi_m$ in $\mathbb{S}_{\lambda_m}$, $1 \leq m \leq M$, we have $R_{m,\pi_m(i)}(\pi_1 \ldots \pi_M \mathcal{S}) \subset R_{m,\pi_m(j)}(\pi_1 \ldots \pi_M \mathcal{S})$ if and only if $R_{m,i}(\mathcal{S}) \subset R_{m,j}(\mathcal{S})$. In particular, $k_m$, the number of restriction classes induced by the relation $i \sim j \Leftrightarrow R_{m,i}(\mathcal{S}) = R_{m,j}(\mathcal{S})$, is invariant to permutations, and so are their cardinalities. These cardinalities specify the degeneracy indices $\sigma_{m,1}, \ldots, \sigma_{m,k_m}$ of $\mathcal{S}$ along the $m$th dimension. Thus, all configurations $\mathcal{S}$ obtained via permutation of $\mathcal{D}$ have the same degeneracy indices as $\mathcal{D}$. Moreover, for a Young diagram $\mathcal{D}$, these degeneracy indices simply count the equivalence classes formed by restrictions of identical size along the same dimension. Thus, the number of dichotomies in $|C(\mathcal{D})|$ is determined as the number of ways to independently assign the indices $\{1, \ldots, \lambda_m\}$ to $k_m$ restriction classes of size $\sigma_{m,1}, \ldots, \sigma_{m,k_m}$ for all $m$, $1 \leq m \leq M$. For each $m$, this number is given by the multinomial coefficient: $\lambda_m!/(\sigma_{m,k_1}! \ldots \sigma_{m,k_m}!)$.

As opposed to the case of random configurations in general position, the many symmetries of the grid-like inputs in $\mathcal{C}_\lambda$ allow one to enumerate dichotomies of specific cardinalities. We define the cardinality of a dichotomy by the size of the set of active pattern it separates. Thus, a perceptron $(\boldsymbol{w}, \theta)$ realizing a $k$-dichotomy is one for which exactly $k$ patterns $c$ in $\mathcal{C}_\lambda$ are such that $\boldsymbol{w}^T \boldsymbol{c} > \theta$. Proposition 7 reduces the problem of counting realizable $k$-dichotomies to that of enumerating realizable Young diagrams $\mathcal{D}$ of size $|\mathcal{D}| = k$. Such an enumeration depends on the number of modules $M$, which sets the dimensionality of the Young diagrams, as well as the periods $\lambda_m$, $1 \leq m \leq M$. Unfortunately, even without considering the constraint of being a realizable Young diagram, there is no convenient way to enumerate Young diagrams of fixed size for general dimension $M$. However, for low cardinality, for example, $k \leq 5$, there are only a few Young diagrams such that $|\mathcal{D}| = k$, and it turns out that all of them are realizable. In the following, and without aiming at exhaustivity, we exploit the latter fact to characterize the sets of $k$-dichotomies for $k \leq 5$ and to compute their cardinalities.

There are $M$ possible $M$-dimensional Young diagram of size 2, according to the dimension along which the two lattice points are positioned. The Young diagram extending along the $m$th dimension, $1 \leq m \leq M$, has degeneracy indices $\sigma_{m,1} = 2$ and $\sigma_{m,2} = \lambda_m - 2$ or $\sigma_{n,1} = 1$ and $\sigma_{n,2} = \lambda_n - 1$ for $n \neq m$. As a result, the number of 2-dichotomies of grid-like inputs is given by

$$\mathcal{N}_2 = \sum_{m=1}^{M} \left( \prod_{n \neq m} \frac{\lambda_n!}{1!(\lambda_n - 1)!} \right) \frac{\lambda_m!}{2!(\lambda_m - 2)!} = \frac{1}{2} \sum_{m=1}^{M} \lambda_m(\lambda_m - 1) \left( \prod_{n \neq m} \lambda_n \right). \tag{36}$$

There are two types of Young diagram of size 3, type (3a) for which the three lattice points span one dimension and type (3b) for which the lattice points span two dimensions. There are $M$ possible M-dimensional Young diagram of type (3a). The degeneracy indices for the Young diagram extending along the mth dimension, $1 \leq m \leq M$, are $\sigma_{m,1} = 3$ and $\sigma_{m,3} = \lambda_m - 3$, and $\sigma_{n,1} = 1$ and $\sigma_{n,2} = \lambda_n - 1$ for $n \neq m$, yielding

$$\mathcal{N}_{3a} = \sum_{m=1}^{M} \left( \prod_{n \neq m} \frac{\lambda_n!}{1!(\lambda_n - 1)!} \right) \frac{\lambda_m!}{3!(\lambda_m - 3)!} = \frac{1}{6} \sum_{m} \left( \prod_{n \neq m} \lambda_n \right) \lambda_m(\lambda_m - 1)(\lambda_m - 2). \tag{37}$$

There are $M(M - 1)/2$ possible $M$-dimensional Young diagram of type (3b), as many as choices of two dimensions among $M$. The degeneracy indices of the Young diagram extending along dimensions $m$ and $n$, $1 \leq m < n \leq M$, are $\sigma_{m,1} = \sigma_{m,2} = 1$ and $\sigma_{m,3} = \lambda_m - 2$, $\sigma_{n,1} = \sigma_{n,2} = 1$ and $\sigma_{n,3} = \lambda_n - 2$, and $\sigma_{k,1} = 1$ and $\sigma_{k,2} = \lambda_k - 1$ for $k \neq m, n$, yielding

$$\mathcal{N}_{3b} = \sum_{1 \leq m < n \leq M} \left( \prod_{k \neq m,n} \frac{\lambda_k!}{1!(\lambda_k - 1)!} \right) \frac{\lambda_m!}{1!1!(\lambda_m - 2)!} \frac{\lambda_n!}{1!1!(\lambda_n - 2)!} \tag{38}$$

$$= \frac{1}{2} \sum_{n \neq m} \left( \prod_{k \neq m,n} \lambda_k \right) \lambda_m(\lambda_m - 1)\lambda_n(\lambda_n - 1). \tag{39}$$

As a result, the number of 3-dichotomies of grid-like inputs is given by

$$\mathcal{N}_3 = \mathcal{N}_3^a + \mathcal{N}_3^b = \frac{1}{2} \prod_m \lambda_m \left( \sum_n (\lambda_n - 1) \left( \frac{\lambda_n - 2}{3} + \sum_{k \neq n} \lambda_k - n + 1 \right) \right). \tag{40}$$

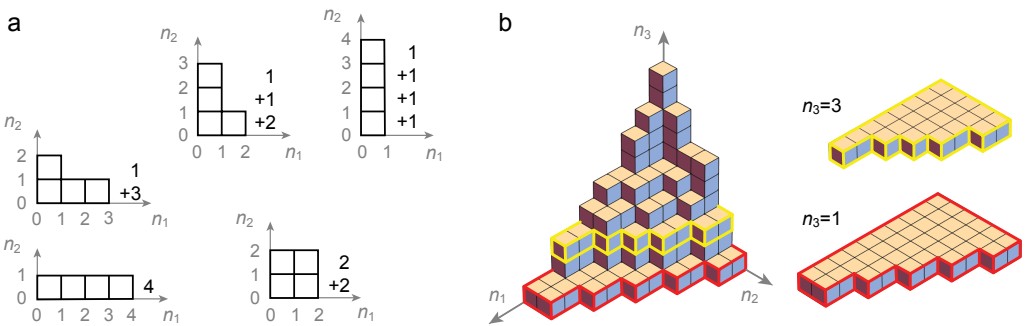

**Appendix 2—figure 1.** Multidimensional Young diagrams. a. Lattice representations of the 2-dimensional Young diagrams of size 4, depicting the integer partitions of 4. b. Lattice representation of a 3-dimensional Young diagram with two 2-dimensional Young diagrams defined as horizontal restrictions.

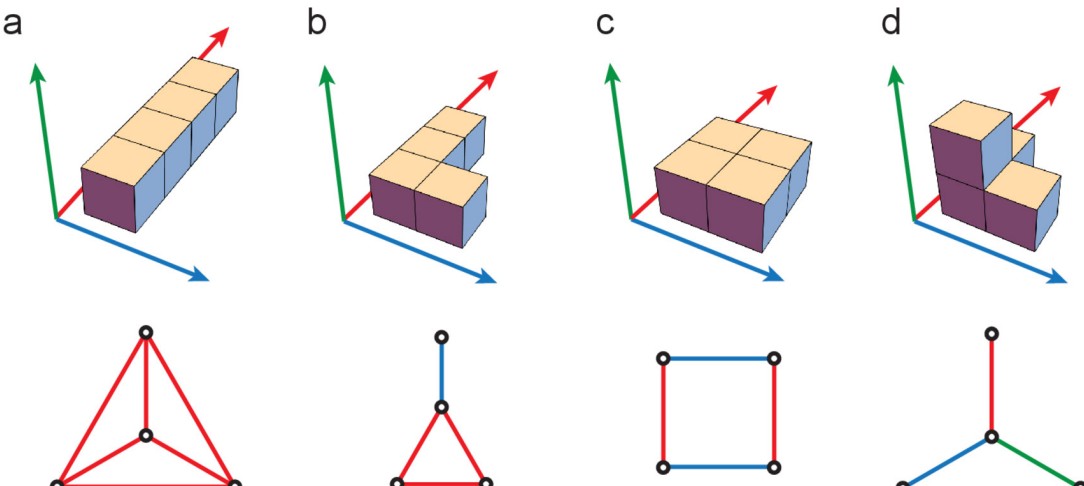

**Appendix 2—figure 2.** Linearly separable 4-dichotomies. Top: there are four possible Young diagrams a, b, c, and d, of size 4, spanning at most three dimensions. Lattice points lying along the mth dimension represent grid-like inputs in $\mathcal{C}_\lambda$ whose coordinates only differ in the mth module. Bottom: Graphical edge structure arising from embedding a Young diagram within $H(\mathcal{C}_\lambda)$, the convex polytope defined by grid-like inputs.

A similar analysis reveals that there are four types of Young diagrams of size 4, which span up to three dimensions if $M \leq 3$. These Young diagrams, denoted by (4a), (4b), (4c), and (4d), are represented in *Figure 6*, where degeneracy indices can be read graphically. As a result, the number of 4-dichotomies of grid-like inputs is given by $\mathcal{N}_4 = \mathcal{N}_4^a + \mathcal{N}_4^b + \mathcal{N}_4^c + \mathcal{N}_4^d$ where the number of type-specific dichotomies is given by

$$\mathcal{N}_{4a} = \frac{1}{24} \sum_{m=1}^{M} \left( \prod_{n \neq m} \lambda_n \right) \lambda_m (\lambda_m - 1)(\lambda_m - 2)(\lambda_m - 3), \tag{41}$$

$$\mathcal{N}_{4b} = \frac{1}{2} \sum_{1 \leq m \leq M} \left( \prod_{k \neq m,n} \lambda_k \right) \lambda_m (\lambda_m - 1)(\lambda_m - 2)\lambda_n(\lambda_n - 1), \tag{42}$$

$$\mathcal{N}_{4c} = \frac{1}{4} \sum_{1 \leq m \leq M} \left( \prod_{k \neq m,n} \lambda_k \right) \lambda_m (\lambda_m - 1)\lambda_n(\lambda_n - 1), \tag{43}$$

$$\mathcal{N}_{4d} = \sum_{1 \leq m \leq M} \left( \prod_{l \neq m,n,k} \lambda_l \right) \lambda_m (\lambda_m - 1)\lambda_n(\lambda_n - 1)\lambda_k(\lambda_k - 1). \tag{44}$$

The classification of dichotomies via Young diagrams also illuminates the geometrical structure of linearly separable $k$-dichotomies, at least for small $k$. In particular, 2-dichotomies are linearly separable if they involve two lattice points forming an edge of the convex polytope, that is, if these points correspond to patterns in $\mathcal{C}_\lambda$ whose coordinates only differ in one module. Similarly, 3-dichotomies are linearly separable if and only if (3a) they involve three lattice points representing patterns in $\mathcal{C}_\lambda$ whose coordinates only differ in one module or (3b) they involve two pairs of lattice points representing patterns in $\mathcal{C}_\lambda$ whose coordinates only differ in one module. Thus, (3a) corresponds to the case of three lattice points specifying a clique of convex-polytope edges, while (3b) corresponds to the case of three lattice points specifying two convex-polytope edges. We illustrate the four geometrical structures of the linearly separable 4-dichotomies in *Figure 6*.

## Numbers of dichotomies for two modules

For two modules of period $\lambda_1$ and $\lambda_2$, recall that each grid pattern in $\mathcal{C}_\lambda$ is a $(\lambda_1 + \lambda_2)$-dimensional vector, which is entirely specified by the indices of its two active neurons: $(i,j)$, $1 \leq i \leq \lambda_1$, $1 \leq j \leq \lambda_2$. Thus, it is convenient to consider a set of grid patterns as a collection of points in the discrete lattice $\{1, \ldots, \lambda_1\} \times \{1, \ldots, \lambda_2\}$. From Proposition 4, we know that linearly separable dichotomies are made of those sets of grid patterns $\mathcal{C}_\lambda$ for which a Young diagram can be formed via permutations of

rows and columns in the lattice (see **Figure 7**). By convention, we consider that the marked lattice points forming a Young diagram define the set of active grid patterns. The remaining unmarked lattice points define the set of inactive grid patterns. To each 2D Young diagrams in the lattice $\{1, \ldots, \lambda_1\} \times \{1, \ldots, \lambda_2\}$ corresponds a class of linearly separable dichotomies. Counting the total number of linearly separable dichotomies when $M = 2$ will proceed in two steps: (i) we first give a slightly stronger result than Proposition about the cardinality of the classes of dichotomies associated to a Young diagram, and (ii) we evaluate the total number of dichotomies by summing class cardinalities over the set of Young diagrams.

## Proposition 7

For two integer periods $\lambda_1$ and $\lambda_2$, let us consider a Young diagram $\mathcal{D}$ in the lattice $\{1, \ldots, \lambda_1\} \times \{1, \ldots, \lambda_2\}$. Without loss of generality, $\mathcal{D}$ can be specified via the degeneracy indices $\sigma_{1,1}, \ldots \sigma_{1,k}$, and $\sigma_{2,1}, \ldots \sigma_{2,k}$, chosen such that

$$\mathcal{D} \text{ has } \sigma_{1,i} \text{ rows of length } \sum_{j=1}^{k+1-i} \sigma_{2,j} \iff \mathcal{D} \text{ has } \sigma_{2,j} \text{ columns of length } \sum_{i=1}^{k+1-i} \sigma_{1,i}. \tag{45}$$

Then, the class of linearly separable dichotomies with Young diagram $\mathcal{D}$, denoted by $C(\mathcal{D})$, has cardinality

$$|C(\mathcal{D})| = \frac{\lambda_1!}{\sigma_{1,1}! \ldots \sigma_{1,k+1}!} \frac{\lambda_2!}{\sigma_{2,1}! \ldots \sigma_{2,k+1}!}, \tag{46}$$

where we have $\sigma_{1,1} + \ldots + \sigma_{1,k+1} = \lambda_1$ and $\sigma_{2,1} + \ldots + \sigma_{2,k+1} = \lambda_2$.

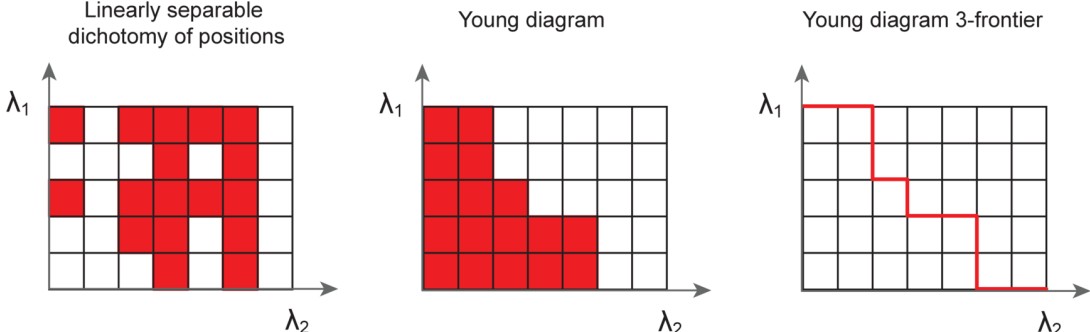

**Appendix 2—figure 3.** Counting 2-module Young diagram. Linearly separable dichotomies (left panel) can be associated to a unique Young diagram (middle panel). These Young diagrams are entirely specified by their frontier path, separating active positions from inactive ones. Enumerating all possible frontier paths allows one to count all the linearly separable dichotomies for two modules.

Proof. Consider a Young diagram $\mathcal{D}$ in $\{1, \ldots, \lambda_1\} \times \{1, \ldots, \lambda_2\}$ with $p$ inactive patterns. The diagram $\mathcal{D}$ is uniquely defined by the row partition $p = r_1 + \ldots + r_{\lambda_1}$, $r_1 \geq \ldots \geq r_{\lambda_1}$, where $r_i$ denotes the occupancy of row , or equivalently by the column partition $p = s_1 + \ldots + s_{\lambda_2}$, $s_1 \geq \ldots \geq s_{\lambda_2}$, where $s_j$ denotes the occupancy of column $j$. The occupancies $\{r_1, \ldots, r_{\lambda_1}\}$ and $\{s_1, \ldots, s_{\lambda_2}\}$ entirely define restrictions along each dimension and each set of occupancies along a dimension is invariant to row and column permutations. The corresponding degeneracy indices can be determined straightforwardly by counting the number of rows or columns with a given occupancy, that is, within a given equivalence class. Denoting the necessarily identical number of rows classes and columns classes by $k \leq \min(\lambda_1, \lambda_2)$, Proposition yields directly the announced result.

## Proposition 8

For two integer periods $\lambda_1$ and $\lambda_2$, the number of linearly separable dichotomies in $\mathcal{C}_{(\lambda_1, \lambda_2)}$ is

$$\mathcal{N}_{\lambda_1, \lambda_2} = \sum_{k=0}^{\min(\lambda_1, \lambda_2)} (k!)^2 S(\lambda_1 + 1, k + 1) S(\lambda_2 + 1, k + 1) = B_{\lambda_2}^{(-\lambda_1)}, \tag{47}$$

where $S(n, k)$ denotes the Stirling numbers of the second kind and where $B_k^{(n)}$ denotes the poly-Bernoulli numbers.

Proof. Our goal is to evaluate the total number of dichotomies $\mathcal{N}_{\lambda_1,\lambda_2}$. To achieve this goal, we will exploit the combinatorics of 2D Young diagrams to specify $\mathcal{N}_{\lambda_1,\lambda_2}$ as

$$\mathcal{N}_{\lambda_1,\lambda_2} = \sum_{\mathcal{D} \subset \{1,\ldots,\lambda_1\} \times \{1,\ldots,\lambda_2\}} |C(\mathcal{D})|, \tag{48}$$

where $\mathcal{D}$ runs over all possible Young diagrams. Because of the multinomial nature of the cardinalities $|C(\mathcal{D})|$, it is advantageous to adopt an alternative representation for Young diagrams. This alternative representation will require utilizing the frontier of a Young diagram. Given a Young diagram $\mathcal{D}$ with $k$ distinct nonempty rows and $k$ distinct nonempty columns, we define its frontier as the path joining the lattice points $(0, \lambda_2)$ and $(\lambda_1, 0)$, via lattice positions in $\mathcal{D}$ separating the active region from the inactive region (see **Figure 7**). Such a path is uniquely defined via $k+1$ downward steps of size $\sigma_{1,k+1}, \ldots, \sigma_{1,1}$ and $k+1$ rightward steps of sizes $\sigma_{2,1}, \ldots, \sigma_{2,k+1}$, which satisfy $\sigma_{1,1} + \ldots + \sigma_{1,k+1} = \lambda_1$ and $\sigma_{2,1} + \ldots + \sigma_{2,k+1} = \lambda_2$. Clearly, the frontier of $\mathcal{D}$ determines the cardinality of $C(\mathcal{D})$ via (46). To evaluate $\mathcal{N}_{\lambda_1,\lambda_2}$ in (48), we partition Young diagrams based on $k$, the number of distinct row and column sizes. For $k = 0$, we have $\sigma_{1,1} = \lambda_1$ and $\sigma_{2,1} = \lambda_2$, corresponding to $N_{\lambda_1,\lambda_2}(0) = 1$ Young diagram, the empty diagram, where all patterns are inactive. For $k = 1$, there is a single row and column size, corresponding to Young diagrams where the active patterns are arranged in a rectangle, with edge lengths $\sigma_{1,1}$ and $\sigma_{2,1}$. Nonempty rectangular diagrams correspond to $\sigma_{1,1} > 0$ and $\sigma_{2,1} > 0$, and thus contribute

$$N_{\lambda_1,\lambda_2}(1) = \sum_{\rho_1=1}^{\lambda_1} \sum_{\sigma_1=1}^{\lambda_2} \frac{\lambda_1!}{\sigma_{1,1}!(\lambda_1 - \sigma_{1,1})!} \frac{\lambda_2!}{\sigma_{2,1}!(\lambda_2 - \sigma_{2,1})!} \tag{49}$$

$$= \left( \sum_{\rho_1=0}^{\lambda_1} \frac{\lambda_1!}{\sigma_{1,1}!(\lambda_1 - \sigma_{1,1})!} - 1 \right) \left( \sum_{\sigma_1=0}^{\lambda_2} \frac{\lambda_2!}{\sigma_{2,1}!(\lambda_2 - \sigma_{2,1})!} - 1 \right) = (2^{\lambda_1} - 1)(2^{\lambda_2} - 1), \tag{50}$$

to the sum (48). The contribution of diagrams with general $k$-frontier, denoted by $N_{\lambda_1,\lambda_2}(k)$, follows from the multinomial theorem, where one ensures that frontiers with less than $k+1$ downward and rightward steps do not get repeated. These $k$-frontiers correspond to $k+1$ sequences of downward and rightward steps for which no step has zero size, except possibly for the first downward step emanating from $(0, \lambda_2)$ and the last rightward step arriving at $(\lambda_1, 0)$. Under these conditions, the downward and rightward steps can be chosen independently, so that we can write $N_{\lambda_1,\lambda_2}(k) = f_k(\lambda_1)f_k(\lambda_2)$, where the factors $f_k(\lambda_1)$ and $f_k(\lambda_2)$ only depend on the downward steps and rightward steps, respectively. Let us focus on the downward steps alone, that is, on the term $f_k(\lambda_1)$. The admissible sequences of steps satisfy $\sigma_{1,1} + \ldots + \sigma_{1,k+1} = \lambda_1$, with $\sigma_{1,1}, \ldots, \sigma_{1,k} \neq 0$. From the multinomial theorem, we have

$$(k+1)^{\lambda_1} = \sum_{\substack{\sigma_{1,1}+\ldots+\sigma_{1,k+1}=\lambda_1 \\ \sigma_{1,1}\ldots\sigma_{1,k}\neq 0}} \frac{\lambda_1!}{\sigma_{1,1}!\ldots\sigma_{1,k+1}!} + \sum_{\substack{\sigma_{1,1}+\ldots+\sigma_{1,k+1}=\lambda_1 \\ \sigma_{1,1}\ldots\sigma_{1,k}=0}} \frac{\lambda_1!}{\sigma_{1,1}!\ldots\sigma_{1,k+1}!}, \tag{51}$$

where the first term of the right-hand side is $f_k(\lambda_1)$ and the second term of the right-hand side collects the contribution of sequences that are not $k$-frontiers. The latter term can be evaluated explicitly via the exclusion-inclusion principle yielding

$$\sum_{\substack{\sigma_{1,1}+\ldots+\sigma_{1,k+1}=\lambda_1 \\ \sigma_{1,1}\ldots\sigma_{1,k}=0}} \frac{\lambda_1!}{\sigma_{1,1}!\ldots\sigma_{1,k+1}!} = \sum_{i=1}^{k} (-1)^{i-1} \binom{k}{i} \sum_{\substack{\sigma_{1,1}+\ldots+\sigma_{1,k+1}=\lambda_1 \\ \sigma_{1,1}=0,\ldots,\sigma_{1,i}=0}} \frac{\lambda_1!}{\sigma_{1,1}!\ldots\sigma_{1,k+1}!}, \tag{52}$$

$$= \sum_{i=1}^{k} (-1)^{i-1} \binom{k}{i} \sum_{\sigma_{1,i+1}+\ldots+\sigma_{1,k+1}=\lambda_1} \frac{\lambda_1!}{\sigma_{1,i+1}!\ldots\sigma_{1,k+1}!} \tag{53}$$

$$= \sum_{i=1}^{k}(-1)^{i-1}\binom{k}{i}(k+1-i)^{\lambda_1}, \tag{54}$$

where we have used the multinomial theorem for the last equality. Together with (51), the above equation allows one to specify $f_k(\lambda)$ in terms of the Sterling numbers of the second kind, denoted by $S(n,k)$, as

$$f_k(\lambda) = \sum_{i=0}^{k}(-1)^i\binom{k}{i}(k+1-i)^{\lambda}, \tag{55}$$

$$= \sum_{i=0}^{k}(-1)^i\binom{k}{i}\sum_{j=0}^{\lambda}\binom{\lambda}{j}(k-i)^j, \tag{56}$$

$$= \sum_{j=0}^{\lambda}\binom{\lambda}{j}\sum_{i=0}^{k}(-1)^i\binom{k}{i}(k-i)^j, \tag{57}$$

$$= k!\sum_{j=0}^{\lambda}\binom{\lambda}{j}S(j,k), \tag{58}$$

$$= k!S(\lambda+1,k+1), \tag{59}$$

where the last equality follows from a well-known identity about Stirling numbers of the second kind. Then, the overall number of dichotomies follows from the fact that the frontier has at most $\min(\lambda_1,\lambda_2)$ distinct values of row/column sizes, which implies

$$\mathcal{N}_{\lambda_1,\lambda_2} = \sum_{k=0}^{\min(\lambda_1,\lambda_2)} N_{\lambda_1,\lambda_2}(k) = \sum_{k=0}^{\min(\lambda_1,\lambda_2)} (k!)^2 S(\lambda_1+1,k+1)S(\lambda_2+1,k+1) = B_{\lambda_2}^{(-\lambda_1)}. \tag{60}$$

where we have recognized the definition of the poly-Bernoulli numbers $B_k^{(n)}$. These numbers are defined via the generating function

$$\frac{\mathrm{Li}_k\left(1-e^{-x}\right)}{1-e^{-x}} = \sum_{n=0}^{\infty} B_k^{(n)}\frac{x^n}{n!}, \tag{61}$$

where $\mathrm{Li}_k$ denotes the poly-logarithm.

Poly-Bernoulli numbers were originally introduced by Kaneko to enumerate the set of binary $k$-by-$n$ matrices that are uniquely reconstructible from their row and column sums (***Kaneko, 1997***). The use of poly-Bernoulli numbers to enumerate permutations of Young tableaux was pioneered by Postnikov while investigating totally Grassmannian cells (***Postnikov, 2006***). While studying the asymptotics of the extremal excedance set statistic, ***de Andrade et al., 2015*** obtained the asymptotics of the poly-Bernoulli numbers along the diagonal:

$$\mathcal{N}_{\lambda,\lambda} = B_\lambda^{(-\lambda)} = \left(\frac{1}{\log 2\sqrt{1-\log 2}} + o(1)\right)\frac{(2\lambda)!}{(2\log 2)^{2\lambda}}. \tag{62}$$

## Appendix 3

### Spatial embedding of the grid code

In this Appendix, we address the limitations entailed by spatially embedding grid-like inputs. First, we define the grid-cell-activity matrix that specifies the spatial assignment of grid-like inputs for 1D space. Second, we show that the contiguous-separating capacity, defined as the maximum spatial extent over which all possible dichotomies are linearly separable, is determined by the rank of the grid-cell-activity matrix. Third, we generalize our results about the separating capacity to spaces of arbitrary dimensions.

### Grid-cell-activity matrix for 1D space

The fundamental object of our combinatorial analysis is the polytope whose vertices have all possible grid-cell patterns as coordinates. Thanks to the many symmetries of this polytope, we can enumerate linearly separable dichotomies of grid-like inputs. However, such an approach makes no explicit reference the actual physical space that these grid-like inputs encode. Making these reference consists in specifying a mapping between spatial positions and grid-like inputs. Unfortunately, this generally involves breaking many of the polytope symmetries, precluding any combinatorial analysis. It is especially true if one considers spaces encoded by a subset of grid-cell patterns, as opposed to the full set $\mathcal{C}_\lambda$, a situation that leads to considering nonsymmetrical polytopes.

Let us explain this point by considering the case of a discrete 1D space where each position is marked by an integer in $\mathbb{Z}$. In this setting, positional information about $\mathbb{Z}$ is encoded by $M$ modules of grid cells with integer periods $\boldsymbol{\lambda} = (\lambda_1, \ldots, \lambda_M)$. Recall that each module comprises $\lambda_m$ cells, each active at a distinct phase within the period $\lambda_m$, and that the corresponding repertoire of grid-like inputs $\mathcal{C}_\lambda$ has cardinality $\Lambda = \prod_{m=1}^{M} \lambda_m$. Because the spatial activity of grid cells is periodic and because we consider a finite number of grid cells, the mappings between spatial positions and grid-like inputs are necessarily periodic functions $A_\lambda : \mathbb{Z} \to \mathcal{C}_\lambda$. Let us denote by $L$ the period of $A_\lambda$. It is then convenient to consider the functions $A_\lambda : \mathbb{Z}/L\mathbb{Z} \to \mathcal{C}_\lambda$ as matrices, called grid-cell-activity matrices, whose jth column is the pattern in $\mathcal{C}_\lambda$ that encodes the jth spatial position in $\{1, \ldots, L\}$, seen as the element $j$ in $\mathbb{Z}/L\mathbb{Z}$. In particular, the matrices $A_\lambda$ have $N = \sum_{m=1}^{M} \lambda_m$ rows, each row corresponding to the periodic activity of a grid cell. Moreover, at every position $j$, $1 \leq j \leq L$, each module has a single active cell. For the sake of clarity, here follows a concrete example of grid-cell-activity matrix for $\boldsymbol{\lambda} = (2, 3, 5)$:

$$
A_{(2,3,5)} = \left(
\begin{array}{cccccccccccccccc}
1 & 0 & 1 & 0 & 1 & 0 & 1 & 0 & 1 & 0 & \ldots & \ldots & 0 & 1 & 0 & 1 & 0 \\
0 & 1 & 0 & 1 & 0 & 1 & 0 & 1 & 0 & 1 & \ldots & \ldots & 1 & 0 & 1 & 0 & 1 \\
\hline
1 & 0 & 0 & 1 & 0 & 0 & 1 & 0 & 0 & 1 & \ldots & \ldots & 0 & 0 & 1 & 0 & 0 \\
0 & 1 & 0 & 0 & 1 & 0 & 0 & 1 & 0 & 0 & \ldots & \ldots & 1 & 0 & 0 & 1 & 0 \\
0 & 0 & 1 & 0 & 0 & 1 & 0 & 0 & 1 & 0 & \ldots & \ldots & 0 & 1 & 0 & 0 & 1 \\
\hline
1 & 0 & 0 & 0 & 0 & 1 & 0 & 0 & 0 & 0 & \ldots & \ldots & 1 & 0 & 0 & 0 & 0 \\
0 & 1 & 0 & 0 & 0 & 0 & 1 & 0 & 0 & 0 & \ldots & \ldots & 0 & 1 & 0 & 0 & 0 \\
0 & 0 & 1 & 0 & 0 & 0 & 0 & 1 & 0 & 0 & \ldots & \ldots & 0 & 0 & 1 & 0 & 0 \\
0 & 0 & 0 & 1 & 0 & 0 & 0 & 0 & 1 & 0 & \ldots & \ldots & 0 & 0 & 0 & 1 & 0 \\
0 & 0 & 0 & 0 & 1 & 0 & 0 & 0 & 0 & 1 & \ldots & \ldots & 0 & 0 & 0 & 0 & 1 \\
\end{array}
\right)
\tag{63}
$$

As the labelling of grid cells is arbitrary within a module, grid-population activity is actually represented by a class of matrices, which is invariant to permutation of the grid cells $(m, i)$, $1 \leq i \leq \lambda_m$, within a module $m$. Here, with no loss of generality, we consider the class representatives obtained by ordering the grid cells by increasing phase within each module. This convention allows us to simply define the activity matrix $A_\lambda$ via the introduction of a spatial shift operator $J_\lambda$. We define the shift operator $J_\lambda$ as the linear operator that cyclically increments the phases by one unit within each module, that is,

$$J_{\pmb{\lambda}} = \begin{pmatrix} J_{\lambda_1} & & & \\ & J_{\lambda_2} & & \\ & & \ddots & \\ & & & J_{\lambda_M} \end{pmatrix} \quad \text{with} \quad J_{\lambda_m} = \begin{pmatrix} 0 & 1 & 0 & 0 & \dots \\ 0 & 0 & 1 & 0 & \dots \\ \vdots & & \ddots & \ddots & \ddots \\ 1 & 0 & \dots & & \end{pmatrix}, \tag{64}$$

where $J_{\lambda_m}$ is the canonical circulant permutation matrix of order $\lambda_m$. We refer to $J_{\pmb{\lambda}}$ as a shift operator because its action on any vector of $A_{\pmb{\lambda}}$ corresponds to a positional shift by one unit of space: if $\pmb{c}_j$, $1 \le j \le L$, denotes the jth column of $A_{\pmb{\lambda}}$, then $J_{\pmb{\lambda}}\pmb{c}_j = \pmb{c}_{j+1}$ if $j < L$, and $J_{\pmb{\lambda}}\pmb{c}_L = \pmb{c}_1$. Thus, we can define the grid-cell-activity matrix as the matrix obtained by enumerating in order the grid-cell patterns $J_{\pmb{\lambda}}^k \pmb{c}_1$, $k \in \mathbb{N}$, up to redundancies. Such a definition of the grid-cell-activity matrix prominently features the relation between the symmetries of the grid code and those of the actual physical space. In particular, it clearly shows that the formulation of our problem is invariant to rotation of the discretized space $1, 2, \dots, L$, that is, by shift in $\mathbb{Z}/L\mathbb{Z}$. We show that grid-cell-activity matrix can be similarly defined for lattice space of higher dimensions in Section C.3, including the relevant case of the 2D hexagonal lattice.

A key observation is that the periodicity $L$, that is, the number of positions univocally tagged by grid-like inputs, is directly related to the periods $\pmb{\lambda}$ via the Chinese remainder theorem. Indeed, by the Chinese remainder theorem, the first redundant grid-like input occurs for $L = \mathrm{lcm}\,(\lambda_1, \dots, \lambda_M)$, therefore specifying the number of columns of the activity matrix. Thus, for pairwise coprime periods $\lambda_m$, $1 \le m \le M$, we have $L = \Lambda$ and the columns of the activity matrix $A_{\pmb{\lambda}}$ exhaustively enumerate all grid-like inputs in $\mathcal{C}_{\pmb{\lambda}}$. As a result, all the combinatorial results obtained for the full set of patterns $\pmb{C}_{\pmb{\lambda}}$ directly apply over the full linear space $\{1, \dots, L\}$ for pairwise coprime periods. In particular, for pairwise coprime periods, we have $\mathrm{rank}\,A_{\pmb{\lambda}} = \sum_{i=1}^{M} \lambda_i - M + 1$ by Proposition 1.

Unfortunately, our combinatorial results do not directly extend to a spatial context for integer periods that are not pairwise coprime or for incomplete spaces $\{1, \dots, L'\}$, $L' < L$. For non-coprime periods, we have $L < \Lambda$, as exemplified by the grid-cell-activity matrix for $\pmb{\lambda} = (2, 8)$ given by

$$A_{(2,8)} = \left( \begin{array}{cccc} 1 & 0 & 1 & 0 \\ 0 & 1 & 0 & 1 \\ \hline 1 & 0 & 0 & 0 \\ 0 & 1 & 0 & 0 \\ 0 & 0 & 1 & 0 \\ 0 & 0 & 0 & 1 \end{array} \right) \tag{65}$$

which comprises only four of the eight patterns of $\mathcal{C}_{2,4}$. Independent of the coprimality of the periods, the grid-cell-activity matrix over incomplete spaces is simply obtained by deleting the columns corresponding to the missing positions. In particular, we clearly have $L' < L \le \Lambda$. Excluding some grid-like inputs has two opposite implications: (i) the total number of dichotomies is reduced in keeping with considering a smaller space but (ii) some dichotomies that were previously not linearly separable can become realizable. Disentangling these opposite implications is obscured by the many broken symmetries of the polytope formed by the subset patterns under consideration. For this reason, we essentially resort to studying spatial embedding of the grid code numerically. Such numerical analysis reveals, perhaps not surprisingly, that a key role is played by the embedding dimension of the grid code, especially in relation to the concept of contiguous-separating capacity.

## Contiguous-separating capacity

We define the contiguous-separating capacity of a grid code as the maximum physical extent over which all possible dichotomies are linearly separable. Classically, for $N$-dimensional inputs in general position, the separating capacity is defined as the maximum number of patterns for which all possible dichotomies are linearly separable, without any reference to contiguity. Within this context, Cover's counting function theorem implies that the separating capacity equals the dimension of the input space. Should the grid-like inputs be in general position in the input space, the separating capacity would thus be equal to $\mathrm{rank}\,A_{\pmb{\lambda}}$. However, being in general position requires that any submatrix formed by $r$ columns of $A_{\pmb{\lambda}}$ be of rank $r$ for $r \le \mathrm{rank}\,A_{\pmb{\lambda}}$. This property does not hold for grid-cell-

activity matrices. Moreover, we are interested in a stronger notion of separating capacity as we require that the grid-like inputs achieving separating capacity represent contiguous spatial position. Thankfully, the spatial symmetry of the grid-cell-activity matrices allows us to show that even under these restrictions the separating capacity is indeed $\mathrm{rank}\, A_{\lambda}$.

## Proposition 9

The contiguous-separating capacity of the generic grid-cell-activity matrix $A_{\lambda}$ is equal to $\mathrm{rank}\, A_{\lambda}$.

Proof.

The proof proceeds in two steps. With no loss of generality, we only consider linear classification via perceptron with zero threshold.

(i) By permutation and shift invariance, it is enough to consider contiguous columns of $A_{\lambda}$ starting from the first column $c_1$. From the definition of $A_{\lambda}$, the $k$ contiguous columns can be generated in terms of the shift operator $J_{\lambda}$ as the sequence: $c_1, J_{\lambda}c_1, \ldots, J_{\lambda}^k c_1$. Let us consider the sequence $\{d_k\}_{k \in \mathbb{N}}$ defined by $d_k = \dim\{c_1, J_{\lambda}c_1, \ldots, J_{\lambda}^k c_1\}$. Posit $r = \mathrm{rank}\, A_{\lambda}$. If there is an integer $n$ such that $d_n = d_{n+1}$, then necessarily $d_k$ is constant for $k \geq n$, and is equal to $\lim_{k \to \infty} d_k = d_L = r$. As $d_1 = 1$ and $d_{k+1} - d_k \in \{0, 1\}$, the preceding observation implies that $d_{k+1} - d_k = 1$ for $1 \leq k < \mathrm{rank}\, A_{\lambda}$. This shows that the contiguous columns $c_i$, $1 \leq i \leq r$, are linearly independent, and thus are in general position in the input space. By Cover's counting function theorem, all dichotomies obtained by labeling the positions $1 \leq i \leq r$ with $r = \mathrm{rank}\, A_{\lambda}$ are linearly separable.

(ii) Considering an extra position, that is, including the column $c_{r+1}$, produces at least a dichotomy that is not linearly separable. We proceed by contradiction. Assume that all dichotomies of the $r + 1$ positions, that is, of the columns $c_i$ with $1 \leq i \leq r + 1$, are linearly separable. By Cover's counting function theorem, this is equivalent to assuming that all dichotomies of the first $r$ positions, that is, of the columns $c_i$ with $1 \leq i \leq r$, can be achieved by an $(r-1)$-dimensional hyperplane passing through $c_{r+1}$. In other words, for all $r$-dichotomies $y$ in $\{0, 1\}^r$, there is a weight vector $w$ such that $y_i(w^T c_i) > 0$ for $1 \leq i \leq r$ and such that $w^T c_{r+1} = 0$. However, by linear dependence, there are nonzero coefficients $a_i$ such that $c_{r+1} = \sum_{i=1}^{r} a_i c_i$, so that for any $r$-dichotomy, we can find $w$ achieving that dichotomy and such that

$$\sum_{i=1}^{r} a_i (w^T c_i) = 0. \tag{66}$$

Considering a dichotomy for which $y_i = a_i/|a_i|$ for nonzero coefficients yields

$$\sum_{i=1}^{r} a_i (w^T c_i) = \sum_{i=1}^{r} |a_i||w^T c_i| > 0. \tag{67}$$

which is a contradiction with (66).

The above proposition specifies $\mathrm{rank}\, A_{\lambda}$ as the contiguous-separating capacity for 1D spatial model. This rank also specifies the dimension of the space containing the subset of grid-like inputs to be linearly classified. For pairwise coprime periods $\lambda$, Proposition 1 shows that $\mathrm{rank}\, A_{\lambda} = \sum_{m=1}^{M} \lambda_m$. The following proposition generalizes this result to generic integer periods.

## Proposition 10

Let $A_{\lambda}$ denote the grid-cell-activity matrix specified by $M$ grid modules with integer periods $\lambda = (\lambda_1, \ldots, \lambda_M)$. The rank of the activity matrix $A_{\lambda}$ is given by

$$\mathrm{rank}\, A_{\lambda} = \sum_{i=1}^{M} \lambda_i - \sum_{1 \leq i < j \leq M} \gcd(\lambda_i, \lambda_j) + \sum_{1 \leq i < j < k \leq M} \gcd(\lambda_i, \lambda_j, \lambda_k) - \cdots + (-1)^{M-1} \gcd(\lambda_1, \ldots, \lambda_M) =$$

$$\sum_{k=1}^{M} (-1)^{k-1} \sum_{S \subset \lambda, |S|=k} \gcd(S) \tag{68}$$

where $S$ is a subset of integer periods and $|S|$ denotes the cardinality of the set $S$. If the periods are pairwise coprime, the above formula yields $\mathrm{rank}\, A_{\lambda} = \sum_{i=1}^{M} \lambda_i - M + 1$.

Proof. The proof will proceed in three steps.

(i) The first step is to realize that $\operatorname{rank} A_\lambda = \operatorname{rank} A_\lambda^T = \dim(V_1 + \ldots + V_M)$, where the vector spaces $V_m$, $1 \leq m \leq M$, are generated by the rows of the mth module of the activity matrix. Then, the exclusion-inclusion principle applied to the sum of $V_1 + \ldots + V_M$ yields an expression for $A_\lambda$ as the alternated sum:

$$\operatorname{rank} A_\lambda = \dim\left(V_1 + \ldots + V_M\right) \tag{69}$$

$$= \sum_{i=1}^{M} \dim V_i - \sum_{1 \leq i < j \leq M} \dim V_i \cap V_j + \sum_{1 \leq i < j < k \leq M} \dim V_i \cap V_j \cap V_k - \ldots . \tag{70}$$

By definition of the activity matrix, the space $V_m$ is generated by $\lambda_m$ row vectors, which are cyclically permuted versions of the $\lambda_m$-periodic vector $\boldsymbol{r}_{\lambda_m} = (1, 0, \ldots, 0|1, 0, \ldots, 0|1, 0, \ldots)$. In particular, these $\lambda_m$ row vectors can be enumerated by iterated application of $J$, the canonical $L$-dimensional circulant permutation operator. The resulting sequence $\boldsymbol{r}_{\lambda_m}, J\boldsymbol{r}, \ldots, J^{\lambda_m - 1}\boldsymbol{r}_{\lambda_m}$ actually forms a basis of $V_m$, identified to the space of $\lambda_m$-periodic vectors of length $L$, and thus $\dim V_m = \lambda_m$. The announced formula will follow from evaluation of the dimension of the intersection of the vector spaces $V_m$.

(ii) The second step is to observe that one can specify the set of spaces $V_m$, $1 \leq m \leq M$, as the span of vectors chosen from a common basis of $\mathbb{R}^L$, where we recall that $L = \operatorname{lcm}(\lambda_1, \ldots, \lambda_M)$. We identify such a common basis by considering the action of the operator $J$ on $L$-dimensional periodic vectors. As a circulant permutation operator, $J$ admits a diagonal matrix representation in the basis of eigenvectors $\{e_i\}$, $1 \leq i \leq L$,

$$\boldsymbol{e}_j = \left(1, \omega_j, \omega_j^2, \ldots, \omega_j^{L-1}\right), \tag{71}$$

associated to the eigenvalue $\omega_j = e^{i\frac{2\pi j}{L}}$, where $i^2 = -1$. Moreover, $J$ clearly preserves periodicity when acting on row vectors in $\mathbb{R}^L$, so that the spaces $V_m$, $1 \leq m \leq M$, are stable by $J$. As a consequence, each space $V_m$ can be represented as the span of a subset of the eigenvectors of $J$. In principle, the existence of a basis spanning the spaces $V_m$, $1 \leq m \leq M$, allows one to compute the dimension of the intersections of these spaces by counting the number of common basis elements in their span.

(iii) The last step is to show that counting the number of common basis elements $\boldsymbol{e}_i$ in the subsets of $\{V_m\}_{1 \leq m \leq M}$ yields the announced formula. Proving this point relies on elementary results from the theory of cyclic groups. Let us first consider the basis elements generating $V_m$, which are the elements $\boldsymbol{e}_j$ that are $\lambda_m$-periodic. These basis elements are precisely those for which $\omega_j^{\lambda_m} = 1$, that is, $\lambda_m j = 0$ in the cyclic group $\mathbb{Z}/L\mathbb{Z}$. Considering the integers $j$ as elements of $\mathbb{Z}/L\mathbb{Z}$, we can then specify the basis vectors generating $V_m$ by invoking the subgroup structure of the cyclic groups. Specifically, the basis elements generating $V_m$ are indexed by the elements of the unique subgroup of order $\lambda_m$ in $\mathbb{Z}/L\mathbb{Z}$. Thus, as expected, the number of basis elements equates the otherwise known dimension of $V_m$. Let us then consider the basis elements generating the intersection space $V_m \cap V_n$, $m \neq n$, which are the elements $\boldsymbol{e}_j$ that are both $\lambda_m$-periodic and $\lambda_n$-periodic. These basis elements correspond to those indices $j$ for which we have $\lambda_m j = 0$ and $\lambda_n j = 0$ in the cyclic group $\mathbb{Z}/L\mathbb{Z}$, that is, for which $\gcd(\lambda_m, \lambda_n)j = 0$ in $\mathbb{Z}/L\mathbb{Z}$. By the subgroup structure of cyclic groups, the basis elements generating $V_m \cap V_n$ are thus indexed by the elements of the unique subgroup of order $\gcd(\lambda_n, \lambda_m)$ in $\mathbb{Z}/L\mathbb{Z}$. Thus, we have $\dim V_m \cap V_n = \gcd(\lambda_m, \lambda_n)$. The above reasoning generalizes straightforwardly to any set of indices $1 \leq m_1 < \ldots < m_k \leq M$, $1 \leq k \leq m$, leading to

$$\dim V_{m_1} \cap \ldots \cap V_{m_k} = \gcd(\lambda_{m_1}, \ldots, \lambda_{m_k}) \tag{72}$$

Specifying the dimension of the intersection spaces in (69) derived from the exclusion-inclusion principle yields the rank formula given in (68).

## Generalization to higher dimensional lattices

Our two results about (i) the number of dichotomies for grid code with two modules and about (ii) the separating capacity for an arbitrary number of modules generalize to an arbitrary number of dimensions. The generalization of (i) is straightforward as our results bear on the set of grid-like inputs with no reference to physical space. The only caveat has to do with the fact that for $d$-dimensional lattice, each module $m$, $1 \leq m \leq M$, contains $\lambda_m^d$ cells so that $\lambda_m^d$ has to be substituted for $\lambda_m$ in formula (47). It turns out that the generalization of (ii) proceeds in the exact same way, albeit

in a less direct fashion. In the following, we prove that the separating capacity for a $d$-dimensional lattice model, including the 2D hexagonal lattice, is still given by the rank of the corresponding activity matrix.

A couple of remarks are in order before justifying the generalization of (ii):

First, let us specify how to construct activity matrices in d-dimensional space by considering a simple example. Consider the hexagonal-lattice model for two modules with $\boldsymbol{\lambda} = (2, 3)$. As illustrated in *Figure 1*, there are four possible 2-periodic lattices and nine possible 3-periodic lattices, each lattice representing the spatial activity pattern of a grid cell. Combining the encoding of the two modules yield a periodic lattice, with lattice mesh comprising $\mathrm{lcm}\,(\lambda_1, \lambda_2)^2 = 36$ positions. Every position within the mesh size is uniquely labeled by the grid-like input, and any subset of positions with larger cardinality has redundancy. Observe moreover that the lattice mesh is equivalent to that of a (2, 3)-square lattice, and in fact, the activity matrix for an (2, 3)-hexagonal lattice model is the same as that for a (2, 3)-square lattice. As a result, the spatial dependence of the grid-cell population is described by a matrix in $\mathbb{R}^{13 \times 36}$ with the following block structure:

$$
A^{(2)}_{(2,3)} = \left(\begin{array}{cccccc} B_{(2)} & 0 & B_{(2)} & 0 & B_{(2)} & 0 \\ 0 & B_{(2)} & 0 & B_{(2)} & 0 & B_{(2)} \\ \hline B_{(3)} & 0 & 0 & B_{(3)} & 0 & 0 \\ 0 & B_{(3)} & 0 & 0 & B_{(3)} & 0 \\ 0 & 0 & B_{(3)} & 0 & 0 & B_{(3)} \end{array}\right) \quad \text{with} \quad \begin{array}{l} B_{(2)} = \left(\begin{array}{cccccc} 1 & 0 & 1 & 0 & 1 & 0 \\ 0 & 1 & 0 & 1 & 0 & 1 \end{array}\right), \\[2em] B_{(3)} = \left(\begin{array}{cccccc} 1 & 0 & 0 & 1 & 0 & 0 \\ 0 & 1 & 0 & 0 & 1 & 0 \\ 0 & 0 & 1 & 0 & 0 & 1 \end{array}\right). \end{array} \tag{73}
$$

In the above matrix $A^{(2)}_{(2,3)}$, the top-two block rows represent the activity of 2-periodic cells, while the bottom-three block rows represent the activity of 3-periodic cells. By convention, we consider blocks $B_{(2)}$ and $B_{(3)}$, comprising respectively two and three cells, represent the activity of grid cells along the horizontal $x$-axis. There are two rows of blocks $B_{(2)}$ and three rows of blocks $B_{(3)}$ to encode 2-periodicity and 3-periodicity, respectively, along the vertical $y$-axis. It is straightforward to generalize this hierarchical block structure to construct an activity matrix $A^{(d)}_\lambda$ for arbitrary periods $\lambda_m$ and arbitrary square-lattice dimension $d$. In particular, the matrix $A^{(d)}_\lambda$ has $\sum_{m=1}^{M} \lambda_m^d$ rows and $L = \mathrm{lcm}\,(\lambda_1, \dots, \lambda_M)^d$ columns.

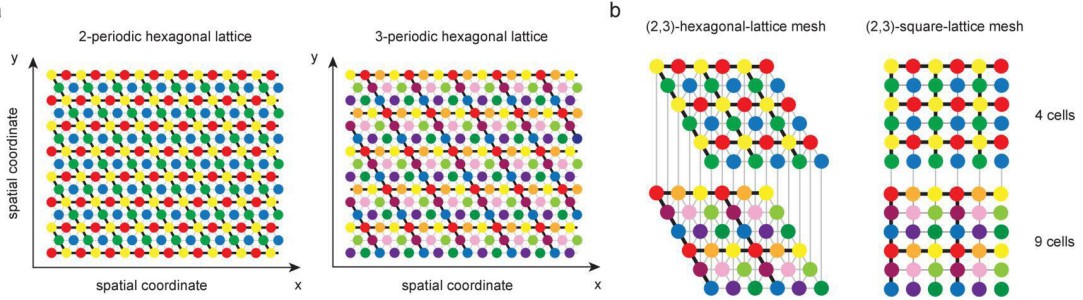

**Appendix 3—figure 1.** Hexagonal and square lattice in two dimensions. (**a**) In two dimensions, 2-periodic and 3-periodic modules comprise respectively four and nine possible grid-cell-activity pattern. For instance, red, green, blue, and yellow patterns in the leftmost lattice correspond to the four possible patterns of activity that a 2-periodic cell can exhibit on an hexagonal lattice. (**b**) The maximum lattice mesh over which each position is uniquely encoded by the grid-like code is given as $6 \times 6 = 2^2 \times 3^2$. Moreover, the hexagonal symmetry plays no role in our capacity calculations and one can consider a square lattice of positions instead.

Second, let us define the notion of contiguous-separating capacity for $d$-dimensional lattice with $d > 1$. In one dimension, we define the contiguous-separating capacity as the maximum spatial extent for which all dichotomies involving its discrete set of positions are linearly separable. We generalize this notion for arbitrary dimensions $d$ by defining the contiguous-separating capacity as the maximum connected component of $d$-dimensional positions for which all dichotomies are possible. Observe that thus-defined, we are rather oblivious about the geometric arrangement of this connected components. This is due to the fact that in dimension $d > 1$, the contiguous-separating capacity can be achieved by many distinct arrangements.

After these preliminary remarks, we can now prove the following proposition.

## Proposition 11

The contiguous-separating capacity of the generic grid-cell-activity matrix $A_\lambda$ is equal to $\text{rank}\,A_\lambda^{(d)}$, where we have

$$\text{rank}\,A_\lambda^{(d)} = \sum_{k=1}^{M}(-1)^{k-1}\sum_{S\subset\lambda,|S|=k}\gcd(S)^d \tag{74}$$

Proof. We only justify the formula for the case $d=2$ as similar arguments apply for all integers $d>1$ (see Remark after this proof). The proof will proceed in two steps: (i) we justify the formula for $\text{rank}\,A_\lambda^{(d)}$ and (ii) we justify that the contiguous-separating capacity equals $\text{rank}\,A_\lambda^{(d)}$.

(i) We follow the same strategy as for dimension 1 to establish the rank formula for $d=2$ via exclusion-inclusion principle. The key point is to exhibit a basis of vectors $(e_1,\ldots,e_L)$ in $\mathbb{R}^{L\times L}$, with $L=\text{lcm}\,(\lambda_1,\ldots,\lambda_M)$, which spans all the vector spaces $V_m$, $1\le m\le M$, where $V_m$ denotes the space of $\lambda_m$ periodic functions on the $(L\times L)$-lattice mesh. To specify such a basis, we consider the two operators $J_x$ and $J_y$ acting on the grid-like inputs and representing the one-unit shift along horizontal $x$-axis and along the vertical $y$-axis, respectively. A basis of the space of $\lambda_m$ periodic functions on the $(L\times L)$-lattice mesh is generated by iterated action of $J_x$ and $J_y$ on the activity lattice of a $\lambda_m$-periodic cell, that is, on a $\{0,1\}$-row vector $r_{\lambda_m}$ of the $m$th module of $\text{rank}\,A_\lambda^{(d)}$. Specifically, a basis of $V_m$ is given by the $\lambda_m^2$ vectors $J_x^k J_y^l r_{\lambda_m}$, with $0\le k<\lambda_m$ and $0\le l<\lambda_m$. Moreover, the operators $J_x$ and $J_y$ commute on $\mathbb{R}^{L\times L}$, as by construction, shifting lattices by $J_x J_y$ yields the same lattice as the one obtained by shifting the original lattice by $J_y J_x$. Thus, if $J_x$ and $J_y$ are diagonalizable, they can be diagonalized in the same basis $\epsilon_{ij}$, $1\le i,j\le L$. Close inspection of the operators $J_x$ and $J_y$ reveals that they admit matrix representations that are closely related to the canonical $L$-dimensional circulant matrix $J_L$:

$$J_x = \begin{pmatrix} J_L & & \\ & \ddots & \\ & & J_L \end{pmatrix}, \quad J_y = \begin{pmatrix} 0 & I_L & \\ & \ddots & I_L \\ I_L & & 0 \end{pmatrix} \quad \text{and} \quad J_x J_y = J_y J_x = \begin{pmatrix} 0 & J_L & \\ & \ddots & J_L \\ J_L & & 0 \end{pmatrix}. \tag{75}$$

Concretely, the operator $J_x$ cyclically shifts columns within each blocks rank $B_\lambda$, whereas the operator $J_y$ cyclically shifts the blocks within $A_\lambda^{(d)}$. Considering the basis of eigenvector $e_i$, $1\le i\le L$, of $J_L$, we define the basis $\epsilon_{ij}$, $1\le i,j\le L$, as $\epsilon_{ij} = \left(e_j|w_i e_j|\ldots|w_i^{L-1}e_j\right)$, where $w_i$ is the eigenvalue associated to $e_i$. We have

$$J_x\epsilon_{ij} = \left(J_L e_j|w_i J_L e_j|\ldots|w_i^{L-1}J_L e_j\right) = \omega_j\left(e_j|w_i e_j|\ldots|w_i^{L-1}e_j\right) = \omega_j\epsilon_{ij}, \tag{76}$$

$$J_y\epsilon_{ij} = \left(w_i e_j|\ldots|w_i^{L-1}e_j|e_j\right) = \omega_i\left(e_j|w_i e_j|\ldots|w_i^{L-1}e_j\right) = \omega_i\epsilon_{ij}, \tag{77}$$

which shows that $\epsilon_{ij}$ is indeed a basis diagonalizing $J_x$ and $J_y$. Moreover, as $J_x$ and $J_y$ stabilize the space $V_m$, the basis $\epsilon_{ij}$ spans the space $V_m$, as well as all the spaces defined as intersections of subsets of $\{V_m\}_{1\le m\le M}$. Consider the set of indices $1\le m_1<\ldots<m_k<M$, $1\le k\le M$, specifying the intersection $V_{m_1}\cap\ldots\cap V_{m_k}$. By the same reasoning as for dimension 1, the basis elements spanning $V_{m_1}\cap\ldots\cap V_{m_k}$ are those eigenvectors $\epsilon_{ij}$ that are $\gcd(\lambda_{m_1},\ldots,\lambda_{m_k})$-periodic in both $x$-direction and $y$-direction. As $J_x\epsilon_{ij} = \omega_j\epsilon_{ij}$ and $J_y\epsilon_{ij} = \omega_i\epsilon_{ij}$, posing $g = \gcd(\lambda_{m_1},\ldots,\lambda_{m_k})$, this is equivalent to $(gi,gj) = (0,0)$ in $\mathbb{Z}/g\mathbb{Z}\times\mathbb{Z}/g\mathbb{Z}$. By the subgroup structure of cyclic group, the basis elements $\epsilon_{ij}$ generating $V_{m_1}\cap\ldots\cap V_{m_k}$ are thus indexed by $(i,j)$ where  and $j$ are elements of the unique subgroup of order $g$ in $\mathbb{Z}/L\mathbb{Z}$. There are $g^2$ such basis elements, showing that

$$\dim V_{m_1}\cap\ldots\cap V_{m_k} = g^2 = \gcd(\lambda_{m_1},\ldots,\lambda_{m_k})^2. \tag{78}$$

The rank formula follows immediately from expressing $\text{rank}\,A_\lambda^{(d)} = \dim\left(V_1+\ldots+V_M\right)$ via the exclusion-inclusion principle.

(ii) Just as for (i), we follow the same strategy as for dimension 1 to show that the contiguous-separating capacity equals $\text{rank}\,A_\lambda^{(d)}$. The only caveat to address is that the grid-like inputs, that is,

the columns of $A_{\lambda}^{(d)}$, are generated by the action of two shift operators instead of one. Specifically, starting from the first column $c_1$ of $A_{\lambda}^{(d)}$, we can generate all subsequent columns by action of the operators $J_{\lambda,x}$ and $J_{\lambda,y}$, whose matrix representations are given by

$$
J_{\lambda,x} = \begin{pmatrix} J_{\lambda_1 \times \lambda_1} & & & \\ & J_{\lambda_2 \times \lambda_2} & & \\ & & \ddots & \\ & & & J_{\lambda_M \times \lambda_M} \end{pmatrix}, \quad \text{with} \quad J_{\lambda_m \times \lambda_m} = \begin{pmatrix} J_{\lambda_m} & & & \\ & J_{\lambda_m} & & \\ & & \ddots & \\ & & & J_{\lambda_m} \end{pmatrix}, \quad (79)
$$

$$
J_{\lambda,y} = \begin{pmatrix} J'_{\lambda_1 \times \lambda_1} & & & \\ & J'_{\lambda_2 \times \lambda_2} & & \\ & & \ddots & \\ & & & J'_{\lambda_M \times \lambda_M} \end{pmatrix}, \quad \text{with} \quad J'_{\lambda_m \times \lambda_m} = \begin{pmatrix} 0 & I_{\lambda_m} & & \\ & \ddots & \ddots & \\ & & \ddots & I_{\lambda_m} \\ I_{\lambda_m} & & & 0 \end{pmatrix}. \quad (80)
$$

Notice that $J_{\lambda,x}$ and $J_{\lambda,y}$ commute. By the same reasoning as for dimension 1, we know that the separating capacity cannot exceed $\operatorname{rank} A_{\lambda}^{(d)}$. Then, to prove that the separating capacity equals $\operatorname{rank} A_{\lambda}^{(d)}$, it is enough to exhibit a linearly independent set of contiguous positions with cardinality $\operatorname{rank} A_{\lambda}^{(d)}$. Let us exhibit such positions. Mirroring the 1D case, let us consider the sequence $d_l^{(1)}$ defined by

$$
d_l^{(1)} = \dim \operatorname{span} \left\{ J_{\lambda,y}^i c_1 \,|\, 1 \le i \le l \right\}. \quad (81)
$$

The above sequence is strictly increasing by unit step until some $l_1$, after which it remains constant at value

$$
d_{l_1}^{(1)} = \dim V_1, \quad \text{with} \quad V_1 = \operatorname{span} \left\{ J_{\lambda,y}^i c_1 \,|\, 1 \le i \le L \right\}. \quad (82)
$$

Let us then consider the sequence

$$
\dim V_L = \dim \left( V_1, J_{\lambda,x} V_1, \ldots, J_{\lambda,x}^L V_1 \right) = \dim \operatorname{span} \left\{ J_{\lambda,y}^i J_{\lambda,x}^j c_1 \,|\, 1 \le i,j \le L \right\} = \operatorname{rank} A_{\lambda}^{(d)}. \quad (83)
$$

The above sequence is also strictly increasing by unit step until some $l_2$, after which it remains constant at value

$$
d_{l_2}^{(2)} = \dim V_2, \quad \text{with} \quad V_2 = V_1 + J_{\lambda,x} V_1. \quad (84)
$$

Moreover, $V_2$ admits for basis the vectors $J_{\lambda,y}^i c_1$, and $J_{\lambda,y}^i c_1, 1 \le i \le l_1$, $J_{\lambda,y}^i J_{\lambda,x} c_1, 1 \le i \le l_2$. We can iterate this construction by repeated action of the operator $J_{\lambda,x}$, yielding a sequence of number $l_k$ and a sequence of space $V_k = V_{k-1} + J_{\lambda,x} V_k$. Necessarily, the sequence $l_k$ becomes eventually zero as

$$
\dim V_L = \dim \left( V_1, J_{\lambda,x} V_1, \ldots, J_{\lambda,x}^L V_1 \right) = \dim \operatorname{span} \left\{ J_{\lambda,y}^i J_{\lambda,x}^j c_1 \,|\, 1 \le i,j \le L \right\} = \operatorname{rank} A_{\lambda}^{(d)} \quad (85)
$$

Let us consider the smallest $k > 1$ for which $l_k = 0$, than the set of vectors

$$
\{ J_{\lambda,y}^i J_{\lambda,x}^j c_1 \,|\, 1 \le j < k, 0 \le i \le l_k \} \quad (86)
$$

is linearly independent by construction and generates the range of $A_{\lambda}^{(d)}$. In particular, we necessarily have $l_1 + \ldots + l_{k-1} = \operatorname{rank} A_{\lambda}^{(d)}$. Observing that these vectors correspond to a connected component of positions concludes the proof.

## Remark

Although we do not give the proof for arbitrary spatial dimension $d > 2$, let us briefly comment on extending the above arguments to higher dimension. Such a generalization is straightforward but requires the utilization of tensor calculus. For integer periods $\boldsymbol{\lambda}$ and generic dimension $d$, the activity tensor can be defined as

$$\mathcal{A}_{\boldsymbol{\lambda}}^{(d)} = \sum_{i_1,\ldots,i_d \in L^d} \sum_{m=1}^{M} \left( y_{i_1}^m \otimes \ldots \otimes y_{i_d}^m \right) \otimes \left( x_{i_1}^\star \otimes \ldots \otimes x_{i_d}^\star \right) \tag{87}$$

where $y_{i_1}^m \otimes \ldots \otimes y_{i_d}^m$ is the canonical basis vector associated to the $(i_1,\ldots,i_d)$ coordinate in $\mathbb{R}^{\lambda_m^d}$, with $(i_1,\ldots,i_d)$ considered as an element of $\left( \mathbb{Z}/\lambda_m\mathbb{Z} \right)^d$, and where $x_{i_1}^\star \otimes \ldots \otimes x_{i_d}^\star$ is the linear form associated to the $(i_1,\ldots,i_d)$ coordinate in $\mathbb{R}^{L^d}$. In tensorial form, the operators $J_k$, $1 \leq k \leq d$, representing unit shift along the kth dimension, have the simple form $J_k = I_L \otimes \ldots \otimes J_L \otimes \ldots \otimes I_L$ such that

$$J_k \left( x_{i_1} \otimes \ldots \otimes x_{i_k} \otimes \ldots \otimes x_{i_L} \right) = \left( I_L \otimes \ldots \otimes J_L \otimes \ldots \otimes I_L \right) \left( x_{i_1} \otimes \ldots \otimes x_{i_k+1} \otimes \ldots \otimes x_{i_L} \right) \tag{88}$$

where $i_k + 1$ is considered as an element of $\mathbb{Z}/L\mathbb{Z}$. The generalization to arbitrary $d$-dimension follows from realizing that $\epsilon_{i_1,\ldots,i_L} = e_{i_1} \otimes \ldots \otimes e_{i_L}$, $i_1,\ldots,i_d \in L^d$, where $e_i$ is the eigenvector of $J_L$ associated to $\omega_i$, form a basis diagonalizing all the operators $J_k$, $1 \leq k \leq d$ with $J_k \epsilon_{i_1,\ldots,i_L} = \omega_{i_k} \epsilon_{i_1,\ldots,i_L}$.

