## [Decision Letter]

**Acceptance summary:**

Hippocampal place cells and entorhinal grid cells are crucial elements of the spatial representation system of the brain, but the mechanisms underlying their emergence are still poorly understood. A long-standing hypothesis in the field is that the properties of place cells can be well described as a non-linear function of a weighted sum of inputs coming from entorhinal grid cells. In this paper, the authors explore the implications of this scenario, in a simplified model with discretized space, where grid cells are part of a discrete set of modules, and each cell has a perfectly periodic firing in space with a period that depends on the module. They compute analytically the number of possible place field arrangements, and the separating capacity, in this scenario, through a very nice extension of the classic Cover calculation for inputs in general position. These calculations show that the number of possible arrangements is much smaller than when inputs are in general position, but that they are more robust.

**Decision letter after peer review:**

Thank you for submitting your article "Where can a place cell put its fields? Let us count the ways" for consideration by *eLife*. Your article has been reviewed by 3 peer reviewers, and the evaluation has been overseen by a Reviewing Editor and Michael Frank as the Senior Editor. The following individual involved in review of your submission has agreed to reveal their identity: Nicolas Brunel, PhD (Reviewer #2).

The reviewers have discussed the reviews with one another and the Reviewing Editor has drafted this decision to help you prepare a revised submission. The reviewers were generally positive about the manuscript, finding that it significantly expands our understanding of the constraints on place field arrangements arising from grid cell inputs, but they would like to see several revisions and clarifications before being able to recommend it for publication. Please see the list of essential revisions below.

Summary:

Hippocampal place cells and entorhinal grid cells are crucial elements of the spatial representation system of the brain, but the mechanisms underlying their emergence are still poorly understood. A long standing hypothesis in the field is that the properties of place cells can be well described as a non-linear function of a weighted sum of inputs coming from entorhinal grid cells. In this paper, the authors explore implications of this scenario, in a simplified model with discretized space, where grid cells are part of a discrete set of modules, and each cell has a perfectly periodic firing in space with a period that depends on the module. They compute analytically the number of possible place field arrangements, and the separating capacity, in this scenario, through a very nice extension of the classic Cover calculation for inputs in general position. These calculations shows that the number of possible arrangements is much smaller than when inputs are in general position, but that they are more robust.

Essential revisions:

1. The questions that are addressed in the manuscript are interesting mathematically but do not map directly to realistic properties of place cells. The reviewers were concerned that many readers won't understand the limitations. Therefore, the limitations of the approach should be acknowledged and spelled out more clearly. The first question, whether grid cell inputs can produce all possible patterns of place cell activity, is quite detached from biological reality because in the vast majority of these patterns the place cell would fluctuate wildly between on and off states as a function of position, whereas in reality place cells fire sparsely. Importantly, the sparseness is not a conclusion or a prediction of the theory because any degree of sparseness can be easily achieved by varying the threshold. Instead, from the point of view of biological realism, sparseness must be imposed.

The work does consider also patterns that are sparse, having K fields over the whole range of input patterns, where K is small. This question, too, is detached from the reality of place cell firing because place cells would clearly exhibit many firing fields (not just a handful of fields) over the vast range of positions that correspond to all input patterns. Place cells can have multiple firing fields in large continuous environments, and each place cell may have a different field in a significant fraction of small environments. Thus, it is important to consider sparse patterns where the number of firing fields is proportional to the range of positions that are represented by the input patterns. In addition, ideally, it would be interesting to consider this question on a large set of disjoint sets of inputs patterns, each corresponding individually to a continuous stretch of positions (one environment) instead of one long stretch (or the full range). The two cases considered in the work, of arbitrary (dense) patterns and of extremely sparse patterns can be thought of as two extremes where it was possible to derive precise results. These results are suggestive of what might happen with more biologically relevant activity patterns, but the limitations should be acknowledged.

2. The reviewers found the discussion on graded receptive fields (lines 429-438) to be unconvincing, and it may convey an incorrect message about graded receptive fields once noise is taken into account. The argument is based on the observation that graded receptive fields can be related to narrow ones by a linear transformation. If this linear transformation is invertible, it does not alter the set of linearly separable patterns. However, the transformation under consideration is a low-pass filter. For all practical purposes, this transformation, which suppresses high frequency components of the input is non-invertible. The slightest amount of high frequency noise in the grid cell inputs would be dramatically amplified by applying the inverse transformation, and will destroy the correspondence with the case of the narrow input vectors. It is perhaps possible to conduct a more thorough analysis with graded receptive fields, either analytically or numerically. If this is beyond what the authors wish to do in this work, the best course of action might be to acknowledge the limitation of the theory and to leave the question of graded receptive fields open for future study.

3. In the model proposed by the authors, all inputs to a hippocampal place cell are grid cells with perfectly periodic firing in space. This is a very idealized setting that is far from the reality – many cells in entorhinal cortex are far from having spatially periodic firing, and even in those that exhibit strong periodicity, there are often significant variations in the average firing rate from peak to peak. This leads to the concern that purely periodic inputs might not represent the relevant scenario for hippocampal place cells. While the authors discuss briefly the addition of noisy non-periodic inputs at the end of the Results section (Figures 7C and D), they only discuss the robustness of place fields generated by periodic grid inputs to such noisy inputs, but not number of arrangements or separating capacity. The reader is left to wonder how the other results presented in the paper (number of arrangements, separating capacity) are affected by such non-periodic inputs. Are these results still relevant in the presence of realistic heterogeneities?

4. The authors show that beyond the scale of the separating capacity, not all place field arrangements are realizable. Could the authors characterize non-realizable place field arrangements? It would be nice in particular to see specific examples in simple situations like the (3,4) case discussed in Figure 5. It would be even nicer if one can derive general results on such non-realizable arrangements, possibly leading to experimental predictions (see also points 3 and 4 below). In addition, it would be nice if the authors could provide non-trivial predictions about the statistical structure of place cells that are due to the fact that place cells are given from a sum of spatially periodic inputs. An obvious prediction is that one would predict periodicity to appear at a sufficiently large spatial scale, but can one say something about this spatial scale given current data on grid cell periods? Are this, or other, predictions, testable experimentally?

5. Currently available recordings of place cells in large scale environments suggest the statistics of place cells are indistinguishable from a spatial Poisson process (see for instance papers from Albert Lee's lab, in Science (2014) and Cell (2020)). The authors should discuss how their results fit with this picture. It seems in particular that in their model, place fields are consistent with Poisson (in the sense that all possible configurations are possible) on short spatial scales (below the separating capacity), but not on larger spatial scales. Is it possible to characterize deviations from Poissoniality induced by the spatial structure in the inputs?

6. The manuscript (especially the first half) is not particularly easy to read even for a computational neuroscientist and the general conclusion was that for an audience composed mainly of non-theoreticians, it is rather inaccessible. The results (and the ideas behind the analyses) can potentially be understood by a broader audience, but the authors need to make a substantial communication effort. For example, Even the abstract, which should be readily understood by all neuroscientists, takes for granted the meaning of "separating capacity" or "unique input coding range". The abstract should be comprehensible before reading the whole paper (not after). We ask the authors to take care to make sure that their manuscript speaks to a broader audience than those well-versed in the theory behind grid and place cells.

---

## [Author Response]

Essential revisions:1. The questions that are addressed in the manuscript are interesting mathematically but do not map directly to realistic properties of place cells. The reviewers were concerned that many readers won't understand the limitations. Therefore, the limitations of the approach should be acknowledged and spelled out more clearly. The first question, whether grid cell inputs can produce all possible patterns of place cell activity, is quite detached from biological reality because in the vast majority of these patterns the place cell would fluctuate wildly between on and off states as a function of position, whereas in reality place cells fire sparsely. Importantly, the sparseness is not a conclusion or a prediction of the theory because any degree of sparseness can be easily achieved by varying the threshold. Instead, from the point of view of biological realism, sparseness must be imposed.The work does consider also patterns that are sparse, having K fields over the whole range of input patterns, where K is small. This question, too, is detached from the reality of place cell firing because place cells would clearly exhibit many firing fields (not just a handful of fields) over the vast range of positions that correspond to all input patterns. Place cells can have multiple firing fields in large continuous environments, and each place cell may have a different field in a significant fraction of small environments. Thus, it is important to consider sparse patterns where the number of firing fields is proportional to the range of positions that are represented by the input patterns. In addition, ideally, it would be interesting to consider this question on a large set of disjoint sets of inputs patterns, each corresponding individually to a continuous stretch of positions (one environment) instead of one long stretch (or the full range). The two cases considered in the work, of arbitrary (dense) patterns and of extremely sparse patterns can be thought of as two extremes where it was possible to derive precise results. These results are suggestive of what might happen with more biologically relevant activity patterns, but the limitations should be acknowledged.

Thank you for this comment. Indeed, as noted by the reviewer, we have covered two regimes in characterizing realizable field arrangements by place cells driven by grid-like inputs: in one regime we do so without regard to sparseness of the arrangements (Table 1), and in the other, we consider "ultra" sparse arrangements (K-sparse, or K fields/cell, where K is a small fixed number), with a small number of fields that does not scale with the number of modules or module periods (and thus with the full range of the code).

We would very much like to generate results in the intermediate regime where place fields are sparse but scale in number proportionally with the full range, as the reviewers note might be the most biologically relevant case. Mathematically, this involves a constraint that is difficult to implement: in the case of counting arrangements, it involves counting the number of Young diagrams with a fixed area.

However, for both non-sparse field arrangements and ultra-sparse field arrangements (K=1,2,3,…), we find that the grid code enables a large number of field arrangements (e.g. relative to just one-hot input codes; we have now added a comparison of K-field arrangement counting of grid-like inputs with one-hot inputs, which we did not have earlier), that are nevertheless a vanishingly small fraction of all arrangements, leading to our conclusion that the grid code's its modular structure enables the formation of many arrangements but that it simultaneously imposes strong structure on the place field arrangements. Thus, as the reviewer notes, given similar conclusions on two extremes, we may expect similar qualitative results on structure and richness in the intermediate regime of sparse but not ultra-sparse field arrangements. This will be the basis of future work.

In both our Results and Discussion sections, we now explicitly comment that we consider dense and ultra-sparse field arrangements but do not have analytical results for the sparse case.

2. The reviewers found the discussion on graded receptive fields (lines 429-438) to be unconvincing, and it may convey an incorrect message about graded receptive fields once noise is taken into account. The argument is based on the observation that graded receptive fields can be related to narrow ones by a linear transformation. If this linear transformation is invertible, it does not alter the set of linearly separable patterns. However, the transformation under consideration is a low-pass filter. For all practical purposes, this transformation, which suppresses high frequency components of the input is non-invertible. The slightest amount of high frequency noise in the grid cell inputs would be dramatically amplified by applying the inverse transformation, and will destroy the correspondence with the case of the narrow input vectors. It is perhaps possible to conduct a more thorough analysis with graded receptive fields, either analytically or numerically. If this is beyond what the authors wish to do in this work, the best course of action might be to acknowledge the limitation of the theory and to leave the question of graded receptive fields open for future study.

We thank the reviewers for this comment, which has allowed us to improve our argument for the generalization of our results to graded receptive fields. In particular, we discussed that an invertible convolution applied to the {0,1} codewords would generate graded tuning curves, and because the transformation is invertible, the linear separability of the {0,1} original codewords would remain unchanged post-convolution. The reviewer notes that if, after convolution, the codewords were perturbed by noise, an inverse convolution would produce very different states than the original codewords. First, note that in going from binary to smoothed tuning, there is no sense in which the system is "allowed" to add high-frequency noise to the smoothed tuning curves: low-dimensional continuous attractor dynamics keep the tuning curve shapes fixed to a canonical set of translationally shifted smooth shapes, and perturbations to the shape count as off-manifold perturbations that are rapidly erased; any high-dimensional/ high-frequency shape-altering noise is projected onto the nearest point on the low-dimensional manifold, resulting at worst in small shifts in the encoded phases of each grid module (the attractor dynamics also collectively maintains the relative phases of all cells within a module); thus, we should think of the convolved codewords and their relative phases as not subject to noise, and the only noise is in collective shifts of the full module phase relative to the actual spatial position. The mapping from internal coding states to positions is not used for counting arguments, and thus this type of noise is not relevant to our discussions.

Second, the argument for why the convolved codewords possess the same geometry as the uninvolved {0,1} codewords can be made without reference to invertibility of the convolution: If the convolution kernel maintains the sufficient statistic of position phase within each cell and module (and it will do so if the kernel exhibits no periodicity on the scale of the period of each module: thus, it cannot be doubly-bumped within a period, or be constant in amplitude across the period), then: (1) the sufficient statistic of each codeword, the phase encoding of position, is maintained; (2) the cells within each module are still equivalent and can be permuted; (3) the code retains its modular structure, lacking permutation invariance of cells across modules; and (4) the module states can be described as independently updating from each other. These properties mean that the qualitative geometry of the convolved code is again the orthogonal product of simplicies, with the individual simplices having the same geometry as the original {0,1} codeword simplicies. Thus, the counting arguments go through unchanged.

Finally, the effect of the convolution is a rescaling of the sides of the convex polytopes, which will affect the robustness (margins) of the codewords to noise relative to the original {0,1} codewords. We discuss this in the section on margins.

In sum, the counting arguments are not affected by convolution of codewords by kernels that convert {0,1} activations into graded phase-encoding activation profiles. Different encodings of phase will affect the margins and noise-robustness of the resulting field arrangements.

We have replaced our previous argument on the structure of graded grid-like codewords based on invertibility, with the second argument above.

3. In the model proposed by the authors, all inputs to a hippocampal place cell are grid cells with perfectly periodic firing in space. This is a very idealized setting that is far from the reality – many cells in entorhinal cortex are far from having spatially periodic firing, and even in those that exhibit strong periodicity, there are often significant variations in the average firing rate from peak to peak. This leads to the concern that purely periodic inputs might not represent the relevant scenario for hippocampal place cells. While the authors discuss briefly the addition of noisy non-periodic inputs at the end of the Results section (Figures 7C and D), they only discuss the robustness of place fields generated by periodic grid inputs to such noisy inputs, but not number of arrangements or separating capacity. The reader is left to wonder how the other results presented in the paper (number of arrangements, separating capacity) are affected by such non-periodic inputs. Are these results still relevant in the presence of realistic heterogeneities?

Thank you for the opportunity to clarify.

We have shown that the field arrangements that are realizable with grid inputs have bigger margins than if driven by shuffled grid codes and random codes, and thus are more robust to noise (Figure 7a-b). Thus, the existing counting and capacity results will be robust to the addition of noise upto the size of the margins: existing field arrangements will not be destabilized by any noise smaller in size than these broad margins, and the number of realizable arrangements will therefore not decrease.

Moreover, we have shown (Figure 7c-d, filled green violins) that the addition of noise or sparse spatial inputs, in addition to mostly not destroying existing field arrangements, creates new realizable field arrangements: this is because the addition of random inputs to the grid inputs moves the overall input vectors towards more general position. At the same time, however, these additional field arrangements are not stable/robust: their margins are much smaller. We have clarified these points in the manuscript.** **

4. The authors show that beyond the scale of the separating capacity, not all place field arrangements are realizable. Could the authors characterize non-realizable place field arrangements? It would be nice in particular to see specific examples in simple situations like the (3,4) case discussed in Figure 5.

Thank you for this suggestion. We have added examples of non-realizable place field arrangements in the caption of Figure 3. Geometrically, a 2-field arrangement with positive labels for a pair of vertices that are not adjacent (directly connected by an edge) and negative labels for all the rest is not realizable. Conceptually, there are many unrealizable field arrangements (we know most are unrealizable because realizable one are a vanishing fraction) including some obvious ones: for the two-module case with co-prime periods, one cannot have a field arrangement with fields only every other lambda1 (e.g. a periodic arrangement with 2*lambda1). One cannot have a field arrangement with fields only at locations 1 and 2 (two adjacent locations) and nowhere else. This is because for the chosen locations to be above threshold, the periodic nature of the grid drive means that other locations, shifted by multiples of the module periods will also be above threshold. Given the very large set of unrealizable field arrangements, it actually is more tractable to characterize the structure expected in realizable arrangements – please see next response.

It would be even nicer if one can derive general results on such non-realizable arrangements, possibly leading to experimental predictions (see also points 3 and 4 below). In addition, it would be nice if the authors could provide non-trivial predictions about the statistical structure of place cells that are due to the fact that place cells are given from a sum of spatially periodic inputs. An obvious prediction is that one would predict periodicity to appear at a sufficiently large spatial scale, but can one say something about this spatial scale given current data on grid cell periods? Are this, or other, predictions, testable experimentally?

This is a very good question – quantification of what structures are present within the special set of realizable arrangements, which we have counted in this work.

We are in the middle of a separate collaborative theory-experimental work on this question, and to deal extensively with it is beyond the scope of this already very full paper.

We have seen that grid-driven place field arrangements are highly constrained such that only a tiny fraction of potential field arrangements within or across environments are realizable. Realizable arrangements can be understood intuitively with a simple picture: A place cell could choose its input weights and threshold to produce a field at one location. But because grid-cell inputs are multiply peaked and non-local, strengthening weights from grid cells with certain phases and periods to obtain a field at one location means that the place cell will also be strongly driven wherever a similar pattern of inputs recurs in the grid input. This will happen periodically at multiples of the full range *L*, but given that the separating capacity is given by a much smaller range, $\Sigma$, it follows that there should also be visible structure on this scale.

Specifically, we expect to see echoes of the grid structure in both grid-place relationships and in relationships between place fields: (i) Grid-place relationships: A place field strongly driven by a grid cell of a certain phase at one location will be more likely to also be driven by those cells at other locations. Thus, we expect an elevation in the conditional probability, given that a place and grid cell have a coincident field, that the next field by that place cell will also coincide with a field from that grid cell. (ii) Place field relationships: The combined drive of multiple grid periods and phases to a place cell makes its responses appear random (Figure, panel B). However, these realizable arrangements will be geometrically constrained in a scaffold, with more regularity in field spacing over the scale of the summed grid module periods than expected from purely random placement. The inter-field interval (IFI) distributions of place fields, if tested along sufficiently long linear tracks with motion and orienting cues but the absence of many spatially localized landmarks, should exhibit peaks that reflect the combination of inter-field intervals [Yoon et al., 2016] in the underlying periodic grid inputs (Figure, panel C).

5. Currently available recordings of place cells in large scale environments suggest the statistics of place cells are indistinguishable from a spatial Poisson process (see for instance papers from Albert Lee's lab, in Science (2014) and Cell (2020)). The authors should discuss how their results fit with this picture. It seems in particular that in their model, place fields are consistent with Poisson (in the sense that all possible configurations are possible) on short spatial scales (below the separating capacity), but not on larger spatial scales. Is it possible to characterize deviations from Poissoniality induced by the spatial structure in the inputs?

This question is very closely tied to question (4), please see our response above showing that Poisson-like field distributions can be consistent with periodic input drive, even though structure in the interfield intervals is visible over similarly short scales. We also show in the proposed new Figure that the interfield interval distribution quantifies deviations from Poissoniality induced by the structure of the inputs.

6. The manuscript (especially the first half) is not particularly easy to read even for a computational neuroscientist and the general conclusion was that for an audience composed mainly of non-theoreticians, it is rather inaccessible. The results (and the ideas behind the analyses) can potentially be understood by a broader audience, but the authors need to make a substantial communication effort. For example, Even the abstract, which should be readily understood by all neuroscientists, takes for granted the meaning of "separating capacity" or "unique input coding range". The abstract should be comprehensible before reading the whole paper (not after). We ask the authors to take care to make sure that their manuscript speaks to a broader audience than those well-versed in the theory behind grid and place cells.

Thank you very much for this comment. We have significantly edited the full manuscript for clarity, including by improving definitions. We have also: (1) edited the full manuscript, including text, figures, and captions to make it more accessible and clear; this includes the addition of more conceptual and high-level overviews and interpretative descriptions; (2) added a new figure (Figure 3) showing the overall approach of the mathematical computations to follow early in Results, to guide readers at a high level through the conceptual steps; (3) added a note about 1assumptions and limitations as suggested by the reviewers, including about place field sparseness.